# RISE : Regression Imbalance Handling using Switching Experts

## Abstract

Deep Imbalanced Regression (DIR) is challenging due to skewed label distributions and the need to preserve target continuity. Existing DIR methods rely on a single, monolithic model, yet empirical analysis shows that standard benchmarks exhibit strong distributional heterogeneity, exposing a core limitation of such approaches. We theoretically prove that this property creates an irreducible bias for any single model, leading to poor performance in data-scarce regions. This creates a core challenge for algorithmic fairness, as these regions often correspond to marginalized demographic groups. To address this, we propose RISE—Regression Imbalance handling via Switching Experts—a modular Mixture-of-Experts–inspired framework, theoretically motivated by our analysis. RISE employs a novel imbalance-aware algorithm to identify underperforming regions via validation loss and trains dedicated experts with targeted upsampling. As a complementary framework, RISE achieves new state-of-the-art performance while improving fairness, highlighting a principled new direction for imbalanced regression.

## 1 Introduction

Imbalanced data distributions—common in real-world settings—create severe challenges for regression models, producing high variance on minority labels and bias toward majority ones Wang et al. (2020a); Gong et al. (2022). Unlike classification, where imbalance has been extensively studied, Deep Imbalanced Regression (DIR) is more complex due to its continuous and unbounded label space. This limitation has critical fairness implications: in healthcare, underestimating rapid disease progression delays care for underrepresented patients Cross et al. (2024), while in environmental policy, smoothing over pollution spikes overlooks harms concentrated in marginalized communities Su et al. (2024)—highlighting DIR as both a technical challenge and a fairness imperative in high-stakes domains.

In Fig.1, we compare state-of-the-art (SOTA) methods for DIR, including LDS-FDS Yang et al. (2021b) and SRL Dong et al. (2025), on Dataset A Moschoglou et al. (2017). While these approaches reduce training error in tail (few-label) regions, their gains vanish at test time, revealing overfitting and poor generalization on underrepresented labels. Standard remedies such as frequency-based oversampling Steininger et al. (2021) partially close this gap in the tail but consistently degrade performance on head (many-label) regions, exposing a persistent head–tail trade-off Xu et al. (2021). A key observation is that performance across label bands is highly sensitive to the specific sampling realization of the training data, suggesting that the observed dataset is but one draw from a richer underlying distribution, and oversampling schemes represent alternative draws.

We hypothesize that the persistent head–tail discrepancy in DIR arises from two factors: (a) different label regions exhibit distinct, and sometimes conflicting, conditional distributions $P(y|x)$; and (b) a single monolithic model lacks the capacity to jointly capture these heterogeneous mappings Sattler et al. (2020). We empirically validate distributional heterogeneity in standard DIR benchmarks, providing the first direct evidence in this setting. First, independent linear predictors trained on frozen ResNet-50 features for the many-, medium-, and few-label bands of Dataset A and Dataset B Rothe et al. (2018b) yield nearly orthogonal weight vectors, with cosine

**Table 1:** Cosine similarities

| Dataset A | $w_{\text{few}}$ | $w_{\text{med}}$ | $w_{\text{many}}$ |
|---|---|---|---|
| $w_{\text{few}}$ | 1.00 | 0.04 | 0.03 |
| $w_{\text{med}}$ | 0.04 | 1.00 | 0.09 |
| $w_{\text{many}}$ | 0.03 | 0.09 | 1.00 |
| **Dataset B** | $w_{\text{few}}$ | $w_{\text{med}}$ | $w_{\text{many}}$ |
| $w_{\text{few}}$ | 1.00 | 0.02 | 0.03 |
| $w_{\text{med}}$ | 0.02 | 1.00 | 0.18 |
| $w_{\text{many}}$ | 0.03 | 0.18 | 1.00 |

similarities as low as 0.03 (Table 1), indicating fundamentally different predictive functions across

**Figure 1:** Dataset A: SOTA DIR methods cut tail error but worsen head, exposing a persistent head–tail trade-off.

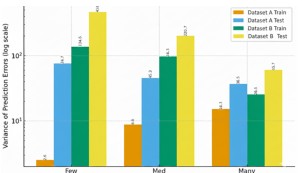

**Figure 2:** Heteroscedasticity in Model Error-SRL Dong et al. (2025)

regions. Second, to demonstrate the consequences of this heterogeneity, we analyze the error profile of a single global model SRL. We find pronounced heteroscedasticity in its prediction errors: on test data, variance in the few-label band is up to $7\times$ higher than in the many-label band, while the opposite trend holds on training data—classic overfitting to scarce samples (Fig. 2). This instability arises precisely because a monolithic model cannot simultaneously capture distinct conditional distributions $P(y|x)$ across regions. Together, these findings show that the core challenge in DIR is not merely label imbalance but distributional heterogeneity, motivating architectures that explicitly specialize across label regions.

This necessitates an architectural shift towards a multi-expert paradigm. We therefore propose RISE (**R**egression **I**mbalance handling using **S**witching **E**xperts), a framework that directly confronts this challenge by learning specialized representations for different data regions. Crucially, RISE is not a generic Mixture of Experts (MoE) Mu & Lin (2025). Its novelty lies in its imbalance-aware algorithm that operationalizes the MoE specifically for DIR. Unlike generic MoEs that partition data by feature similarity, RISE identifies expert domains by analyzing the failure modes of a global model revealed through its validation loss. Each expert is then trained with targeted upsampling, ensuring it focuses on the underrepresented data that challenges a single, monolithic network. This end-to-end approach transforms the MoE from a general tool for heterogeneity into a targeted, principled solution for DIR.

Below we summarize our key contributions:

1. To the best of our knowledge, we are the first to identify and empirically validate that standard DIR benchmarks exhibit distributional heterogeneity, reframing the core challenge from mere label imbalance to representational imbalance.

2. We prove that any monolithic model in DIR suffers from an irreducible heterogeneity bias amplified by imbalance (Theorem 1), and show that targeted expert specialization trades bias reduction against estimation variance (Theorem 2).

3. Building on this, we propose **RISE**, a modular and model-agnostic framework that complements existing SOTA methods by explicitly addressing distributional heterogeneity, overcoming the persistent head–tail trade-off, and improving performance across all regions (as shown in Fig. 1).

4. RISE sets new SOTA on multiple DIR benchmarks Moschoglou et al. (2017), Rothe et al. (2018b), outperforming all baselines, highlighting its effectiveness and establishing a new direction for DIR.

## 2 IMBALANCED REGRESSION PROBLEM FORMULATION

In DIR, we are given a dataset $\mathcal{D} = \{(x_i, y_i)\}_{i=1}^{n}$ with inputs $x_i \in \mathcal{X} \subset \mathbb{R}^p$ and continuous labels $y_i \in \mathcal{Y} \subset \mathbb{R}$. The label marginal $p(y)$ is highly non-uniform (long-tailed), producing majority and scarce (tail) regions where conventional models systematically fail. Motivated by empirical evidence (Sec. 1), we argue that the core difficulty is not merely imbalance in $p(y)$, but a deeper *distributional heterogeneity* in the conditional $P(y \mid x)$. We posit a latent partition of the problem space into $K$ regions, with region $k$ comprising fraction $\rho_k = n_k/n$ of the data and governed by a distinct conditional distribution $P_k(y \mid x)$. Because the fractions $\{\rho_k\}$ are highly non-uniform, a single monolithic predictor trained on the pooled data is dominated by majority regions and induces a persistent bias in scarce ones, a limitation we formalize in Theorem 1. This heterogeneity makes a MoE architecture the natural modeling choice. We therefore model the global conditional distribution

as a mixture of these latent, region-specific distributions:

$$P(y|x) = \sum_{k=1}^{K} \pi_k(x) P_k(y|x), \tag{1}$$

where each component $P_k(y|x)$ is modeled by an expert network $E_k$ and the mixing coefficients $\pi_k(x)$ are determined by a gating network $g_\phi$. The final prediction is the expectation under this mixture: $\hat{y} = \sum_{k=1}^{K} g_\phi(x)_k \cdot E_k(x)$. The learning task is thus transformed from fitting a single complex function into discovering this latent partition (the gate) and learning specialized solutions for each sub-problem (the experts), even when data is sparse—the core challenge our RISE framework is designed to solve.

## 2.1 RELATED WORK

**Deep Imbalanced Regression:** DIR is challenging as it must preserve label continuity under skewed distributions. Prior methods modify loss functions or label densities: LDS-FDS Yang et al. (2021b) and Balanced-MSE Ren et al. (2022) address global imbalance but ignore local heterogeneity; RankSim Gong et al. (2022), ConR Keramati et al. (2024), and SRL Dong et al. (2025) add feature-space regularization (ranking, contrastive, or latent uniformity) yet assume homogeneous features. Regression-via-classification methods Pintea et al. (2023); Pu et al. (2025); Xiong & Yao (2024) discretize labels into fixed bins—a key limitation in continuous regression, where naïve binning often yields scattered or incoherent groups.

**Ensembling and Mixture of Experts:** A common approach to imbalance is partitioning data by class sizes and training separate experts. Ensemble-based methods Xiang et al. (2020); Cui et al. (2023); Cai et al. (2021) follow this strategy in classification but do not extend naturally to regression, where targets are continuous and lack softmax-style aggregation. In long-tailed recognition, multi-expert models such as BBN Zhou et al. (2020) (two-branch fusion for head/tail) and RIDE Wang et al. (2020a) (diversity-regularized experts) reduce bias, yet their applicability to DIR—where label continuity and regional heterogeneity are central—remains unexplored.

# 3 THEORETICAL INSIGHTS: WHY MONOLITHIC MODELS FAIL ON DIR

We formalize the core difficulty we empirically observe in DIR: when data comes from a mixture of region-specific mechanisms, a single global predictor suffers cross-region interference, amplified by label imbalance. To study this, we adopt a simplified linear regression setting, a standard tool for analyzing generalization in complex models Belkin et al. (2018); Lin et al. (2023).

**Setup.** We consider heterogeneous linear regression with $K$ latent regions, each occurring with probability $\rho_k = n_k/n$ (Sec. 2). For a sample $(x, y)$ from region $k$, such that $x \sim \mathcal{N}(0, \Sigma)$, and $y = w_k^{*\top} x + \varepsilon$, where $w_k^* \in \mathbb{R}^p$ is the region-specific parameter, $\varepsilon \sim \mathcal{N}(0, \sigma_k^2)$ is independent noise, and $\Sigma \succ 0$ is the common feature covariance matrix [1]. Heterogeneity is captured entirely by $\{w_k^*\}$, which define distinct $P_k(y \mid x)$. Stacking all $n = \sum_{k=1}^{K} n_k$ samples gives the design matrix $X \in R^{n \times p}$ and the label vector $Y \in R^n$. The pooled(or global) Ordinary Least Squares (OLS) estimator is $\hat{w} = (X^\top X)^{-1} X^\top Y$, trained on all $n$ samples. We evaluate performance by the *region-weighted generalization error*: $\mathcal{G}_\rho(\hat{w}) = \sum_{k=1}^{K} \rho_k \|\hat{w} - w_k^*\|^2$.

**Theorem 1** (Generalization error under imbalance and heterogeneity). *Let* $w_{\text{avg}} = \sum_{k=1}^{K} \rho_k w_k^*$ *and* $\bar{\sigma}^2 = \max_k \sigma_k^2$. *Under Gaussian design with* $n > p + 1$, *the expected region-weighted error of the pooled OLS estimator decomposes as*

$$E[\mathcal{G}_\rho(\hat{w})] = \underbrace{\frac{\bar{\sigma}^2 \operatorname{tr}(\Sigma^{-1})}{n-p-1}}_{\text{Estimation Variance (shrinks with n)}} + \underbrace{\sum_{k=1}^{K} \rho_k \|w_k^* - w_{\text{avg}}\|^2}_{\text{Heterogeneity Bias (persists)}}, \tag{2}$$

---

[1] In Appendix A we relax this assumption to region-dependent covariances $\Sigma_k$ and noise $\sigma_k^2$ and show the same qualitative conclusions hold.

**Proof sketch.** The decomposition follows from $\mathcal{G}_\rho(\widehat{w}) = \|\widehat{w} - w_{\mathrm{avg}}\|^2 + \sum_k \rho_k \|w_k^* - w_{\mathrm{avg}}\|^2$, since $\sum_k \rho_k(w_k^* - w_{\mathrm{avg}}) = 0$. The first term is bounded using inverse-Wishart moments for Gaussian design, yielding the variance term. The second term is deterministic and captures irreducible heterogeneity. Full derivations, and generalizations to $\Sigma_k, \sigma_k^2$ are provided in Appendix A.

**Implications.** Theorem 1 shows that imbalance amplifies heterogeneity: $w_{\mathrm{avg}}$ is dominated by head regions, yielding persistent error on tails when $w_t^*$ lies far away. Even with infinite data, a monolithic model converges to this biased average. Since the Heterogeneity Bias cannot be reduced by more data or reweighting, a natural remedy is architectural: partition the space and assign specialized predictors, so each operates in a more homogeneous region and achieves better generalization.

## 4 PROPOSED METHOD: RISE

Our proposed method, **RISE**, as illustrated in Fig. 3, operates as a complementary framework designed to systematically enhance any pre-trained DIR baseline. Its core architectural choice—replacing a single monolithic model with a system of specialized experts—is a direct response to the distributional heterogeneity we identified in Sec. 1. First, RISE-Identify takes the trained baseline model ($f_\theta$) and analyzes its performance on a held-out validation set to discover its specific failure modes. By using held-out data, we identify regions of true generalization error, not artifacts of training set memorization. Second, RISE-Train creates a set of dedicated experts, each one targeting a specific failure region identified in the first stage. These experts are trained on the train-dataset with targeted upsampling, a strategy that encourages specialization while regularizing against overfitting. Finally, RISE-Inference learns a gating mechanism, also on the held-out set, that dynamically routes new inputs to the most appropriate expert at test time. Complete implementation details and pseudo code are provided in Appendix D.1.

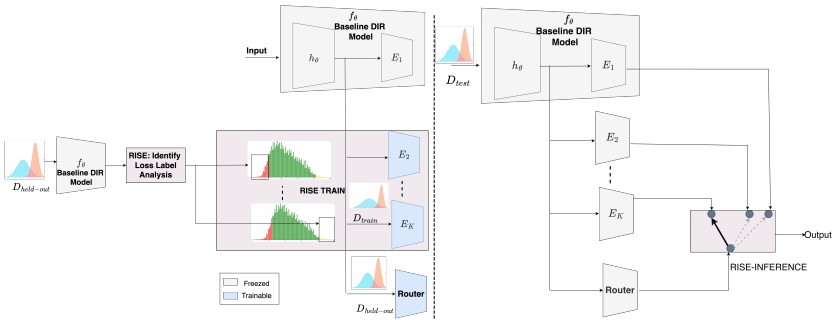

**Figure 3:** Overview of RISE framework.

### 4.1 RISE-IDENTIFY: DISCOVERING LATENT FAILURE REGIONS

The first stage of RISE is to identify the latent regions where a baseline model fails, corresponding to the distinct components of the heterogeneous data distribution we posited in our problem formulation. The overall dataset $\mathcal{D}$ is first split into a training set $\mathcal{D}_{\mathrm{train}}$ and a held-out validation set $\mathcal{D}_{\mathrm{val}}$. A naive approach, implicitly used by frequency-based methods Cui et al. (2023); Yang et al. (2021b), is to partition data using label-density bins from $\mathcal{D}_{\mathrm{train}}$. Specifically, the continuous label space is first discretized into bins, and the frequency of labels in each bin is computed Yang et al. (2021b). A $K'$-component Gaussian Mixture Model (GMM) is then fitted to these frequencies:

**Table 2:** Frequency-Loss Relationship Analysis for Dataset A

| Label Band | Freq | Held-out Loss |
|---|---|---|
| 0-20 | 231 | 8.86 |
| 20-40 | 4,913 | 6.30 |
| 40-60 | 4,609 | 7.44 |
| 60-80 | 2,244 | 7.79 |
| 80-100 | 208 | 9.34 |

$p(\nu) = \sum_{j=1}^{K'} \pi_j' \mathcal{N}(\nu | \mu_j, \sigma_j^2)$ where $\nu$ denotes the bin frequency. Component with the largest weight $\pi_j'$ corresponds to the majority region, while the remaining components capture minority regions. However, this frequency-based approach is a flawed proxy for two key reasons. First, as our analysis shows in Table 2, error (or held-out loss) and frequency are not perfectly correlated; a region can have moderate data density yet still exhibit high generalization error. The 40–60 label band shows

higher loss (7.44) than the 20–40 band (6.30), despite similar sample sizes, indicating that frequency alone does not explain model error—performance is not strictly inversely proportional to frequency, aligning with the observation of Yang et al. (2021b). Second, frequency-based partitioning often creates non-contiguous regions in the label space as shown in Fig. 4a, which is problematic for regression tasks where nearby labels are highly correlated and should be modeled coherently Yang et al. (2021b); Gong et al. (2022).

RISE adopts a more direct and principled strategy: we identify regions based on the model's generalization error, a direct signal of where the single, monolithic model is failing. First, we take a pre-trained DIR baseline, $f_\theta$, trained on $\mathcal{D}_{\text{train}}$. We then use this model to make predictions on the disjoint $\mathcal{D}_{\text{val}}$. For each sample $(x_i, y_i) \in \mathcal{D}_{\text{val}}$, we compute its pointwise prediction error, $e_i = \mathcal{L}(f_\theta(x_i), y_i)$, where $\mathcal{L}$ is a loss function such as the absolute error (L1) or squared error (L2). To identify contiguous regions of high error, we model the joint distribution of these errors and their corresponding labels. This joint modeling ensures that identified regions are contiguous

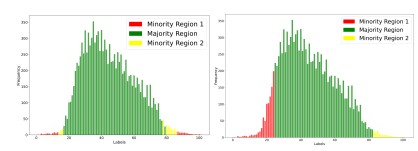

**Figure 4:** Comparison of minority region identification approaches on Dataset A. a) Frequency-based analysis leads to disconnected minority regions (red, yellow) separated by majority regions (green). b) Loss-Label Distribution analysis produces contiguous minority regions

in label space—nearby labels with similar error patterns are grouped together—which is crucial for regression tasks where adjacent target values should be handled by similar predictive functions. Following Yang et al. (2021b) we partition the continuous label range of $\mathcal{D}_{\text{val}}$ into $B$ disjoint, uniform-width bins, $\{B_1, \ldots, B_B\}$. For each bin $b$, we define the set of sample indices it contains as $\mathcal{I}_b = \{i \mid y_i \in B_b\}$ and compute its average generalization error: $\ell_b = \frac{1}{|\mathcal{I}_b|} \sum_{i \in \mathcal{I}_b} e_i$. We model the resulting distribution of (average error, label bin center) pairs, $\{(\ell_b, y_b)\}_{b=1}^B$, using a $K'$-component GMM:

$$p(\ell, y) = \sum_{j=1}^{K'} \pi'_j \mathcal{N}((\ell, y)|\boldsymbol{\mu}_j, \boldsymbol{\Sigma}_j). \tag{3}$$

The GMM naturally clusters the label bins into $K'$ distinct and contiguous performance regions $\mathcal{R}'_j$. The component with the lowest mean error (along the error dimension of $\boldsymbol{\mu}_j$) is designated the well-performing "majority" region, while the remaining components correspond to distinct failure modes requiring specialized experts. $K'$ is a hyperparameter that needs to be tuned. Using a held-out set ensures the identified regions correspond to true generalization failures, not training set memorization—a phenomenon we explicitly observe (Fig. 1).As illustrated in Fig. 4b, this approach produces continuous minority regions, aligning with the principle of region similarity and enabling more homogeneous expert training. By defining regions based on error, we directly target the heterogeneity bias identified as the key limitation in Theorem 1.

## 4.2 RISE-Train: Expert Training

Having identified the baseline model's failure regions, the next stage is to train a dedicated expert for each one. To maintain computational efficiency and leverage the powerful representations learned by the baseline $f_\theta$, we adopt a parameter-efficient fine-tuning approach Kirichenko et al. (2023). As shown in Fig. 3 each expert $E_j$ shares the frozen backbone of the pre-trained model; only its final layers are trained for specialization. For RISE-Train, we evaluate two strategies for expert specialization. A naive strategy (**T1: Subgroup-Specific Training**) trains each expert exclusively on its assigned data partition. This hard partitioning forces experts to learn from a severely restricted support of the data distribution, inducing high estimation variance and overfitting. It also prevents learning smooth functions across the label space in regression, leading to poor generalization and discontinuities at region boundaries. We therefore propose a more robust and principled strategy: **T2: Cross-Group Training with Upsampling**. For each expert $E_j$, we train on the full dataset $\mathcal{D}_{\text{train}}$ with sample weights: target region samples $\mathcal{R}'_j$ are upsampled by $\alpha_j > 1$, others by 1. This weighted empirical risk minimization both regularizes and specializes: exposure to the full dataset prevents high variance and discontinuities by constraining experts to remain well-behaved across the manifold, while $\alpha_j > 1$ amplifies gradients from $\mathcal{R}'_j$, biasing the expert toward its designated failure regions. Our ablations (Sec. 6.3) empirically confirm that T2 is a far superior strategy, and we adopt it for all experiments.

## 4.3 RISE-INFERENCE: EXPERT SWITCHING STRATEGY

The final stage of RISE is to dynamically route each new input to the most suitable expert at test time. We compare three strategies: **(I1) Expert Averaging**, a simple ensemble baseline which aggregates predictions from all experts via weighted averaging; **(I2) Train-based Router**, a gating network $g_\phi$ trained on the training set $\mathcal{D}_{\text{train}}$, our proposed **(I3) Held-out-based Router**; and **(I4) Train + Held-out based Router**, a gating network $g_\phi$ trained on the held-out validation set $\mathcal{D}_{\text{val}}$. Our ablations (Sec. 6.4) show that the held-out router (I3) is decisively superior. A router trained on $\mathcal{D}_{\text{train}}$ tends to select experts that best fit training artifacts, whereas training on $\mathcal{D}_{\text{val}}$ turns routing into a meta-learning task: it learns to pick the expert that *generalizes* best. We therefore adopt I3 as our standard strategy. The router is implemented as a small multi-layer perceptron (MLP) that takes the shared features from the baseline's backbone as input and outputs a probability distribution over the $K'$ experts. The final prediction $\hat{y}$ is the output of the single expert selected by the router: $\hat{y} = E_{j^*}(x)$, where $j^* = \arg\max_j g_\phi(x)_j$.

## 5 THEORETICAL JUSTIFICATION FOR RISE

Having established in Theorem 1 that a global model suffers an irreducible heterogeneity bias, the natural question is: under what conditions can a region-specialized architecture overcome this limitation? We provide a formal result showing when RISE strictly outperforms the pooled model.

**Theorem 2** (Generalization advantage of RISE). *Building on the heterogeneous regression setup of Theorem 1, let $K'$ experts be trained with per-region upsampling factors $\alpha_j$ and routing probabilities $q_k(j)$ (the probability that a sample from region $k$ is assigned to expert $j$). The effective sample size for expert $j$ is $n_{\text{eff}}^{(j)} = (\alpha_j - 1)n_j + n$. From Theorem 1 we know the pooled/global model incurs region-weighted risk (or generalisation error) $\mathcal{G}_{\text{pooled}} = V_{\text{glob}} + \Delta_{\text{glob}}$, where $\Delta_{\text{glob}} = \sum_{k=1}^{K} \rho_k \|w_k^* - w_{\text{avg}}\|$, and $V_{\text{glob}} = O(p/n)$ is the estimation variance of the global model, while generalisation error of RISE satisfies*

$$\mathcal{G}_{\text{RISE}} = B_{\text{det}}(\alpha, q) + V_{\text{est}}(\alpha, q) + R_{\text{cross}}(\alpha, q),$$

*where $B_{\text{det}}$ is deterministic bias from imperfect specialization (including possible $K' \neq K$ or overlapping experts), $V_{\text{est}}(\alpha, q) = O(p/n_{\text{eff}}^{(j)})$ is expert estimation variance, and $R_{\text{cross}}(\alpha, q) = O(\sqrt{p/n_{\text{eff}}^{(j)}})$ are vanishing cross-terms. RISE outperforms the pooled model whenever*

$$\Delta_{\text{glob}} - B_{\text{det}}(\alpha, q) > V_{\text{est}}(\alpha, q) - V_{\text{glob}} + R_{\text{cross}}(\alpha, q).$$

**Proof Sketch and Implications.** The pooled model converges to the data-weighted average $w_{\text{avg}}$, incurring a persistent heterogeneity bias $\Delta_{\text{glob}}$. RISE reduces this bias by upsampling scarce regions and routing them to specialized experts, so their effective targets move closer to $w_k^*$. Any mismatch between the number of experts and true regions ($K' \neq K$ or overlaps) is absorbed into the deterministic bias term $B_{\text{det}}(\alpha, q)$. The trade-off is increased finite-sample variance $V_{\text{est}}(\alpha, q) = O(p/n_{\text{eff}}^{(j)})$ and negligible cross-terms $R_{\text{cross}}(\alpha, q) = O(\sqrt{p/n_{\text{eff}}^{(j)}})$, both of which decay with sample size. Thus, whenever the bias reduction dominates these penalties, RISE achieves strictly better generalization than the pooled model. Detailed proofs are in Appendix B. In practice, imbalance-aware upsampling (RISE-Train T2) increases $n_{\text{eff}}$ in scarce regions and the learned router (RISE-Inference I3) keeps the maximum routing error $\epsilon = \max_k(1 - q_k(k))$ small, directly satisfying the theorem's condition $\mathcal{G}_{\text{RISE}} < \mathcal{G}_{\text{pooled}}$. We provide empirical validation of this effect in Sec. 6.5.

## 6 EXPERIMENTS AND RESULTS

We evaluate the utility of RISE through the following research questions-

- **RQ1:** How effective is RISE compared to SOTA baselines across different datasets?
- **RQ2:** How do expert training strategies and hyperparameters affect RISE performance?
- **RQ3:** How do different RISE-INFERENCE strategies affect overall performance?
- **RQ4:** How Practically Achievable are the Theoretical Conditions (Theorem 2) for RISE's Success?
- **RQ5:** Do RISE's performance gains stem from its specialized architecture or model capacity?

## 6.1 EXPERIMENTAL SETUP

**Algorithms:** We compare RISE with four SOTA DIR methods and a *Vanilla* ResNet-50 backbone He et al. (2016). Since RISE is a modular framework that complements existing approaches, we evaluate it in combination with *Vanilla*, *LDS+FDS* Yang et al. (2021b), *RankSIM* Gong et al. (2022), *BalancedMSE* Ren et al. (2022), and *SRL* Dong et al. (2025). For baselines we use released weights or official implementations. All RISE experts are trained with MSE loss. We tune the number of experts $K'$, upsampling ratio $\alpha$, selecting the best configuration by validation performance. Further details are in Appendix D.2.

**Datasets:** We evaluate RISE on four DIR benchmarks across modalities: Dataset A Moschoglou et al. (2017)(images, target values in range 0–101), Dataset B Rothe et al. (2018b) (images, range 0–186), STS-B Cer et al. (2017a) from GLUE Wang et al. (2018) (text, similarity 0–5), and UCI-Abalone Nash et al. (1994) (tabular, range 1–29). Following confidentiality requirements, we anonymize Dataset A and Dataset B by omitting their names. Full details are in Appendix C.

**Metrics:** Following Yang et al. (2021b); Gong et al. (2022); Dong et al. (2025), we report performance overall and across Many (>100 samples), Medium (20–100), and Few (<20) label bands. For Dataset A and B, we use Mean Absolute Error (MAE)↓, Mean Squared Error (MSE)↓, and Geometric Mean Error (GMEAN)↓. For STS-B, we additionally report Pearson↑ and Spearman↑ correlation. To assess fairness — defined as minimizing performance disparities across these bands—we also report balanced-MAE (bMAE)↓ Ren et al. (2022), which averages MAE over uniformly partitioned label bins to capture regional performance gaps (see Appendix Section E.2).

## 6.2 RQ1: PERFORMANCE OF RISE ON PUBLIC BENCHMARK DATASETS

Table 4 shows that RISE consistently improves strong baselines (*LDS+FDS, RankSIM, SRL*) on Dataset A across all label bands ( similar results for other datasets are provided in Appendix Sec. E.1 ). The largest relative gains occur in the Few and Medium regions, where monolithic models suffer most. For example, *SRL+RISE* reduces Few-MAE by $15\%$ while simultaneously lowering Many-MAE by $10\%$, thereby overcoming the common head-tail performance trade-off. The performance gains from RISE scale directly

**Table 3:** Balanced-MAE (bMAE) ↓ on Dataset A

| | bMAE ↓ | | | |
|---|---|---|---|---|
| Method | All | Many | Med | Few |
| SRL | 8.32 | 6.64 | 8.34 | 11.74 |
| SRL + RISE | **7.39** | **6.00** | **7.25** | **10.33** |

with the quality of the learned router. Weak backbones (e.g., *Vanilla*, with a router accuracy of $\approx 0.44$) lead to unstable tail performance. In contrast, strong backbones (e.g., *SRL*, with a router accuracy of $\approx 0.87$) enable RISE to fully realize the theoretical advantage of specialization (Theorem. 2). This confirms that the benefit from reducing heterogeneity bias dominates once the routing error is sufficiently low, while the variance cost remains controlled. Additional results (Appendix E.4) show that using an optimal router trained on the best feature representation yields significantly better performance than the baseline router, due to higher routing accuracy.

We assess fairness via bMAE in Table 3 (full results in Appendix Sec. E.2). By significantly improving Few and Medium-band performance while preserving Many-band accuracy, *SRL+RISE* directly mitigates the bias towards head regions exhibited by the baseline. This reduces performance disparities across label bands and demonstrably more equitable performance across all label bands.

## 6.3 RQ2: ABLATION ON EXPERT TRAINING AND HYPERPARAMETERS

We ablate RISE's core design choices on Dataset A with SRL as backbone in Tables 5 and 6. Results on Dataset B is in Appendix Sec. E.3. Our adopted expert training strategy, **T2** (full-dataset training with region specific upsampling), consistently outperforms **T1** (region specific training). T1's hard partitioning causes severe overfitting, whereas T2's full-dataset exposure acts as a powerful regularizer that promotes smooth generalization while upsampling encourages specialization. Our analysis of the number of experts ($K'$) and upsampling ratio ($\alpha$) reveals a clear U-shaped performance curve. This empirically validates our theory's cost-benefit trade-off (Theorem 2) and directly operationalizes it: the upsampling factor $\alpha$ is a key lever to control the expert's estimation variance ($V_{\text{est}}$) while still achieving the primary goal of reducing heterogeneity bias ($B_{\text{det}}$). Performance peaks at moderate values (e.g., $K' = 3, \alpha = 3$) before degrading as the costs of data fragmentation and overfitting outweigh the benefits of heterogeneity reduction.

**Table 4:** Results on Dataset AMoschoglou et al. (2017). For each baseline/RISE pair, the better score is in **bold**; the best overall is underlined. Router accuracy of RISE is shown in parentheses.

| Method | L1 (MAE) ↓ | | | | GMEAN ↓ | | | | MSE ↓ | | | |
|---|---|---|---|---|---|---|---|---|---|---|---|---|
| | All | Many | Med | Few | All | Many | Med | Few | All | Many | Med | Few |
| VANILLA | 11.05 | 9.96 | 12.79 | **16.53** | 7.06 | 6.27 | 8.37 | 13.48 | 202.09 | 165.09 | 270.75 | **361.74** |
| +RISE (0.44) | **10.43** | **9.40** | **11.62** | 16.93 | **6.55** | **5.85** | **7.47** | **13.16** | **181.61** | **148.38** | **221.57** | 384.95 |
| BalancedMSE | 8.70 | 8.44 | 8.99 | 10.26 | 5.58 | 5.44 | 5.87 | **6.17** | 127.05 | 118.69 | 133.94 | **187.01** |
| +RISE (0.47) | **7.71** | **7.23** | **8.16** | **10.02** | **4.83** | **4.52** | **5.10** | 6.87 | **103.39** | **91.14** | **114.84** | 187.41 |
| LDS+FDS | 7.47 | 6.91 | 8.27 | 10.58 | 4.77 | 4.44 | 5.33 | 6.87 | 95.32 | 79.71 | 118.52 | 178.58 |
| +RISE (0.53) | **7.28** | **6.79** | **8.07** | **9.72** | **4.49** | **4.25** | **4.88** | **6.04** | **92.79** | **78.88** | **116.49** | **158.63** |
| RankSIM | 7.02 | 6.58 | 7.86 | 9.72 | 4.55 | 4.14 | 5.39 | 6.97 | 83.55 | 74.34 | 99.30 | 149.51 |
| +RISE (0.54) | **6.94** | **6.50** | **7.38** | **9.10** | **4.35** | **4.08** | **4.80** | **6.04** | **82.70** | **71.96** | **91.20** | **138.15** |
| SRL | 7.23 | 6.64 | 8.28 | 9.85 | 4.53 | 4.17 | 5.32 | 6.35 | 91.79 | 77.20 | 115.83 | 163.15 |
| +RISE (0.87) | **6.57** | **6.16** | **7.36** | **8.30** | **3.61** | **3.40** | **4.14** | **4.33** | **82.01** | **70.88** | **91.20** | **134.93** |

**Table 5:** Ablation on Dataset A: MAE for varying upsampling (with fixed $K' = 3$, left) and varying experts (with fixed $\alpha = 3$, right). Best RISE configuration beating baseline SRL is in **bold**.

| Config | L1 (MAE) ↓ | | | |
|---|---|---|---|---|
| | All | Many | Med | Few |
| SRL | 7.23 | 6.64 | 8.28 | 9.85 |
| SRL+RISE | | | | |
| $\alpha=2$ | 6.72 | 6.23 | 7.69 | 8.66 |
| $\alpha=3$ | **6.57** | **6.16** | **7.36** | **8.30** |
| $\alpha=4$ | 6.73 | 6.32 | 7.51 | 8.43 |
| $\alpha=5$ | 6.89 | 6.52 | 7.49 | 8.68 |

| Config | L1 (MAE) ↓ | | | |
|---|---|---|---|---|
| | All | Many | Med | Few |
| SRL | 7.23 | 6.64 | 8.28 | 9.85 |
| SRL+RISE | | | | |
| $K'=2$ | 6.88 | 6.41 | 7.70 | 9.06 |
| $K'=3$ | **6.57** | **6.16** | **7.36** | **8.30** |
| $K'=4$ | 6.89 | 6.48 | 7.38 | 9.29 |
| $K'=5$ | 7.29 | 6.93 | 7.67 | 9.58 |

## 6.4 RQ3: Ablation on RISE-Inference Strategies

We compare four routing strategies as mentioned in Section 4.3 on Dataset A in Table 7 (full results in Appendix Sec. E.3 and detailed ablation in Appendix Sec. I): **I1** (expert averaging), **I2** (router trained on the training set), our proposed **I3** (router trained on a held-out validation set) and **I4** (router trained on train+held-out dataset). We observe that I3 is significantly superior. The reason is fundamental—routers trained on the training set (I2 & I4) overfit to features already captured by experts, whereas I3 learns which expert generalizes best, providing a robust signal for routing. This confirms that RISE's advantage stems from its effective use of held-out data for what is essentially a meta-learning task—learning to select the best generalizing expert.

**Table 6:** Ablation of RISE-Expert training. Best results in **bold**. Full results in Appendix Sec. E.3

| Method | L1 (MAE) ↓ | | | |
|---|---|---|---|---|
| | All | Many | Med | Few |
| RISE (T1) | 7.23 | 6.77 | 7.95 | 9.61 |
| RISE (T2) | **6.57** | **6.16** | **7.36** | **8.30** |

**Table 7:** Ablation of RISE inference strategies. Best results in **bold**. Full results in Appendix Sec. E.3

| Method | L1 (MAE) ↓ | | | |
|---|---|---|---|---|
| | All | Many | Med | Few |
| Baseline SRL | 7.23 | 6.64 | 8.28 | 9.85 |
| Expert average (I1) | 7.23 | 6.72 | 8.13 | 9.54 |
| Train-based Router (I2) | 7.26 | 6.61 | 8.34 | 10.33 |
| Held-out-based Router (I3) | **6.57** | **6.16** | **7.36** | **8.30** |
| Train+Val based Router (I4) | 7.24 | 6.65 | 8.31 | 9.98 |

## 6.5 RQ4: Empirical validation of Theoretical Trade-offs in Practice

Theorem 2 predicts that RISE outperforms a pooled (or monolithic) model whenever the bias reduction from specialization outweighs the added estimation variance and routing cost. To empirically validate this, we conduct controlled experiments on Dataset A with RISE using SRL as the backbone. We simulate router behavior with accuracy $p \in \{0.01, \ldots, 1.0\}$, where the correct expert is chosen with probability $p$. We systematically vary (i) the upsampling factor $\alpha$ (Fig. 5) and (ii) the number of experts $K'$ (Fig. 6), averaging over 20 trials. Keeping fixed $K' = 3$, Fig. 5 shows that higher upsampling reduces error under accurate routing but increases sensitivity to poor routing, consistent with $\alpha$ reducing bias while amplifying variance. Keeping $\alpha = 3$ fixed in Fig. 6 shows that larger

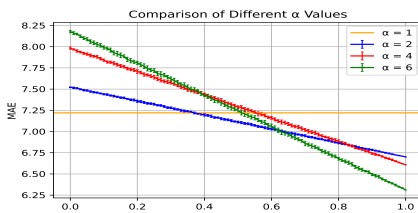
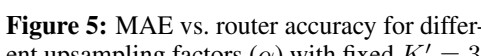
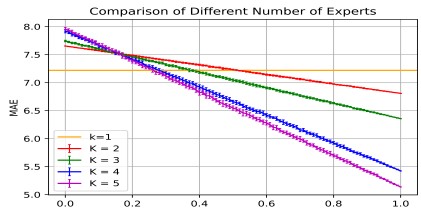

**Figure 5:** MAE vs. router accuracy for different upsampling factors ($\alpha$) with fixed $K' = 3$

**Figure 6:** MAE vs. router accuracy for varying numbers of experts with fixed $\alpha = 3$

$K'$ improves accuracy when routing is reliable but offers diminishing returns and greater instability when routing is noisy. In both cases, $\alpha = 1$ or $K' = 1$ reduces RISE to the pooled baseline (or standalone SRL model). Overall, the empirical gain, $\mathcal{G}_{\text{pooled}} - \mathcal{G}_{\text{RISE}}$, becomes positive once router accuracy exceeds $\sim 60\%$ (with moderate $\alpha, K'$), confirming that RISE successfully operationalizes the theoretical trade-off and remains robust to realistic routing imperfections.

### 6.6 RQ5: Ablation: RISE vs. High-Capacity Ensembles.

A critical question is whether RISE's gains stem from its principled architecture or simply from an increased parameter count. To isolate this, we compare RISE against strong, high-capacity ensembles (Table 8, complete results in Appendix Sec. E.3). We train ensembles of 3 and 5 SRL models resulting in significantly additional model size than a RISE-augmented model, where each member is trained on a random data

**Table 8:** Comparison of RISE vs. traditional ensembles on Dataset A. Best results in **bold**

| Experiment | Additional Parameters | L1 (MAE) ↓ | | | |
|---|---|---|---|---|---|
| | | All | Many | Median | Few |
| *SRL* | 0 | 7.23 | 6.64 | 8.28 | 9.85 |
| *SRL*+ RISE (K'=3) | +2,100,224 | **6.57** | **6.16** | **7.36** | **8.30** |
| *SRL*: 3 ensemble | +3,150,336 | 7.22 | 6.63 | 8.28 | 9.86 |
| *SRL*: 5 ensemble | +5,250,560 | 7.22 | 6.62 | 8.30 | 9.90 |

subset to induce diversity. We observe that RISE consistently and significantly outperforms these ensembles, even with their much higher capacity. This highlights a fundamental architectural difference. Standard ensembles create diversity through *unstructured*, random data sampling. In contrast, RISE employs a *principled, structured specialization*: it uses validation loss to deterministically identify the model's specific failure modes and trains experts to explicitly target those weaknesses. This confirms that RISE's performance gains are not a product of raw model capacity but are a direct result of its intelligent, data-driven approach to resolving distributional heterogeneity.

## 7 Conclusion, Broader Impact, and Limitations

We presented RISE (Regression Imbalance handling via Switching Experts), a novel framework that addresses the fundamental challenge of distributional heterogeneity in Deep Imbalanced Regression (DIR). RISE employs a three-stage approach: identifying failure regions via validation loss analysis rather than frequency-based heuristics, training experts with cross-group upsampling to encourage specialization while maintaining smoothness, and learning a gating mechanism, that dynamically routes new inputs to the most appropriate expert at test time. This approach consistently outperforms existing methods, improving both predictive accuracy and fairness, especially for underrepresented regions of the target distribution. RISE is broadly applicable to any regression problem with imbalance issues, advancing the development of more reliable and fair AI systems for critical decision-making.

**Limitations:** RISE introduces additional computational overhead due to training multiple experts and a router network; however, this is partially offset by training experts on last-layer features only. The framework also depends on a high-quality, representative validation set for effective minority subgroup identification and router training. The method's performance and fairness gains can degrade if the validation set is noisy or biased, potentially reinforcing existing biases through expert specialization. Future work could explore adaptive validation strategies and more efficient training schemes to further mitigate these limitations.

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

# APPENDIX

## A Proof of Theorem 1

We restate Theorem 1 from the main paper, together with its assumptions, before presenting a complete proof and a refinement using matrix concentration to obtain a tighter bound. In addition, we extend the analysis to *region-dependent feature covariances*, where feature distributions may differ across regions, to make the theory more realistic. This extension leads to the same qualitative conclusion as in the main paper.

### A.1 Assumptions and notation

We work in the classical fixed-$p$ regime. Let $p, n \in$ with $n > p + 1$. The condition $n > p + 1$ ensures that the expectation of the inverse-Wishart distribution exists, which is needed to evaluate the estimation variance. For vectors and matrices we use the Euclidean norm $\| \cdot \|$ and the spectral norm $\| \cdot \|_{\mathrm{op}}$; $(\cdot)$ denotes the trace.

**Assumption 1** (Gaussian design). *Fix a positive definite covariance matrix $\Sigma \in R^{p \times p}$ with eigenvalues $0 < \lambda_{\min}(\Sigma) \leq \lambda_{\max}(\Sigma) < \infty$. Let $x_1, \ldots, x_n \overset{i.i.d.}{\sim} \mathcal{N}(0, \Sigma)$ be the design rows, stacked into $X \in R^{n \times p}$.*

*The label space is partitioned into $K$ regions indexed by $k = 1, \ldots, K$. Each observation $i$ has a region label $z_i \in \{1, \ldots, K\}$, drawn independently of $X$, with*

$$P(z_i = k) = \rho_k, \qquad \rho_k > 0, \quad \sum_{k=1}^{K} \rho_k = 1.$$

*Let $n_k = \sum_{i=1}^{n} \mathbf{1}\{z_i = k\}$ be the (random) region counts, with $\mathbf{E}[n_k] = n\rho_k$.*

**Remark 1** (On independence of $z$ and $x$). *The assumption $z_i \perp x_i$ is restrictive but crucial for tractability. In practice (e.g., econometrics, biostatistics), features are often predictive of group membership, in which case off-diagonal terms would appear in conditional covariances and the analysis would require more advanced tools.*

**Assumption 2** (Linear region-specific models). *For each region $k$ there exists a parameter vector $w_k^* \in R^p$. Observations in region $k$ follow*

$$y_i = x_i^\top w_{z_i}^* + \varepsilon_i, \qquad \varepsilon_i \overset{ind}{\sim} \mathcal{N}(0, \sigma_{z_i}^2),$$

*with $\varepsilon_i$ independent of $x_i$ and other noise variables. Label vector $y \in R^n$.*

*Define the population-weighted average parameter*

$$w_{\mathrm{avg}} := \sum_{k=1}^{K} \rho_k w_k^*,$$

*and the centered deviations*

$$v_k := w_k^* - w_{\mathrm{avg}}, \qquad \sum_{k=1}^{K} \rho_k v_k = 0.$$

[Pooled OLS and risk] The pooled ordinary least squares estimator is

$$\widehat{w} = (X^\top X)^{-1} X^\top y,$$

which is well-defined almost surely for $n > p$.

We measure performance by the $\rho$-weighted mean squared error

$$\mathcal{G}_\rho(\widehat{w}) = \sum_{k=1}^{K} \rho_k \|\widehat{w} - w_k^*\|^2.$$

## A.2 EXACT DECOMPOSITION OF EXPECTED ERROR

**Theorem 1.** *Under Assumptions 1–2, for $n > p + 1$,*

$$\mathbf{E}\big[\mathcal{G}_\rho(\widehat{w})\big] = \underbrace{\mathbf{E}\big[\|(X^\top X)^{-1}X^\top \varepsilon\|^2\big]}_{\text{estimation variance}} + \underbrace{\mathbf{E}\big[\|(X^\top X)^{-1}X^\top \delta\|^2\big]}_{\text{mismatch term}} + \underbrace{\Delta}_{\text{irreducible heterogeneity}}, \qquad (4)$$

*where $\delta \in R^n$ has entries $\delta_i = x_i^\top v_{z_i}$, and*

$$\Delta := \sum_{k=1}^K \rho_k \|w_k^* - w_{\text{avg}}\|^2 = \sum_{k=1}^K \rho_k \|v_k\|^2.$$

*Moreover,*

$$\mathbf{E}\big[\|(X^\top X)^{-1}X^\top \varepsilon\|^2\big] = \frac{(\Sigma^{-1})}{n - p - 1}\Big(\sum_{k=1}^K \rho_k \sigma_k^2\Big). \qquad (5)$$

*Proof.* Expanding $\sum_k \rho_k \|\widehat{w} - w_k^*\|^2$ yields

$$\mathcal{G}_\rho(\widehat{w}) = \|\widehat{w} - w_{\text{avg}}\|^2 + \Delta.$$

The response can be written as $y = Xw_{\text{avg}} + \delta + \varepsilon$, where $\delta_i = x_i^\top v_{z_i}$. Therefore

$$\widehat{w} - w_{\text{avg}} = (X^\top X)^{-1}X^\top(\delta + \varepsilon).$$

Squaring gives

$$\|\widehat{w} - w_{\text{avg}}\|^2 = \|(X^\top X)^{-1}X^\top \varepsilon\|^2 + \|(X^\top X)^{-1}X^\top \delta\|^2$$
$$+ 2\langle (X^\top X)^{-1}X^\top \varepsilon, (X^\top X)^{-1}X^\top \delta\rangle.$$

Taking expectation: the cross-term vanishes because conditional on $(X, z)$, $\delta$ is fixed and $\mathbf{E}[\varepsilon|X, z] = 0$. This proves (4).

For (5), let $A = (X^\top X)^{-1}X^\top$. Then

$$\mathbf{E}\|A\varepsilon\|^2 = \mathbf{E}\big(A\,\mathbf{E}[\varepsilon\varepsilon^\top|z]\,A^\top\big)$$
$$= \mathbf{E}_{X,z}\Big((X^\top X)^{-1}X^\top(\sigma_{z_1}^2, \ldots, \sigma_{z_n}^2)X(X^\top X)^{-1}\Big).$$

Independence of $z$ and $X$ implies

$$\mathbf{E}_z[(\sigma_{z_1}^2, \ldots, \sigma_{z_n}^2)] = \Big(\sum_{k=1}^K \rho_k \sigma_k^2\Big)I_n.$$

Thus

$$\mathbf{E}\|A\varepsilon\|^2 = \Big(\sum_k \rho_k \sigma_k^2\Big)\mathbf{E}((X^\top X)^{-1}).$$

Since $X^\top X \sim \mathcal{W}_p(\Sigma, n)$,

$$\mathbf{E}[(X^\top X)^{-1}] = \frac{\Sigma^{-1}}{n - p - 1}, \qquad n > p + 1,$$

hence the trace formula (5). $\qquad \square$

## A.3 REMARKS

- The decomposition (4) provides a transparent separation of error sources: (i) variance due to noise, (ii) a design-dependent mismatch term induced by parameter heterogeneity, and (iii) the irreducible population heterogeneity $\Delta$.

- The estimation variance admits an exact closed form (5), scaling as $O(1/n)$ for fixed $p$.

- The mismatch term is always nonnegative. Its precise asymptotics depend on higher-order Wishart moment identities; deriving tight general rates is delicate and left for future work.

- As $n \to \infty$ with $p$ fixed, the total expected error approaches $\Delta$, which is the asymptotic bias from heterogeneity.

- Ill-conditioning of $\Sigma$ (large $(\Sigma^{-1})$) inflates the variance term and slows convergence to $\Delta$.

- These conclusions hold in the fixed-$p$, large-$n$ regime. In high-dimensional settings with $p/n \not\to 0$, ridge regularization and random matrix theory tools are needed.

**Assumption 3** (Sub-Gaussian heterogeneous design). *For each region $k$, the covariates $x_{k,i}$ are independent mean-zero $K_\psi$–sub-Gaussian vectors with covariance $\Sigma_k \succ 0$, i.e. for every unit vector $u \in R^p$ and $t \in R$,*

$$E \exp\left(t\, u^\top x_{k,i}\right) \le \exp\left(K_\psi^2 t^2/2\right).$$

*Define the mixture covariance*

$$\Sigma_{\mathrm{mix}} := \sum_{k=1}^K \rho_k \Sigma_k,$$

*and assume $\lambda_{\min}(\Sigma_{\mathrm{mix}}) > 0$.*

**Proposition 1** (Sample-covariance concentration). *Under Assumption 3, there exist constants $c_0, C_0 > 0$ depending only on $K_\psi$ such that if $n \ge C_0\left(p + \log(1/\delta)\right)$ then with probability at least $1 - \delta$,*

$$\|\widehat{\Sigma} - \Sigma_{\mathrm{mix}}\|_{\mathrm{op}} \le c_0 \|\Sigma_{\mathrm{mix}}\|_{\mathrm{op}} \sqrt{\frac{p + \log(1/\delta)}{n}}.$$

*Consequently, on this event $\lambda_{\min}(\widehat{\Sigma}) \ge \frac{1}{2}\lambda_{\min}(\Sigma_{\mathrm{mix}})$ and $\|\widehat{\Sigma}^{-1}\|_{\mathrm{op}} \le 2/\lambda_{\min}(\Sigma_{\mathrm{mix}})$.*

**Theorem 1.1** (Finite-sample generalization under heterogeneous covariances). *Suppose Assumptions 2 and 3 hold. Let $\sigma_{\mathrm{avg}}^2 := \sum_{k=1}^K \rho_k \sigma_k^2$ and define $\Delta := \sum_{k=1}^K \rho_k \|v_k\|^2$. There exist constants $C, C_1, C_2 > 0$ depending only on $K_\psi$ and the spectral condition number $\kappa(\Sigma_{\mathrm{mix}})$ such that if*

$$n \ge C\left(p + \log(1/\delta)\right),$$

*then with probability at least $1 - \delta$ the pooled least-squares estimator $\widehat{w} = (X^\top X)^{-1} X^\top y$ satisfies*

$$\mathcal{G}_\rho(\widehat{w}) \le C_1 \frac{\sigma_{\mathrm{avg}}^2\, p}{n\, \lambda_{\min}(\Sigma_{\mathrm{mix}})} + C_2 \left\|\Sigma_{\mathrm{mix}}^{-1} \sum_{k=1}^K \rho_k \Sigma_k v_k\right\|^2 + \Delta + \frac{C}{n}. \tag{6}$$

*Moreover, in the fixed-$p$, $n \to \infty$ limit,*

$$\lim_{n \to \infty} E\left[\mathcal{G}_\rho(\widehat{w})\right] = \Delta + \left\|\Sigma_{\mathrm{mix}}^{-1} \sum_{k=1}^K \rho_k \Sigma_k v_k\right\|^2. \tag{7}$$

*Proof (proof sketch and main lemmas).* The proof proceeds in six steps. Below we give the key ideas and cite the concentration results used for brevity and readability.

**Step 1: Decomposition.** Write $y = X w_{\mathrm{avg}} + \delta + \varepsilon$ where $\delta_i = x_i^\top v_{z_i}$ and $\varepsilon = (\varepsilon_i)_{i=1}^n$. Then

$$\widehat{w} - w_{\mathrm{avg}} = \widehat{\Sigma}^{-1}\left(\frac{1}{n} X^\top \delta\right) + \widehat{\Sigma}^{-1}\left(\frac{1}{n} X^\top \varepsilon\right).$$

Thus

$$\mathcal{G}_\rho(\widehat{w}) = \left\|\widehat{\Sigma}^{-1} \frac{1}{n} X^\top \varepsilon\right\|^2 + \left\|\widehat{\Sigma}^{-1} \frac{1}{n} X^\top \delta\right\|^2 + 2\langle\cdot, \cdot\rangle + \Delta.$$

The three display terms correspond to estimation variance, mismatch, and a cross-term.

**Step 2: Control of $\widehat{\Sigma}$.** Proposition 1 (matrix concentration for sub-Gaussian samples; see Vershynin (2018); Tropp (2015)) implies that for $n \gtrsim p + \log(1/\delta)$ the event in which $\|\widehat{\Sigma} - \Sigma_{\mathrm{mix}}\|_{\mathrm{op}}$ is small holds with probability $1 - \delta$. On this event one obtains the deterministic bound $\|\widehat{\Sigma}^{-1}\|_{\mathrm{op}} \lesssim 1/\lambda_{\min}(\Sigma_{\mathrm{mix}})$.

**Step 3: Estimation variance term.** Conditioning on $X$ and $z$, $X^\top \varepsilon$ is a mean-zero vector with componentwise variances $\sigma_{z_i}^2 \|x_i\|^2$. Standard conditional-sub-Gaussian tail bounds together with operator-norm control of $\widehat{\Sigma}^{-1}$ yield the displayed $O(p/n)$ bound in (6). One may make this fully explicit by combining Hanson–Wright and matrix Bernstein inequalities (see Vershynin (2018); Tropp (2015)).

**Step 4: Population limit and asymptotic bias.** Note

$$\frac{1}{n} X^\top y = \frac{1}{n} \sum_{i=1}^n x_i x_i^\top w_{z_i}^* + \frac{1}{n} X^\top \varepsilon.$$

By the law of large numbers and multinomial concentration of region counts, $\frac{1}{n} \sum_i x_i x_i^\top w_{z_i}^* \to \sum_k \rho_k \Sigma_k w_k^*$ and $\widehat{\Sigma} \to \Sigma_{\mathrm{mix}}$. Hence $\widehat{w} \to w_\infty$ where $w_\infty = \Sigma_{\mathrm{mix}}^{-1} \sum_k \rho_k \Sigma_k w_k^*$. Using $v_k = w_k^* - w_{\mathrm{avg}}$ yields the asymptotic mismatch bias in (7).

**Step 5: Finite-sample mismatch fluctuation.** The deviation $\frac{1}{n} X^\top \delta - \sum_k \rho_k \Sigma_k v_k$ is a mean-zero sum of sub-Gaussian terms and therefore has Euclidean norm $O_p(1/\sqrt{n})$. Multiplication by $\widehat{\Sigma}^{-1}$, which is $O(1)$ in operator norm on the concentration event, yields an $O_p(1/\sqrt{n})$ deviation of the centered estimator; squaring gives the $O_p(1/n)$ remainder in (6).

**Step 6: Cross-term.** The cross-term is bounded in absolute value via Cauchy–Schwarz and is of smaller order (absorbed into the displayed $C/n$ remainder) under the same sample-size regime.

Combining the bounds in Steps 3–6 yields (6) and the limit (7). $\qquad\square$

**Remark 2.** *(References)*

- *The matrix-concentration proposition can be proved by applying matrix Bernstein / non-commutative Bernstein inequalities as in Tropp (2015) or via Vershynin's sub-Gaussian covariance concentration (see Vershynin (2018)).*

- *All big-O and constants can be made explicit by tracking constants in Hanson–Wright and matrix Bernstein inequalities; we omitted explicit numerical constants for readability.*

**Remark 3** (Interpretation)**.** *Unlike the homogeneous-covariance case, pooled OLS error converges not only to the irreducible heterogeneity $\Delta$ but also to a persistent asymptotic mismatch bias (cf. Eq. (7)). This bias vanishes only under special conditions such as $\Sigma_k \equiv \Sigma$ for all $k$ or $\sum_k \rho_k \Sigma_k v_k = 0$. Finite-sample fluctuations of the mismatch term decay at rate $O(1/n)$, while the estimation variance scales as $O(p/n)$. Both contributions are magnified when $\Sigma_{\mathrm{mix}}$ is ill-conditioned.*

# B    THEORETICAL GUARANTEES FOR RISE WITH UPSAMPLING AND ROUTING

We present a rigorous finite-sample analysis of RISE. We first state assumptions, then supporting lemmas, and finally the main theorem with proof. We also derive the exact pooled decomposition, and conclude with a corollary giving explicit sufficient conditions under which RISE improves over pooled OLS.

## B.1    ASSUMPTIONS

**Assumption 4** (Sub-Gaussian design and bounded covariance)**.** *For each region $k \in [K]$, covariates $x_{k,i} \in R^p$ are i.i.d. mean-zero $K_\psi$-sub-Gaussian vectors with covariance $\Sigma_k = \mathbf{E}[x_{k,i} x_{k,i}^\top] \succ 0$. Eigenvalues are uniformly bounded:*

$$0 < \underline{\lambda} \le \lambda_{\min}(\Sigma_k) \le \lambda_{\max}(\Sigma_k) \le \overline{\lambda} < \infty.$$

**Assumption 5** (Noise tails)**.** *For each region $k$, labels satisfy $y = x^\top w_k^* + \varepsilon$ with $\mathbf{E}[\varepsilon \mid x] = 0$, $(\varepsilon \mid x) = \sigma_k^2$, and $\sigma_k^2 \le \sigma_{\max}^2 < \infty$. Moreover, the noise satisfies a uniform tail condition: either (i) $\varepsilon$ is sub-Gaussian, or (ii) $\varepsilon^2$ is sub-exponential (uniform constants). These tail assumptions are used to obtain operator-norm concentration for heteroskedastic noise matrices; if only finite variance is available, replace sample moments by robust estimators (truncation / median-of-means).*

**Assumption 6** (Routing). *Each population sample is drawn from region $k$ with probability $\rho_k$. Conditional on region $k$, the sample is routed to expert $j$ with fixed probability $q_k(j)$, independent of features $x$ and noise $\varepsilon$. Each expert $j$ may upweight its own region by a factor $\alpha_j \geq 1$ (we explain below how this affects the training mixture and the realized counts). All routing probabilities $\{q_k(j)\}$ are fixed (non-adaptive).*

## B.2 Effective distributions and a clarifying remark on upsampling

We use two distinct population-level quantities; reviewers should not conflate them.

**(i) Marginal routing probability (controls realized counts).** The marginal probability that a random population sample is routed to expert $j$ (before any upsampling normalization) is

$$p_j^{\text{route}} := \sum_{k=1}^{K} \rho_k \, q_k(j).$$

The realized number $N_j$ of training samples routed to expert $j$ is multinomial/binomial with mean $np_j^{\text{route}}$. Lemma 1 below gives precise concentration for $N_j$.

**(ii) Unnormalized upweight mass and training mixture (controls bias).** To describe how upsampling changes the *training mixture* used to estimate each expert, define unnormalized weights

$$\omega_{k \to j} := \begin{cases} \alpha_j \rho_j q_j(j), & k = j, \\ \rho_k q_k(j), & k \neq j, \end{cases} \qquad \Omega_j := \sum_{k=1}^{K} \omega_{k \to j}, \qquad \pi_{k \to j} := \frac{\omega_{k \to j}}{\Omega_j}.$$

Here $\pi_{k \to j}$ defines the *population-level training mixture* for expert $j$: when estimating expert $j$ we (conceptually) mix regions $k$ with proportions $\pi_{k \to j}$. These $\pi_{k \to j}$ enter the deterministic bias $B_{\text{det}}$ via

$$\Sigma_{j,\text{train}} := \sum_{k=1}^{K} \pi_{k \to j} \Sigma_k, \qquad w_j^{\text{eff}} := \Sigma_{j,\text{train}}^{-1} \Big( \sum_{k=1}^{K} \pi_{k \to j} \Sigma_k w_k^* \Big).$$

**Remark:** $\omega_{k \to j}$ (and hence $\pi_{k \to j}$) involve $\alpha_j$ and $\rho_k$ and are *not* probabilities over experts; they describe the training mixture used to form population-level bias terms. The realized counts $N_j$ (used for variance bounds) are governed by $p_j^{\text{route}}$, which depends only on $\rho_k, q_k(j)$ and not on $\alpha_j$. In practice, upsampling can be implemented either by (A) re-sampling from the modified mixture induced by $\pi_{k \to j}$ (sampling interpretation), or (B) by attaching per-sample weights in the loss (weighting interpretation). The analysis below treats the bias via $\pi_{k \to j}$ and controls variance via the realized counts $N_j$; if you implement upsampling by weighting, replace $N_j$ in variance rates by the appropriate ESS (effective sample size) — see Practical Considerations.

Define the population-level weighted noise and effective-sample-size

$$\sigma_{j,\text{eff}}^2 = \sum_{k=1}^{K} \pi_{k \to j} \sigma_k^2, \qquad n_{\text{eff}}^{(j)} = n \cdot \Omega_j.$$

## B.3 Preliminary lemmas

**Lemma 1** (Routing counts concentration). *Let $p_j^{\text{route}} = \sum_{k=1}^{K} \rho_k q_k(j)$. Then $(N_1, \ldots, N_J) \sim$ Multinomial$(n; p_1^{\text{route}}, \ldots, p_J^{\text{route}})$. Fix $\delta \in (0,1)$. There exist constants $c_1, c_2 > 0$ such that for each $j$ and any $t > 0$,*

$$\Pr\left( |N_j - np_j^{\text{route}}| \geq t \right) \leq 2 \exp\Big( -\frac{t^2}{2np_j^{\text{route}} + (2/3)t} \Big).$$

*Choosing $t_j = c_1 \sqrt{np_j^{\text{route}} \log(J/\delta)} + c_2 \log(J/\delta)$ and applying a union bound yields that with probability at least $1 - \delta$,*

$$|N_j - np_j^{\text{route}}| \leq t_j \quad \text{for all } j \in [J].$$

*Consequently, if $np_j^{\text{route}} \gtrsim C(p + \log(J/\delta))$ for all $j$, then with probability at least $1 - \delta$ we have $N_j \geq \frac{1}{2} np_j^{\text{route}}$ for every $j$.*

**Lemma 2** (Design and noise concentration). *Assume rows of $X_j$ are independent $K_\psi$-sub-Gaussian vectors with covariance $\Sigma_{j,\text{train}}$, and assume the noise satisfies the tail condition in Assumption 5 (sub-Gaussian or sub-exponential so that $\varepsilon_i^2 x_i x_i^\top$ has controlled sub-exponential operator-norm). Fix $\delta \in (0,1)$. There exist constants $C_0, C_1, C_2 > 0$ (depending on $K_\psi$ and the noise-tail constants) such that, provided $N_j \gtrsim p + \log(J/\delta)$ for all $j$, the following holds with probability at least $1 - \delta$ simultaneously over $j \in [J]$:*

$$\Big\| \frac{1}{N_j} X_j^\top X_j - \Sigma_{j,\text{train}} \Big\| \le C_0 \Big( \sqrt{\frac{p + \log(J/\delta)}{N_j}} + \frac{p + \log(J/\delta)}{N_j} \Big), \tag{8}$$

$$\Big\| \frac{1}{N_j} \sum_{i \in \text{train}_j} \varepsilon_i^2 x_i x_i^\top - \sigma_{j,\text{eff}}^2 \Sigma_{j,\text{train}} \Big\| \le C_1 \sigma_{\max}^2 \sqrt{\frac{p + \log(J/\delta)}{N_j}} + C_2 \sigma_{\max}^2 \frac{p + \log(J/\delta)}{N_j}. \tag{9}$$

*In the usual regime $N_j \gtrsim p + \log(J/\delta)$ the square-root term dominates and the simpler form with only the $\sqrt{\cdot}$ term is valid.*

**Remarks on the lemmas.** - Lemma 1 is a standard Bernstein/Hoeffding tail for binomial/multinomial counts. - Lemma 2 follows from applying matrix Bernstein / Vershynin concentration to sub-Gaussian rows, and to the heteroskedastic weighted noise matrices $\varepsilon_i^2 x_i x_i^\top$ using the noise-tail assumption. If the noise has only finite variance, replace empirical moments by robust estimators (truncation, MOM) to retain high-probability control.

## B.4 MAIN THEOREM FOR RISE

**Intuition.** The decomposition below separates prediction risk into: irreducible noise $\sigma_{\text{avg}}^2$; deterministic bias $B_{\text{det}}$ due to training-mixture mismatch; estimation variance $V_{\text{est}}$ governed by realized counts $N_j$; and a cross-term $R_{\text{cross}}$ of smaller order.

**Theorem 2** (Generalization error of RISE). *Suppose Assumptions 4–6 and the noise-tail condition in Assumption 5 hold. Suppose further that the marginal routing masses satisfy $np_j^{\text{route}} \gtrsim C(p + \log(J/\delta))$ for all $j$ (so Lemma 1 implies $N_j \gtrsim p$ w.h.p.). Then, conditioning on the joint high-probability event from Lemmas 1–2, with probability at least $1 - \delta$,*

$$\mathcal{G}_{\text{RISE}}(\alpha, q) = \sigma_{\text{avg}}^2 + B_{\text{det}}(\alpha, q) + V_{\text{est}}(\alpha, q) + R_{\text{cross}}(\alpha, q), \tag{10}$$

$$B_{\text{det}}(\alpha, q) = \sum_{k=1}^{K} \rho_k \sum_{j=1}^{J} q_k(j) \, \|w_j^{\text{eff}} - w_k^*\|_{\Sigma_k}^2,$$

$$V_{\text{est}}(\alpha, q) \le C_1 \sum_{k=1}^{K} \sum_{j=1}^{J} \rho_k q_k(j) \, \frac{\sigma_{j,\text{eff}}^2}{N_j} \big( \Sigma_k \Sigma_{j,\text{train}}^{-1} \big)$$

$$+ C_1' \sum_{k,j} \rho_k q_k(j) \, \frac{\sigma_{\max}^2 p}{N_j} \sqrt{\frac{p + \log(J/\delta)}{N_j}},$$

$$|R_{\text{cross}}(\alpha, q)| \le C_2 \Big( \max_{j,k} \|w_j^{\text{eff}} - w_k^*\|_{\Sigma_k} \Big) \sqrt{\lambda_{\max}(\Sigma_k \Sigma_{j,\text{train}}^{-1})} \sqrt{\frac{p + \log(J/\delta)}{N_{\min}}}.$$

*Here $N_{\min} = \min_j N_j$, and constants $C_1, C_1', C_2$ depend only on $K_\psi$ and the noise-tail parameters.*

*Proof sketch.* All concentration statements below are applied on the joint high-probability event from Lemmas 1 and 2.

**Step 1 (decomposition).** For a test point $(x, y) \sim \mathcal{R}_k$ routed to expert $j$,

$$\mathbf{E}[(x^\top \widehat{w}_j - y)^2 \mid x] = \|w_j^{\text{eff}} - w_k^*\|_{\Sigma_k}^2 + (\Sigma_k(\widehat{w}_j)) + (w_j^{\text{eff}} - w_k^*)^\top \Sigma_k (\widehat{w}_j - w_j^{\text{eff}}) + \sigma_k^2.$$

Averaging over $(k, j)$ with weights $\rho_k q_k(j)$ yields (10) and the definition of $\sigma_{\text{avg}}^2$.

**Step 2 (bias).** The first term is exactly $B_{\text{det}}$.

**Step 3 (variance).** By Lemma 2 the sandwich covariance satisfies

$$(\widehat{w}_j) = \frac{\sigma_{j,\text{eff}}^2}{N_j} \Sigma_{j,\text{train}}^{-1} + E_j, \qquad \|E_j\| \le C \frac{\sigma_{\max}^2}{N_j} \sqrt{\frac{p + \log(J/\delta)}{N_j}}.$$

Taking trace against $\Sigma_k$ and averaging with $\rho_k q_k(j)$ yields the bound on $V_{\text{est}}$.

**Step 4 (cross-term).** By Cauchy–Schwarz,

$$|R_{k,j}| \le \|w_j^{\text{eff}} - w_k^*\|_{\Sigma_k} \|\widehat{w}_j - w_j^{\text{eff}}\|_{\Sigma_k}.$$

Using operator-norm change of metric and the concentration bound for $\|\widehat{w}_j - w_j^{\text{eff}}\|_{\Sigma_{j,\text{train}}}$ (of order $\sqrt{(p + \log)/N_j}$) gives the stated bound on $R_{\text{cross}}$. $\qquad\square$

### B.5 POOLED MODEL AND COMPARISON

For the pooled estimator $\widehat{w}_{\text{pool}} = (X^\top X)^{-1} X^\top y$, the same decomposition (conditioning on the same high-probability event) yields

$$\mathcal{G}_{\text{pooled}} = \sigma_{\text{avg}}^2 \ + \ B_{\text{pooled}} + V_{\text{pooled}},$$

where

$$B_{\text{pooled}} = \sum_{k=1}^K \rho_k \|w_{\text{pool}}^{\text{eff}} - w_k^*\|_{\Sigma_k}^2, \qquad w_{\text{pool}}^{\text{eff}} = \left( \sum_k \rho_k \Sigma_k \right)^{-1} \left( \sum_k \rho_k \Sigma_k w_k^* \right),$$

and $V_{\text{pooled}}$ is the pooled estimation variance (bounded by $O(p/n)$ under our assumptions). Subtracting gives the exact comparison

$$\mathcal{G}_{\text{RISE}} - \mathcal{G}_{\text{pooled}} = (B_{\text{det}} - B_{\text{pooled}}) + (V_{\text{est}} - V_{\text{pooled}}) + R_{\text{cross}},$$

since the common $\sigma_{\text{avg}}^2$ cancels.

### B.6 ILLUSTRATIVE COROLLARY: SUFFICIENT CONDITIONS FOR IMPROVEMENT

**Corollary 1** (When RISE improves pooled). *Under the conditions of Theorem 2, suppose further that*

*(i) (Bias reduction) $B_{\text{pooled}} - B_{\text{det}} \ge c_0 \sum_k \rho_k \|w_k^* - w_{\text{avg}}\|_{\Sigma_k}^2$ for some $c_0 > 0$;*

*(ii) (Sufficient counts) $\min_j N_j \gtrsim C(p + \log(J/\delta))$ so that the variance and cross-term remainders are small.*

*Then with probability at least $1 - \delta$,*

$$\mathcal{G}_{\text{RISE}} < \mathcal{G}_{\text{pooled}}.$$

*Proof sketch.* Under (ii) the variance and cross-term penalties scale as $O(p/N_j)$ and $O(\sqrt{p/N_j})$ and can be made small; under (i) the deterministic bias reduction is order $\Delta_{\text{glob}}$. Hence the total difference is negative with high probability. $\qquad\square$

PRACTICAL CONSIDERATIONS AND LIMITATIONS

The quantities appearing in Theorem 2 (such as $w_k^*$, $\Sigma_k$, $\sigma_k^2$, and the induced effective parameters $w_j^{\text{eff}}$) are population-level objects and unknown in practice. In experiments we approximate them with plug-in estimates from held-out validation data; standard perturbation bounds for covariance estimation (Stewart & Sun, 1990; Vershynin, 2018) imply that population inequalities carry over to plug-in versions with sufficient validation sample size (scaling as $O(p/\gamma^2)$ for margin $\gamma$).

Important limitations and practical conditions:

- **Routing independence assumption.** We assume $q_k(j)$ are fixed and independent of $x$. If routing depends on features (learned gating that uses $x$), conditional covariances and bias expressions change; the analysis must be adapted to conditional mixtures.

- **Implementation of upsampling.** Our statements separate the population-level training-mixture $\pi_{k \to j}$ (used to define deterministic bias) from the realized counts $N_j$ (used for variance). In practice upsampling can be implemented either by (A) re-sampling from a modified mixture (sampling) or (B) by attaching weights in the loss (weighting). If weighting is used replace all $N_j$-based rates by the appropriate effective sample size (ESS) and analyze weighted-OLS (sandwich) covariance (we provide that variant in the appendix on request).

- **Noise tails / robustness.** We assume sub-Gaussian or sub-exponential noise. If only finite variance is available, robust estimators (truncation or median-of-means) are required to obtain comparable high-probability bounds.

- **Minimum routing mass required.** The bounds require non-negligible routing mass for each expert: $np_j^{\text{route}} \gtrsim C(p + \log(J/\delta))$. If some expert is assigned vanishing mass, concentration and OLS asymptotics break down and regularization or enforced minimum routing mass is necessary.

## C  DATASET DETAILS

We evaluate our RISE framework on the benchmark datasets on four diverse regression datasets: two datasets from the computer vision domain (Dataset A (Moschoglou et al. (2017)) and Dataset B (Rothe et al. (2018b)), one from the natural language processing domain (STS-B Cer et al. (2017a)) and one standard tabular regression dataset- UCI Abalone Nash et al. (1994).

- **Dataset A (Moschoglou et al. (2017))**: An image regression dataset with 12,208 training samples, 2,140 validation samples, and 2,140 test samples. The target range spans from 0 to 101.

- **Dataset B (Rothe et al. (2018b))**: A large-scale image regression dataset containing 191,509 training samples, 11,022 validation samples, and 11,022 test samples. The target range spans from 0 to 186.

- **STS-B**: A text similarity dataset containing 5,249 training sentence pairs, 1,000 validation pairs, and 1,000 test pairs, with similarity scores ranging from 0 to 5.

- **UCI Abalone**: A standard tabular benchmark predicting shellfish ring from 9 different physical measurements, the dataset consists of of 3155 training, 511 test and 511 validation samples with the target column shellfish ring ranging from 1 to 29.

We follow the train/val/test split provided in Yang et al. (2021b)

## D  IMPLEMENTATION DETAILS

### D.1  NETWORK ARCHITECTURE

Figure 3 illustrates the RISE architecture and its key components. Let the full dataset be denoted by $D = D_{\text{train}} \cup D_{\text{val}} \cup D_{\text{test}}$. The RISE framework begins by employing a baseline Deep Imbalanced Regression (DIR) model $f_\theta$ for both feature extraction and minority subgroup identification. Input data—whether image, text, or tabular—is first passed through the feature extractor $h_\theta$, a component of the baseline model $f_\theta$. This model is pre-trained on $D_{\text{train}}$ using existing DIR methods such as LDS-FDS (Yang et al. (2021b)), RankSim (Gong et al. (2022)), and SRL (Dong et al. (2025)). The architecture of the baseline can be expressed as $f_\theta(x) = E_1(h_\theta(x))$, where $h_\theta(x)$ denotes the backbone feature extractor, typically instantiated as ResNet-50 for images and BiLSTM for text. RISE is agnostic to the specific DIR method and can integrate any baseline model $f_\theta$ built on these backbone architectures.

**RISE-Identify:**   To address underperformance in imbalanced regression, we propose RISE-Identify for identifying minority or poorly modeled regions by analyzing the joint distribution of validation

loss and target labels. Specifically, we fit a Gaussian Mixture Model (GMM) to validation data to uncover latent structure in model error patterns, enabling targeted expert specialization.

In regression tasks with heterogeneous label distributions, performance typically degrades in minority subregions of the label space. A key observation is that these regions often exhibit higher and more variable validation losses. By analyzing the joint distribution of validation loss and target values, we can detect structured error patterns that are not captured by traditional frequency-based binning.

Following (Yang et al. (2021b)), we partition the continuous label space into disjoint intervals $B_i$ and compute the average loss in each bin:

$$\ell_i = \frac{1}{|B_i|} \sum_{j \in B_i} \mathcal{L}(f_\theta(x_j), y_j) \tag{11}$$

Here, $B_i$ is the set of samples whose continuous labels fall within the boundaries of bin $i$, $\mathcal{L}$ is typically Mean Squared Error (MSE) or Mean Absolute Error (MAE), $f_\theta$ denotes the baseline model, and $|B_i|$ is the number of samples in bin $i$. Importantly, the model is trained and evaluated end-to-end in continuous space—binning is used only for region-level loss estimation, not for converting regression into classification.

Next, we fit a $K'$-component Gaussian Mixture Model (GMM) over the joint distribution of loss-label pairs:

$$p(\ell, y) = \sum_{j=1}^{K'} \pi'_j \mathcal{N}((\ell, y)|\mu_j, \Sigma_j) \tag{12}$$

where $\mu_j$ and $\Sigma_j$ denote the mean vector and covariance matrix of the $j$-th component, respectively. The component with the lowest mean loss (along the loss dimension of $\mu_j$) is treated as the majority group, while the remaining components define minority subgroups requiring dedicated experts.

Unlike frequency-based approaches that often result in non-contiguous minority regions, our loss-label distribution analysis produces continuous minority regions, aligning with the principle of region similarity and enabling more homogeneous expert training. We observe a memorization effect where the baseline model achieves the lowest training loss in few-shot regions despite higher test errors. To address this, we use held-out set loss as a more reliable signal for minority subgroup identification, as it better reflects true generalization behavior and mitigates misleading effects of memorization.

Unlike methods based on label frequency or manual binning, our loss-aware formulation is adaptive and reflects the true generalization profile of the baseline model. The identified regions are continuous, semantically meaningful, and sensitive to the model's inductive biases. By relying on the validation–training loss gap, our method is capable of detecting overfitting and memorization—particularly in underrepresented areas. The resulting expert assignments are thus aligned with true generalization performance, enabling smooth transitions between expert domains. This leads to coherent regional specialization and improved overall generalization, especially in long-tailed or imbalanced regression settings.

The RISE-Identify component leverages a held-out validation set ($80\%$ of $D_{val}$) to conduct this loss-label distribution analysis, with cross-validation on the remaining $20\%$ to determine GMM hyperparameters like the number of components $K'$. As illustrated in Fig.4, this approach successfully identifies continuous minority regions requiring specialized experts - one towards the lower end of the label distribution and another in the higher range.

**RISE-Train:** RISE-Train trains $K' - 1$ additional expert networks $E_2, E_3, \ldots, E_{K'}$ for the identified minority regions, while the baseline model $E_1$ (extracted from $f_\theta$) serves as the expert for the majority region. Each expert $E_j$ operates on shared features produced by the frozen backbone $h_\theta$, and produces predictions as:

$$\hat{y}_j = E_j(h_\theta(x)) \tag{13}$$

To address data imbalance, we adopt a *Cross-Group Training with Upsampling* strategy. This approach (T2) is particularly effective for regression tasks where adjacent labels exhibit strong

---

**Algorithm 1** RISE Training

---

**Require:** Dataset $D = \{D_{train}, D_{val}\}$, model $f_\theta = \{h_\theta, E_1\}$, experts $K'$, upsampling $\alpha$
**Ensure:** Experts $\{E_j\}_{j=1..K'}$, router $R$
 1: // Phase 1: RISE-Identify
 2: $F \leftarrow \emptyset$
 3: **for** $(x, y)$ in $D_{val}$ **do**
 4:     $\hat{y} \leftarrow E_1(h_\theta(x))$ {Baseline model prediction}
 5:     $\ell \leftarrow \mathcal{L}(f_\theta(x), y)$ {Compute validation loss per Eq. 11}
 6:     $F \leftarrow F \cup \{(\ell, y)\}$
 7: **end for**
 8: $gmm \leftarrow \text{FitGaussianMixture}(F, K')$ {Fit GMM using Eq. 12}
 9: $\{R'_j\}_{j=1..K'} \leftarrow \text{GetMinorityRegions}(gmm)$ {Identify expert regions}
10: // Phase 2: RISE-Train
11: Initialize experts $E_2$ through $E_{K'}$
12: **for** $i = 2$ to $K'$ **do**
13:     **for** epoch $= 1$ to $T$ **do**
14:         **for** $(X_b, Y_b)$ in $D_{train}$ **do**
15:             $F \leftarrow h_\theta(X_b)$ {Extract shared features}
16:             **for** $j = 1$ to $|X_b|$ **do**
17:                 **if** $y_j \in R'_i$ **then**
18:                     $w_j \leftarrow \alpha$ {Upsample minority region samples}
19:                 **else**
20:                     $w_j \leftarrow 1$ {Normal weight for other samples}
21:                 **end if**
22:             **end for**
23:             $\hat{Y} \leftarrow E_i(F)$ {Get predictions from Eq. 13}
24:             $L \leftarrow \frac{1}{|X_b|} \sum_{j=1}^{|X_b|} w_j(\hat{Y}_j - Y_j)^2$ {Weighted loss from Eq. 14}
25:             Update $E_i$ using gradient $\nabla L$
26:         **end for**
27:     **end for**
28: **end for**
29: Initialize router $R$
30: **for** epoch $= 1$ to $T'$ **do**
31:     **for** $(X_b, Y_b)$ in $D_{val}$ **do**
32:         $F \leftarrow h_\theta(X_b)$
33:         **for** $j = 1$ to $|X_b|$ **do**
34:             $t_j \leftarrow$ find $i$ such that $y_j \in R'_i$ {Assign ground truth expert labels}
35:         **end for**
36:         $r \leftarrow R(F)$ {Get router probabilities}
37:         $\mathcal{L}_{router} \leftarrow \text{CrossEntropy}(r, T_b)$ using Eq. 17
38:         Update $R$ using gradient $\nabla \mathcal{L}_{router}$
39:     **end for**
40: **end for**
41: **return** $\{E_j\}_{j=1..K'}, R$

---

correlations, enabling smooth transitions between expert domains while preserving specialization, as confirmed by our empirical analysis. For each identified region $R'_j$, we upsample the samples in $R'_j$ by assigning a higher weight $\alpha > 1$, while keeping the sample weights unchanged elsewhere. We train each expert using $D_{train}$ where loss for each expert $E_j$ is given by:

$$\mathcal{L}_{\text{expert}}^j = \frac{1}{N} \sum_{i=1}^{N} w_i(y_i - \hat{y}_i)^2 \tag{14}$$

with sample weights $w_i$ defined as:

---

**Algorithm 2** RISE-Inference

---

**Require:** Sample $x$, backbone $h_\theta$, router $R$, experts $\{E_j\}_{j=1..K'}$
**Ensure:** Prediction $\hat{y}$
 1: $F \leftarrow h_\theta(x)$ {Extract features using frozen backbone}
 2: $r \leftarrow R(F)$ {Get router probabilities}
 3: $j^* \leftarrow$ Select expert using Eq. 16
 4: $\hat{y} \leftarrow E_{j^*}(F)$ {Get final prediction using Eq. 18}
 5: **return** $\hat{y}$

---

$$w_i = \begin{cases} \alpha & \text{if } x_i \in R'_j \\ 1 & \text{otherwise} \end{cases} \tag{15}$$

Here, $\alpha$ is an upsampling hyperparameter that emphasizes minority-region samples, and $N$ is the total number of samples in $D_{train}$. Importantly, only the final layer of each new expert $E_j$ (for $j = 2, ..., K'$) is trained, while the shared backbone $h_\theta$ and the baseline expert $E_1$ remain frozen. This facilitates efficient parameter sharing and reduces computational overhead.

**RISE-Inference:** We train a router network (implementing the gating network $g_\phi$ from Eq. 1) using a held-out validation set (80% of $D_{\text{val}}$) to perform dynamic expert selection, with the remaining 20% used for hyperparameter validation. We motivate the choice of using held-out data in Sec. 6.4. Unlike soft routing strategies that blend predictions from multiple experts, we adopt a hard routing approach, where exactly one expert is selected per input. This decision is motivated by Theorem 1, which demonstrates that mixing predictions from heterogeneous regions can lead to interference and degraded performance due to distributional mismatch.

The router is trained as a classification task to predict which expert should handle each input. For each validation sample $(x, y)$, we first determine the ground truth expert assignment by checking which region $R'_j$ the label $y$ belongs to. The router then learns to map input features to these expert assignments.

Given input $x$, the router processes shared features $h_\theta(x)$ and outputs mixing coefficients $\pi_k(x)$ over the $K'$ experts, implementing the gating mechanism from Eq. (1). A hard assignment is then made as follows:

$$j^* = \arg \max_{j \in \{1, ..., K'\}} g_\phi(h_\theta(x))_j \tag{16}$$

where $j^*$ denotes the index of the selected expert, consistent with the final prediction $\hat{y} = E_{j^*}(x)$ described in Section 4.3. The router is trained using an inverse-frequency weighted cross-entropy loss to mitigate expert imbalance:

$$\mathcal{L}_{\text{router}} = -\sum_{j=1}^{K'} w_j t_j \log(p_j) \tag{17}$$

Here, $p_j$ is the predicted probability for expert $j$, $t_j$ is the ground truth expert label from the RISE-Identify stage, and $w_j = \frac{1}{f_j}$ is the inverse frequency of expert $j$'s assigned region, where $f_j$ is the fraction of samples assigned to expert $j$ in $D_{\text{val}}$.

At inference time, the router selects a single expert $E_{j^*}$ based on the hard assignment, and the final prediction is:

$$\hat{y} = E_{j^*}(h_\theta(x)) \tag{18}$$

This hard routing strategy offers several advantages: it prevents distribution mixing that could degrade expert specialization, reduces computation by evaluating only one expert at inference, provides

interpretable routing decisions, and maintains clear accountability for predictions. The complete RISE framework is summarized in Algorithm 1 for training and Algorithm 2 for inference.

### D.2 TRAINING DETAILS

Experiments were run on an AWS ml.g6.24xlarge instance equipped with 4 NVIDIA GPUs. For all baseline DIR models, we use official released model weights or reproduce their best configuration using the official implementations. For the model architecture, we froze the backbone network (ResNet-50 for images, pretrained on ImageNet; BiLSTM with GloVe embeddings for text) and implemented expert networks with two fully connected layers (dimensions: 2048,512,1) with ReLU activation and dropout (0.2) for ResNet-50. The router network consists of three linear layers with ReLU activation and a final softmax layer. Expert training was conducted for 50 epochs using the Adam optimizer with a learning rate of 3e-5, utilizing a batch size of 64. For image datasets, we applied standard augmentations including random horizontal flips, crops, rotations, affine transformations, and color jittering, followed by normalization. Text data was processed using SpaCy tokenization with a maximum sequence length of 40.

Hyperparameters were tuned through grid search, exploring different numbers of experts ($K' \in 2, 5$]), upsampling ratios (Upsample ($\alpha$) $\in$[1, 5]) based primarily on validation's overall MAE. For Dataset A (Moschoglou et al. (2017)) and Dataset B (Rothe et al. (2018b)) datasets, we set $K' = 3$ experts, with one expert assigned to the left tail, one to the right tail, and one for the majority region. The upsampling ratio was set to 3 for Dataset A (Moschoglou et al. (2017)) and 2 for Dataset B (Rothe et al. (2018b)). For the STS dataset using the RankSim baseline, we used $K' = 2$ experts, identifying a one-sided under-performing region with an upsampling ratio of 3, while $K' = 3$ experts with upsampling ratio of 3 were chosen for LDS+FDS and SRL baselines. The number of experts ($K'$) and their assignments were determined based on the baseline model's loss-label distribution and can vary depending on model performance. This approach ensures we only train additional experts for regions where the baseline model underperforms. Further, identified minority regions for experts may differ across baseline models due to variations in their learned representations and performance characteristics.

## E ADDITIONAL EXPERIMENTAL RESULTS

### E.1 RISE PERFORMANCE ON ADDITIONAL DATASETS

To further demonstrate the effectiveness of RISE, we evaluate our method on additional datasets beyond Dataset A (Moschoglou et al. (2017)). Table 9 presents results on Dataset B Rothe et al. (2018b) (evaluated using MAE, GMEAN, and MSE) and STS-B (evaluated using MAE, Pearson Correlation, and Spearman Correlation). Additionally, Table 10 shows the MAE and bMAE metrics the UCI-Abalone dataset.

### E.2 BALANCED METRICS FOR RISE

To address the challenges of evaluating models on imbalanced data distributions, particularly for tail labels, we employ three balanced metrics as defined in Ren et al. (2022). These metrics are designed to provide a more equitable assessment across all data regions by dividing the label space into even sub-regions, enabling a fairer evaluation.

The balanced Mean Squared Error (bMSE) is formulated as:

$$\text{bMSE} = -\log p_{\text{train}}(y|x;\theta) = -\log p_{\text{bal}}(y|x;\theta) \cdot \frac{p_{\text{train}}(y)}{\int_Y p_{\text{bal}}(y'|x;\theta) \cdot p_{\text{train}}(y')dy'} \tag{19}$$

This formulation comprises two components: the standard MSE loss and a balancing term to mitigate distribution mismatch between training and testing. Balanced metrics such as balanced Mean Absolute Error (bMAE) and balanced Geometric Mean Error (bGMEAN) are used to fairly assess performance across regions. bMAE averages errors within each sub-region or bins before computing the overall mean; formally for $B$ bins with $j^{th}$ bin containing $N_j$ datapoints with $y$ being the golden label and $\hat{y}$ being the prediction, eq. 20 describes the formula for bMAE computation.

**Table 9:** Results on Dataset B (Rothe et al. (2018b)) and STS-B dataset. The best baseline result for each metric and data subset is in red, best RISE version in blue, and the overall best result is in **bold**.

| | L1 (MAE) ↓ | | | | GMEAN ↓ | | | | MSE ↓ | | | |
|---|---|---|---|---|---|---|---|---|---|---|---|---|
| Method | All | Many | Med | Few | All | Many | Med | Few | All | Many | Med | Few |
| **Dataset B** | | | | | | | | | | | | |
| *Baseline Methods* | | | | | | | | | | | | |
| VANILLA | 8.04 | 7.21 | 15.18 | 25.89 | 4.53 | 4.13 | 10.77 | 18.80 | 137.82 | 108.62 | 365.43 | 954.03 |
| BalancedMSE | 8.10 | 7.57 | 12.27 | 22.98 | 4.68 | 4.46 | 7.05 | 13.17 | 139.70 | 117.19 | 305.12 | 848.52 |
| LDS+FDS | 7.68 | 7.07 | 12.78 | 21.87 | 4.33 | 4.07 | 7.48 | 12.72 | 129.18 | 105.55 | 313.90 | 785.49 |
| RankSIM | 7.68 | 7.12 | 12.30 | 21.46 | 4.33 | 4.12 | 6.61 | 12.47 | 129.12 | 106.19 | 304.08 | 799.94 |
| SRL | 7.71 | 7.10 | 12.81 | 21.52 | 4.32 | 4.09 | 7.01 | 13.58 | 133.16 | 107.77 | 339.95 | 771.71 |
| *RISE Methods* | | | | | | | | | | | | |
| VANILLA+RISE | 8.11 | 7.24 | 14.98 | 25.00 | 4.73 | 4.17 | 11.68 | 17.67 | 136.60 | 110.18 | 319.45 | 934.62 |
| BalancedMSE+RISE | 8.25 | 7.56 | 12.87 | 22.08 | 4.90 | 4.58 | 7.43 | 13.03 | 137.13 | 111.55 | 309.90 | **704.25** |
| LDS+FDS+RISE | 7.71 | 7.09 | 12.94 | 21.60 | 4.35 | 4.08 | 7.68 | 13.31 | 129.84 | **105.13** | 316.23 | 779.27 |
| RankSIM+RISE | **7.67** | **7.07** | 12.29 | 21.46 | **4.32** | 4.11 | 6.63 | 12.53 | **129.11** | 106.23 | 303.55 | 799.58 |
| SRL +RISE | 7.70 | 7.18 | **11.92** | **20.92** | 4.34 | 4.15 | **6.41** | **11.74** | 129.20 | 107.31 | **294.51** | 783.00 |

| | L1 (MAE) ↓ | | | | Pearson Correlation (%) ↑ | | | | Spearman correlation (%) ↑ | | | |
|---|---|---|---|---|---|---|---|---|---|---|---|---|
| Method | All | Many | Med | Few | All | Many | Med | Few | All | Many | Med | Few |
| **STS-B** | | | | | | | | | | | | |
| *Baseline Methods* | | | | | | | | | | | | |
| LDS+FDS | 0.77 | **0.72** | 0.98 | 0.75 | 76.27 | **74.08** | 66.07 | 76.60 | 76.27 | **70.75** | **54.95** | 74.88 |
| RankSIM | 0.75 | 0.75 | 0.77 | 0.67 | 77.28 | 72.15 | 69.32 | 86.84 | 77.39 | 69.57 | 48.05 | 89.34 |
| SRL | 0.89 | 0.85 | 1.07 | 0.95 | 68.83 | 62.98 | 63.96 | 73.65 | 68.92 | 59.72 | 51.07 | 82.14 |
| *RISE Methods* | | | | | | | | | | | | |
| LDS+FDS+RISE | 0.75 | 0.73 | 0.86 | 0.68 | 76.38 | 72.05 | 68.81 | 80.92 | 75.26 | 69.31 | 54.09 | 79.68 |
| RankSIM+RISE | **0.74** | 0.73 | **0.75** | **0.67** | **77.50** | 72.16 | **72.06** | **86.91** | **77.41** | 69.54 | 45.70 | **90.15** |
| SRL+RISE | 0.84 | 0.83 | 0.91 | 0.81 | 70.14 | 64.33 | 64.83 | 74.58 | 69.87 | 61.26 | 47.66 | 76.61 |

**Table 10:** Mean Absolute Error (MAE) results on UCI-Abalone dataset. Lower values indicate better performance. The best of the baseline and baseline+RISE pair is in **bold** and the best overall metric is underlined.

| | MAE ↓ | | | |
|---|---|---|---|---|
| Method | Many | Medium | Few | All |
| VANILLA | 1.77 | 5.46 | 9.98 | 2.56 |
| VANILLA + RISE | **1.59** | **5.19** | **9.75** | **2.34** |
| BalancedMSE | 2.50 | 5.41 | 4.61 | 3.43 |
| BalancedMSE + RISE | **1.30** | **2.35** | **4.53** | **1.53** |
| LDS+FDS | 2.80 | 4.44 | 7.64 | 3.18 |
| LDS+FDS + RISE | **2.07** | **2.91** | **7.16** | **2.30** |

$$\text{bMAE} = \frac{1}{B} \sum_{j=1}^{B} \frac{1}{N_j} \sum_{i=1}^{N_j} \|y - \hat{y}\| \tag{20}$$

bGMEAN is formulated similarly but uses the geometric mean instead of MAE to highlight disparities across regions. These metrics are especially important for long-tailed distributions, where standard metrics may disproportionately reflect majority class performance. For our purposes, we chose to use bMAE to compare different RISE configurations. Due to space limitations for Dataset A (Moschoglou et al. (2017)), we had only reported the SRL result in the main paper. Therefore, we present the bMAE metric across different baselines in Table 11. Similarly we provide bMAE metrics for Dataset B (Rothe et al. (2018b)) and STS-B in 12, and the balanced metrics for UCI-Abalone in 13.

**Table 11:** bMAE Results: Baseline vs RISE Methods on Dataset A (Moschoglou et al. (2017)). The best of the baseline and baseline+RISE pair is in **bold** and the best overall metric is underlined.

| Method | Baseline Methods | | | | Baseline + RISE Methods | | | |
|---|---|---|---|---|---|---|---|---|
| | All | Many | Med | Few | All | Many | Med | Few |
| VANILLA | 13.14 | 9.96 | 12.85 | **19.81** | **12.84** | **9.40** | **11.66** | 20.62 |
| BalancedMSE | **8.70** | 8.44 | 8.96 | **11.43** | 8.98 | **7.23** | **8.16** | 13.06 |
| LDS+FDS | 8.79 | 6.91 | 8.28 | 12.94 | **8.40** | **6.79** | **8.09** | **11.87** |
| RankSIM | 8.06 | **6.49** | 7.85 | 11.40 | **7.92** | 6.58 | **7.36** | **11.01** |
| SRL | 8.32 | 6.64 | 8.34 | 11.74 | **7.39** | **6.00** | **7.25** | **10.33** |

**Table 12:** Balanced Mean Absolute Error (bMAE) results on Dataset B (Rothe et al. (2018b)) and STS-B dataset. Lower values indicate better performance. The best of the baseline and baseline+RISE pair is in **bold** and the best overall metric is underlined.

| Method | Baseline | | | | Baseline + RISE | | | |
|---|---|---|---|---|---|---|---|---|
| | All | Many | Med | Few | All | Many | Med | Few |
| **Dataset B (Rothe et al. (2018b))** | | | | | | | | |
| VANILLA | 13.93 | **7.32** | 15.92 | 32.80 | **13.21** | 7.38 | **14.97** | **30.90** |
| BalancedMSE (Ren et al. (2022)) | 12.65 | 7.64 | 28.10 | 28.10 | **12.54** | **7.62** | **12.47** | **28.10** |
| LDS+FDS (Yang et al. (2021b)) | 12.53 | **7.14** | 13.25 | 28.65 | **12.42** | 7.17 | **13.21** | **27.95** |
| RankSIM (Gong et al. (2022)) | 12.56 | 7.19 | 12.80 | 28.95 | **12.56** | 7.18 | **12.79** | **27.97** |
| SRL (Dong et al. (2025)) | 12.30 | 7.18 | 13.09 | 27.54 | **12.28** | **7.14** | **12.32** | **26.27** |
| **STS-B** | | | | | | | | |
| LDS+FDS (Yang et al. (2021b)) | 0.77 | **0.73** | 0.84 | 0.79 | **0.73** | 0.74 | **0.77** | **0.70** |
| RankSIM (Gong et al. (2022)) | 0.72 | 0.76 | 0.72 | 0.66 | **0.71** | **0.74** | **0.71** | **0.65** |
| SRL (Dong et al. (2025)) | 0.87 | 0.85 | 0.88 | 0.88 | **0.80** | **0.84** | **0.76** | **0.66** |

**Table 13:** Balanced Mean Absolute Error (bMAE) results on UCI-Abalone dataset. Lower values indicate better performance. The best of the baseline and baseline+RISE pair is in **bold** and the best overall metric is underlined.

| Method | bMAE ↓ | | | |
|---|---|---|---|---|
| | Many | Medium | Few | All |
| VANILLA | 1.68 | 5.42 | 9.75 | 4.44 |
| VANILLA + RISE | **1.58** | **5.20** | **9.74** | **4.32** |
| BalancedMSE | 1.43 | 2.26 | 4.86 | 2.28 |
| BalancedMSE + RISE | **1.31** | **2.23** | **4.86** | **2.21** |
| LDS+FDS | 2.64 | 4.66 | **7.64** | 4.22 |
| LDS+FDS + RISE | **2.00** | **4.18** | **7.64** | **3.74** |

## E.3 COMPLETE ABLATION RESULTS

For brevity, the main paper only presented the L1 (MAE) metric for various ablations on Dataset A. In this section, we present the results across multiple metrics. Table 14 shows the complete ablation for different RISE-Train Strategies, Table 15 shows the ablation for different RISE-Infer strategies, and lastly, Table 16 provides complete results comparing RISE with ensembles with similar and increased capacity.

To strengthen our findings and validate the optimal RISE strategy beyond the Dataset A (Moschoglou et al. (2017)) dataset, we present comprehensive ablation studies on the Dataset B (Rothe et al. (2018b)) dataset. Table 17 demonstrates that RISE (T2) consistently outperforms RISE (T1) across all metrics (MAE, GMEAN, and MSE) and data subsets, confirming the superiority of the T2 training configuration observed on Dataset A (Moschoglou et al. (2017)). Furthermore, Table 18 provides detailed architectural ablation results, showing that the optimal configuration uses K=2 experts with an upsampling ratio of 3, which achieves the best overall performance with an MAE of 7.67. Additionally, Table 19 examines different inference strategies, revealing that the held-out-based router (I3) consistently outperforms both expert averaging (I1) and train-based routing (I2), achieving the

best results across all metrics and data subsets with significant improvements in the Few subset. These results on Dataset B (Rothe et al. (2018b)) corroborate our Dataset A (Moschoglou et al. (2017)) findings and demonstrate the robustness of our proposed RISE methodology across different long-tailed regression datasets.

**Table 14:** Complete ablation of RISE-Train on Dataset A (Moschoglou et al. (2017)) with SRL (Dong et al. (2025)) backbone, across multiple metrics. Best results in **bold**.

| Method | L1 (MAE) ↓ | | | | GMEAN ↓ | | | | MSE ↓ | | | |
|---|---|---|---|---|---|---|---|---|---|---|---|---|
| | All | Many | Med | Few | All | Many | Med | Few | All | Many | Med | Few |
| RISE (T1) | 7.23 | 6.77 | 7.95 | 9.61 | 4.44 | 4.15 | 4.94 | 6.19 | 92.54 | 80.12 | 110.96 | 158.86 |
| RISE (T2) | **6.57** | **6.16** | **7.36** | **8.30** | **3.61** | **3.40** | **4.14** | **4.33** | **82.01** | **70.88** | **100.90** | **134.93** |

**Table 15:** Complete ablation of RISE inference strategies with SRL backbone on Dataset A, across multiple metrics. Best results in **bold**

| Method | L1 (MAE) ↓ | | | | GMEAN ↓ | | | | MSE ↓ | | | |
|---|---|---|---|---|---|---|---|---|---|---|---|---|
| | All | Many | Med | Few | All | Many | Med | Few | All | Many | Med | Few |
| Baseline SRL | 7.23 | 6.64 | 8.28 | 9.85 | 4.53 | 4.17 | 5.32 | 6.35 | 91.79 | 77.20 | 115.83 | 163.15 |
| Expert average (I1) | 7.23 | 6.72 | 8.13 | 9.54 | 4.51 | 4.20 | 5.16 | 6.16 | 91.73 | 78.85 | 112.50 | 156.00 |
| Train-based Router (I2) | 7.26 | 6.61 | 8.34 | 10.33 | 4.56 | 4.15 | 5.48 | 6.76 | 92.11 | 76.48 | 116.26 | 173.01 |
| Held-out-based Router (I3) | **6.57** | **6.16** | **7.36** | **8.30** | **3.61** | **3.40** | **4.14** | **4.33** | **82.01** | **70.88** | **100.90** | **134.93** |
| Train+Held-out Router (I4) | 7.24 | 6.65 | 8.31 | 9.98 | 4.55 | 4.19 | 5.34 | 6.37 | 92.25 | 75.37 | 116.51 | 165.69 |
| Train+Held-out Baseline | 7.18 | 6.62 | 8.15 | 9.84 | 4.42 | 4.11 | 5.09 | 5.92 | 90.79 | 76.68 | 112.53 | 163.86 |

**Table 16:** Complete comparison of RISE vs. traditional ensembles on Dataset A, across multiple metrics. Best results in **bold**

| Experiment | Additional Parameters | MSE ↓ | | | | L1 (MAE) ↓ | | | |
|---|---|---|---|---|---|---|---|---|---|
| | | All | Many | Median | Few | All | Many | Median | Few |
| *SRL* | 0 | 91.79 | 77.20 | 115.83 | 163.15 | 7.23 | 6.64 | 8.28 | 9.85 |
| *SRL*+ RISE (K=3) | 2,100,224 | **80.72** | **69.06** | **99.88** | **137.95** | **6.45** | **6.00** | **7.22** | **8.49** |
| *SRL*: 3 ensemble | 3,150,336 | 91.66 | 77.04 | 115.65 | 163.43 | 7.22 | 6.63 | 8.28 | 9.86 |
| *SRL*: 5 ensemble | 5,250,560 | 91.56 | 76.75 | 115.83 | 164.31 | 7.22 | 6.62 | 8.30 | 9.90 |

**Table 17:** Ablation results for Dataset B Rothe et al. (2018b) comparing different RISE-TRAIN configurations. The overall best result is in **bold**.

| Method | L1 (MAE) ↓ | | | | GMEAN ↓ | | | | MSE ↓ | | | |
|---|---|---|---|---|---|---|---|---|---|---|---|---|
| | All | Many | Median | Few | All | Many | Median | Few | All | Many | Median | Few |
| RISE (T1) | 7.94 | 7.46 | 12.66 | 22.75 | 4.50 | 4.33 | 6.89 | 14.66 | 139.69 | 118.35 | 339.35 | 829.19 |
| RISE (T2) | **7.67** | **7.11** | **12.29** | **21.46** | **4.32** | **4.11** | **6.63** | **12.53** | **129.11** | **106.23** | **303.55** | **799.58** |

**Table 18:** Ablation results for K=2 with varying upsampling rates (left) and for $\alpha$=3 with varying expert numbers (K) (right) on Dataset B( Rothe et al. (2018b)). L1 (MAE) metric is shown. The overall best result is in **bold**.

| Config | L1 (MAE) ↓ | | | |
|---|---|---|---|---|
| | All | Many | Median | Few |
| $\alpha$=1 | 7.86 | 7.17 | 13.67 | 23.15 |
| $\alpha$=2 | 7.81 | 7.15 | 13.35 | 22.71 |
| $\alpha$=3 | **7.67** | **7.11** | **12.29** | **21.46** |
| $\alpha$=4 | 7.70 | 7.12 | 12.48 | 21.64 |
| $\alpha$=5 | 7.68 | 7.12 | 12.55 | 21.79 |

| Config | L1 (MAE) ↓ | | | |
|---|---|---|---|---|
| | All | Many | Median | Few |
| K=2 | **7.67** | **7.11** | 12.29 | 21.46 |
| K=3 | 7.69 | 7.17 | **11.89** | **20.90** |
| K=4 | 7.78 | 7.20 | 12.61 | 22.27 |
| K=5 | 8.28 | 7.43 | 15.67 | 25.46 |

**Table 19:** Ablation results comparing different RISE-INFERENCE configurations on Dataset B (Rothe et al. (2018b)). The best baseline result for each metric and data subset is in red, and the overall best result is in **bold**.

| | L1 (MAE) ↓ | | | | GMEAN ↓ | | | | MSE ↓ | | | |
|---|---|---|---|---|---|---|---|---|---|---|---|---|
| Method | All | Many | Median | Few | All | Many | Median | Few | All | Many | Median | Few |
| *Baseline Methods* | | | | | | | | | | | | |
| RankSIM | 7.68 | 7.12 | 12.30 | 21.46 | 4.33 | 4.12 | 6.61 | 12.47 | 129.12 | 106.19 | 304.08 | 799.94 |
| *RISE Inference Strategies* | | | | | | | | | | | | |
| Expert average (I1) | 8.32 | 7.62 | 14.22 | 17.33 | 4.79 | 4.47 | 8.91 | 17.33 | 143.35 | 117.02 | 351.47 | 855.11 |
| Train-based Route (I2) | 8.00 | 7.37 | 13.37 | 14.28 | 4.57 | 4.30 | 7.97 | 14.28 | 135.82 | 111.01 | 329.48 | 826.43 |
| Held-out-based router (I3) | **7.67** | **7.11** | **12.29** | **12.53** | **4.32** | **4.11** | **6.63** | **12.53** | **129.11** | **106.23** | **303.55** | **799.58** |

### E.4 RISE PERFORMANCE WITH BEST-PERFORMING ROUTER CONFIGURATION

To assess the robustness of our approach, we perform five independent experimental runs and report the mean and standard deviation for Dataset A, B & STS-B on each performance metric in Table 20 and the balanced metrics with error bars for Dataset A are reported in Table 21. This evaluation provides statistical insight into the consistency and reliability of the results. For each run, the router is trained and the backbone model achieving the highest routing accuracy on the validation set $D_{val}$ is selected for reporting. Router with the SRL backbone is picked for the the Dataset A (Moschoglou et al. (2017)) dataset, while RankSim backbone is utilized for both IMDB and STS datasets.

Our proposed RISE paradigm consistently outperforms its corresponding baseline methods across multiple metrics, with particularly notable gains in medium- and few-shot regions—where imbalanced regression models typically underperform. These improvements are statistically significant, often exceeding standard error margins. For instance, on the Dataset A (Moschoglou et al. (2017)) dataset, SRL+RISE achieves a 13.7% reduction in Few-shot MAE ($9.85 \rightarrow 8.50$) and a 12.6% reduction in Medium-shot MAE ($8.35 \rightarrow 7.30$), alongside a 28.4% improvement in Few-shot GMEAN ($6.34 \rightarrow 4.54$). Similar trends are observed in Dataset B (Rothe et al. (2018b)), where BalancedMSE+RISE lowers Few-shot MAE by 9.8% ($23.24 \rightarrow 20.97$), and in STS, where LDS+FDS+RISE improves Medium-shot MAE by 11.2% ($0.98 \rightarrow 0.87$).

While RISE generally maintains or improves performance in majority (Many-shot) regions, there are isolated instances where baseline models marginally outperform RISE. For example, in Dataset A (Moschoglou et al. (2017)), RankSIM achieves a slightly lower Many-shot MAE (6.48 vs. 6.56), and in Dataset B (Rothe et al. (2018b)), LDS+FDS reports a marginally better Many-shot MSE (106.61 vs. 107.06). However, these differences are minor and fall within overlapping standard deviation intervals.

Importantly, RISE demonstrates strong generalization by significantly improving performance in minority regions while preserving accuracy on majority classes. This balance highlights the effectiveness of RISE in addressing the fundamental challenge of imbalanced regression, offering a scalable and principled solution for real-world settings.

## F BROADER IMPACT

RISE offers a practical and efficient alternative to end-to-end training by leveraging pre-trained models. Unlike typical deep learning approaches, it requires training only the expert heads and router network while keeping the backbone frozen. This lightweight design makes it feasible for large-scale models and suitable for scenarios where full retraining is impractical. While our experiments used the full training set, RISE can potentially be adapted for final-layer tuning using only a small validation set, as supported by recent adaptation methods (Kirichenko et al. (2023)).

RISE differs from standard fine-tuning by targeting specific regions of poor performance—often underrepresented or minority subgroups—through expert specialization. This targeted improvement enhances fairness, particularly in sensitive applications like healthcare or finance, where disparities in prediction can have serious consequences. By improving minority performance without sacrificing majority accuracy, RISE moves toward more equitable and efficient machine learning systems.

**Table 20:** Comparison of RISE-paired with the baseline methods across Dataset A (Moschoglou et al. (2017)), Dataset B Rothe et al. (2018b), and STS datasets. Results show MAE, GMEAN, and MSE metrics for different data segments (All, Many-shot, Medium-shot, Few-shot). Values are reported as mean ± standard deviation. Best results for each metric and data subset are in bold, we also report the router accuracy for each RISE configuration in parentheses.

| Method | MAE ↓ | | | | GMEAN ↓ | | | | MSE ↓ | | | |
|---|---|---|---|---|---|---|---|---|---|---|---|---|
| | All | Many | Med | Few | All | Many | Med | Few | All | Many | Med | Few |
| **Dataset A** | | | | | | | | | | | | |
| VANILLA | 11.06 ±0.01 | 9.99 ±0.05 | 12.90 ±0.14 | 16.65 ±0.29 | 7.08 ±0.03 | 6.30 ±0.05 | 8.41 ±0.15 | 13.57 ±0.34 | 203.69 ±1.13 | 165.70 ±2.32 | 275.75 ±7.13 | 367.13 ±11.50 |
| VANILLA+RISE (0.60) | **10.07** ±0.04 | **9.20** ±0.08 | **11.19** ±0.14 | **15.33** ±0.21 | **6.18** ±0.04 | **5.52** ±0.06 | **7.19** ±0.11 | **11.99** ±0.28 | **173.84** ±0.77 | **146.76** ±2.37 | **211.69** ±6.80 | **328.25** ±6.86 |
| BalancedMSE | 8.71 ±0.06 | 8.45 ±0.04 | 9.02 ±0.20 | 10.30 ±0.14 | 5.59 ±0.06 | 5.45 ±0.07 | 5.94 ±0.14 | 6.07 ±0.17 | 127.28 ±1.57 | 118.71 ±1.30 | 133.87 ±5.93 | 191.28 ±3.86 |
| BalancedMSE+RISE (0.72) | **7.62** ±0.05 | **7.53** ±0.04 | **7.90** ±0.15 | **8.79** ±0.15 | **4.58** ±0.06 | **4.56** ±0.06 | **4.63** ±0.11 | **4.74** ±0.19 | **106.40** ±1.31 | **103.18** ±1.89 | **111.68** ±4.43 | **158.98** ±3.92 |
| LDS+FDS | 7.47 ±0.08 | 6.92 ±0.11 | 8.23 ±0.13 | 10.52 ±0.25 | 4.77 ±0.06 | 4.46 ±0.07 | 5.30 ±0.10 | 6.84 ±0.23 | 95.23 ±1.85 | **79.98** ±2.73 | 118.33 ±4.26 | 177.20 ±5.09 |
| LDS+FDS+RISE (0.56) | **7.27** ±0.08 | **6.85** ±0.10 | **7.91** ±0.13 | **9.54** ±0.24 | **4.51** ±0.06 | **4.30** ±0.08 | **4.81** ±0.10 | **6.08** ±0.21 | **92.59** ±1.81 | 80.18 ±2.61 | **113.40** ±4.31 | **153.64** ±6.32 |
| RankSIM | 7.01 ±0.04 | **6.48** ±0.06 | 7.82 ±0.08 | 9.85 ±0.09 | 4.55 ±0.04 | 4.14 ±0.04 | 5.37 ±0.05 | 7.04 ±0.19 | 83.23 ±0.77 | **71.48** ±1.37 | 98.21 ±3.40 | 154.26 ±1.35 |
| RankSIM+RISE (0.55) | **6.93** ±0.04 | 6.56 ±0.06 | **7.34** ±0.08 | **9.25** ±0.09 | **4.34** ±0.04 | **4.07** ±0.04 | **4.79** ±0.03 | **6.11** ±0.17 | **82.47** ±0.72 | 73.88 ±1.35 | **90.22** ±3.48 | **143.25** ±1.37 |
| SRL | 7.20 ±0.02 | 6.59 ±0.04 | 8.35 ±0.08 | 9.85 ±0.25 | 4.50 ±0.03 | 4.14 ±0.03 | 5.34 ±0.12 | 6.34 ±0.33 | 91.67 ±0.64 | 76.09 ±0.55 | 118.91 ±1.70 | 165.16 ±6.24 |
| SRL +RISE (0.87) | **6.43** ±0.02 | **5.96** ±0.03 | **7.30** ±0.09 | **8.50** ±0.22 | **3.36** ±0.02 | **3.13** ±0.03 | **3.87** ±0.12 | **4.54** ±0.22 | **80.70** ±0.65 | **67.98** ±0.50 | **103.09** ±1.60 | **140.35** ±5.82 |
| **Dataset B** | | | | | | | | | | | | |
| VANILLA | 8.04 ±0.03 | **7.20** ±0.03 | 15.18 ±0.12 | 26.20 ±0.15 | 4.51 ±0.03 | **4.11** ±0.02 | 10.69 ±0.09 | 18.81 ±0.25 | 137.96 ±0.87 | **108.17** ±0.63 | 366.46 ±6.46 | 972.01 ±7.55 |
| VANILLA+RISE (0.85) | **7.91** ±0.03 | 7.22 ±0.03 | **13.65** ±0.13 | **24.73** ±0.13 | **4.45** ±0.03 | 4.15 ±0.03 | **8.38** ±0.07 | **16.59** ±0.22 | **135.34** ±0.90 | 108.90 ±0.67 | **333.20** ±6.67 | **925.69** ±7.15 |
| BalancedMSE | 8.10 ±0.03 | 7.56 ±0.03 | 12.27 ±0.17 | 23.24 ±0.21 | 4.68 ±0.01 | 4.45 ±0.01 | 7.10 ±0.11 | 13.25 ±0.26 | 139.62 ±1.55 | 116.96 ±1.15 | 302.67 ±9.31 | 868.31 ±14.02 |
| BalancedMSE+RISE (0.81) | **7.73** ±0.03 | **7.28** ±0.02 | **12.12** ±0.13 | **20.97** ±0.25 | **4.41** ±0.01 | **4.29** ±0.00 | **6.82** ±0.07 | **11.99** ±0.52 | **136.36** ±1.50 | **108.79** ±1.01 | **300.42** ±8.45 | **820.76** ±13.03 |
| LDS+FDS | 7.70 ±0.01 | 7.13 ±0.01 | 12.54 ±0.06 | 21.84 ±0.39 | 4.32 ±0.01 | 4.11 ±0.01 | 7.55 ±0.10 | 12.75 ±0.38 | **129.91** ±0.85 | **106.61** ±0.57 | 310.90 ±2.90 | 781.84 ±21.50 |
| LDS+FDS+RISE (0.81) | **7.64** ±0.02 | **7.11** ±0.01 | **12.09** ±0.06 | **21.24** ±0.38 | **4.27** ±0.01 | **4.07** ±0.01 | **6.46** ±0.09 | **12.17** ±0.37 | 131.02 ±0.86 | 107.06 ±0.61 | **301.96** ±3.08 | **768.11** ±20.25 |
| RankSIM | 7.69 ±0.02 | 7.12 ±0.02 | 12.33 ±0.12 | 21.55 ±0.37 | 4.33 ±0.01 | 4.12 ±0.01 | 6.65 ±0.08 | 12.68 ±0.41 | 129.14 ±0.72 | **106.78** ±0.43 | 302.58 ±5.76 | 802.83 ±28.19 |
| RankSIM+RISE (0.8) | **7.66** ±0.02 | **7.11** ±0.02 | **12.08** ±0.12 | **20.38** ±0.37 | **4.30** ±0.01 | **4.09** ±0.01 | **6.48** ±0.08 | **12.54** ±0.41 | **127.49** ±0.74 | 108.04 ±0.33 | **298.66** ±5.90 | **800.72** ±29.06 |
| SRL | 7.70 ±0.02 | **7.13** ±0.03 | 12.66 ±0.09 | 21.94 ±0.50 | **4.34** ±0.01 | **4.13** ±0.01 | 6.93 ±0.04 | 12.93 ±0.47 | 131.96 ±1.11 | **107.38** ±0.69 | 337.57 ±5.73 | **768.85** ±25.85 |
| SRL+RISE (0.79) | **7.68** ±0.02 | 7.19 ±0.02 | **11.98** ±0.09 | **19.39** ±0.50 | 4.35 ±0.02 | 4.15 ±0.01 | **6.43** ±0.04 | **11.22** ±0.47 | **130.07** ±1.11 | 107.54 ±0.70 | **298.14** ±5.63 | 773.48 ±24.93 |
| **STS-B** | | | | | | | | | | | | |
| LDS+FDS | 0.77 ±0.00 | **0.72** ±0.01 | 0.98 ±0.02 | 0.76 ±0.02 | 0.38 ±0.01 | 0.33 ±0.01 | 0.67 ±0.02 | 0.45 ±0.01 | **0.91** ±0.01 | 0.81 ±0.01 | **1.06** ±0.05 | 0.94 ±0.06 |
| LDS+FDS+RISE (0.51) | **0.75** ±0.00 | 0.73 ±0.00 | **0.87** ±0.02 | **0.66** ±0.02 | **0.30** ±0.01 | **0.25** ±0.01 | **0.56** ±0.01 | **0.34** ±0.02 | 0.92 ±0.01 | **0.79** ±0.00 | 1.08 ±0.06 | **0.76** ±0.04 |
| RankSIM | 0.76 ±0.00 | 0.74 ±0.01 | **0.75** ±0.01 | 0.64 ±0.04 | 0.50 ±0.02 | 0.47 ±0.02 | **0.54** ±0.01 | 0.37 ±0.03 | 0.86 ±0.01 | 0.86 ±0.02 | **0.85** ±0.01 | **0.63** ±0.03 |
| RankSIM+RISE (0.55) | **0.73** ±0.01 | **0.73** ±0.01 | 0.75 ±0.01 | **0.63** ±0.05 | **0.39** ±0.01 | **0.37** ±0.01 | 0.54 ±0.02 | **0.36** ±0.04 | **0.84** ±0.02 | **0.84** ±0.02 | 0.85 ±0.02 | 0.67 ±0.09 |
| SRL | 0.89 ±0.01 | 0.84 ±0.01 | 1.07 ±0.04 | 0.98 ±0.07 | 0.63 ±0.02 | 0.57 ±0.02 | 0.79 ±0.05 | 0.69 ±0.09 | 1.17 ±0.02 | 1.07 ±0.02 | 1.57 ±0.08 | 1.30 ±0.12 |
| SRL+RISE (0.57) | **0.82** ±0.01 | **0.80** ±0.00 | **0.93** ±0.03 | **0.76** ±0.07 | **0.43** ±0.01 | **0.39** ±0.02 | **0.70** ±0.04 | **0.36** ±0.05 | **1.06** ±0.018 | **1.01** ±0.01 | **1.24** ±0.08 | **1.14** ±0.17 |

**Table 21:** Comparison of RISE with baseline methods for Dataset A (Moschoglou et al. (2017)) with balanced metrics. Values are reported as mean ± standard deviation. Best results for each metric and data subset are in bold.

| Method | bMAE ↓ | | | | bGMEAN ↓ | | | | bMSE ↓ | | | |
|---|---|---|---|---|---|---|---|---|---|---|---|---|
| | All | Many | Med | Few | All | Many | Med | Few | All | Many | Med | Few |
| **Dataset A** | | | | | | | | | | | | |
| VANILLA | 13.18 | 9.99 | 12.94 | 19.84 | 7.30 | 6.30 | 8.41 | 13.57 | 271.42 | 165.70 | 276.55 | 483.80 |
| | ±0.06 | ±0.05 | ±0.16 | ±0.25 | ±0.08 | ±0.05 | ±0.15 | ±0.34 | ±3.33 | ±2.32 | ±8.22 | ±10.75 |
| VANILLA+RISE | **12.15** | **9.20** | **11.18** | **18.81** | **6.10** | **5.52** | **7.19** | **11.99** | **242.73** | **146.76** | **210.97** | **458.86** |
| | ±0.04 | ±0.08 | ±0.15 | ±0.17 | ±0.08 | ±0.06 | ±0.11 | ±0.28 | ±2.29 | ±2.37 | ±7.28 | ±8.16 |
| BalancedMSE | 9.35 | 8.45 | 8.99 | 11.40 | 6.44 | 5.45 | 5.94 | 6.07 | 153.18 | 118.71 | 132.34 | 236.83 |
| | ±0.07 | ±0.04 | ±0.22 | ±0.14 | ±0.10 | ±0.07 | ±0.14 | ±0.17 | ±1.89 | ±1.30 | ±6.34 | ±3.38 |
| BalancedMSE+RISE | **8.44** | **7.53** | **7.82** | **10.61** | **5.35** | **4.56** | **4.63** | **4.70** | **134.30** | **103.18** | **110.51** | **217.58** |
| | ±0.07 | ±0.04 | ±0.17 | ±0.16 | ±0.11 | ±0.06 | ±0.11 | ±0.19 | ±1.72 | ±1.89 | ±4.80 | ±3.79 |
| LDS+FDS | 9.35 | 8.45 | 8.99 | **11.40** | 5.74 | 5.45 | 5.94 | **6.07** | 153.18 | 118.71 | 132.34 | 236.83 |
| | ±0.07 | ±0.04 | ±0.22 | ±0.14 | ±0.11 | ±0.07 | ±0.14 | ±0.17 | ±1.89 | ±1.30 | ±6.34 | ±3.38 |
| LDS+FDS+RISE | **8.31** | **6.85** | **7.91** | 11.55 | **5.61** | **4.30** | **4.81** | 6.08 | **122.83** | **80.18** | **112.24** | **216.64** |
| | ±0.07 | ±0.10 | ±0.14 | ±0.25 | ±0.09 | ±0.08 | ±0.10 | ±0.31 | ±1.05 | ±2.61 | ±4.48 | ±6.09 |
| RankSIM | 8.07 | **6.48** | 7.81 | 11.46 | 6.14 | 4.14 | 5.37 | 7.04 | 111.49 | **71.48** | 97.46 | 202.09 |
| | ±0.04 | ±0.06 | ±0.08 | ±0.05 | ±0.07 | ±0.04 | ±0.05 | ±0.19 | ±0.47 | ±1.37 | ±3.37 | ±0.88 |
| RankSIM+RISE | **7.92** | 6.56 | **7.31** | **11.08** | **5.09** | **4.07** | **4.79** | **6.11** | **108.90** | 73.88 | **89.22** | **192.92** |
| | ±0.03 | ±0.06 | ±0.08 | ±0.05 | ±0.06 | ±0.04 | ±0.03 | ±0.17 | ±0.43 | ±1.35 | ±3.45 | ±1.04 |
| SRL | 8.28 | 6.59 | 8.41 | 11.65 | 5.15 | 4.14 | 5.34 | 6.34 | 121.11 | 76.09 | 118.76 | 214.45 |
| | ±0.04 | ±0.04 | ±0.08 | ±0.18 | ±0.06 | ±0.03 | ±0.12 | ±0.33 | ±1.26 | ±0.55 | ±1.58 | ±4.64 |
| SRL+RISE | **7.36** | **5.96** | **7.32** | **10.24** | **4.40** | **3.13** | **3.87** | **4.54** | **105.65** | **67.98** | **102.10** | **184.75** |
| | ±0.04 | ±0.03 | ±0.09 | ±0.17 | ±0.05 | ±0.03 | ±0.12 | ±0.22 | ±1.26 | ±0.50 | ±1.50 | ±4.43 |

## G  EXPERIMENTS DEMONSTRATING FUNCTIONAL HETEROGENEITY IN DIR

To provide stronger evidence that the head vs. tail regions in DIR datasets A and B (Moschoglou et al. (2017); Rothe et al. (2018b)) correspond to fundamentally different predictive functions, we conducted two additional experiments that directly target this concern: **(1) Freeze-and-Probe:** testing feature transferability, and **(2) Gradient Cosine Similarity (GCS):** measuring optimization conflict.

### G.1  FREEZE-AND-PROBE: TESTING FEATURE TRANSFERABILITY

**Experimental Setup.** The goal of this experiment is to isolate *feature transferability* as the only factor under study. To do so, we fix the entire ResNet–50 backbone in both models and train only a newly initialized linear layer on the scarce Tail-Train data (e.g., label values $< 15$). By freezing all convolutional layers, we eliminate effects from forgetting, overfitting, or capacity differences, ensuring that any performance difference must arise solely from the quality of the underlying feature representation.

Both models are trained under identical conditions: identical linear probe architecture, identical L2 regularization, identical optimization hyperparameters, and identical early stopping based on the Tail-Val set. The *only* difference is the source of the frozen backbone:

- **Model B (General-Feature Baseline):** Frozen ImageNet-pretrained ResNet–50 backbone. This represents strong, general-purpose features not biased toward any label region in our dataset.

- **Model A-Probe (Head-Feature Test):** ResNet–50 backbone first fine-tuned *only on the Head region* (e.g., label values 20–40), then frozen. This tests whether features specialized for the head region transfer effectively to the tail.

We train only the linear layer for both models using the same Tail-Train data and evaluate the best checkpoint (chosen via Tail-Val early stopping) on the held-out Tail-Test set.

**Table 22:** Freeze-and-Probe: Tail Test MAE Comparison

| Dataset | ImageNet (B) | Head-pretrained (A-Probe) | Relative Drop |
|---------|--------------|---------------------------|---------------|
| *Dataset A* | $3.1844 \pm 0.06$ | $4.7558 \pm 0.07$ | $33.00\%$ |
| *Dataset B* | $2.2211 \pm 0.01$ | $2.9510 \pm 0.01$ | $24.00\%$ |

**Observation.** As shown in Table 22, Model A-Probe performs substantially worse than Model B on both datasets. Because all other variables are held fixed, this degradation cannot be attributed to scarcity or overfitting. Instead, it provides a direct, unconfounded demonstration of **negative transfer**: features optimized for the Head region are not only suboptimal but actively harmful for Tail predictions, supporting our claim that the two regions correspond to fundamentally different predictive functions.

### G.2  GRADIENT COSINE SIMILARITY (GCS): EVIDENCE OF OPTIMIZATION CONFLICT

**Initial MSE-Based Analysis (Confounded).** We first computed GCS using the standard MSE loss between balanced batches from the head and tail regions for Dataset A using a monolithic model with ResNet-50 as backbone. Let $\mathcal{B}_h, \mathcal{B}_t$ be two balanced mini-batches sampled from the head and tail regions. For a parameter vector $\theta$ (or a chosen layer's parameters) define

$$g_h \;=\; \frac{1}{|\mathcal{B}_h|} \sum_{(x,y)\in\mathcal{B}_h} \nabla_\theta \ell(x,y), \qquad g_t \;=\; \frac{1}{|\mathcal{B}_t|} \sum_{(x,y)\in\mathcal{B}_t} \nabla_\theta \ell(x,y).$$

The Gradient Cosine Similarity (GCS) is

$$\mathrm{GCS}(g_h, g_t) \;=\; \frac{\langle g_h, g_t \rangle}{\|g_h\|_2 \, \|g_t\|_2}.$$

In practice we compute GCS per-layer and for the final- fully connected layer (fc) by flattening the corresponding parameter gradients into vectors. We report epoch-wise means over multiple runs (10 random seeds) and over several balanced mini-batches.

As shown in Table 23, deeper layers exhibit strongly negative GCS values during training. However, this signal is *mechanically confounded*. For a monolithic regressor $\hat{y} = w^\top \phi(x)$ with loss $\ell = \frac{1}{2}(y - \hat{y})^2$, the gradient

$$\nabla_w \ell = -(y - \hat{y})\, \phi(x)$$

is scaled by the *signed residual*. Since head and tail typically lie on opposite sides of the model's current prediction, their residuals have opposite signs, forcing the gradients to be antiparallel even when the underlying feature gradients $\nabla \phi$ are aligned. Thus, negative MSE-GCS does not reliably indicate functional conflict; it is induced by the regression loss itself.

**Table 23:** Average GCS (Head vs. Tail) using MSE Loss on Dataset A

| Epoch | layer1 | layer4 | fc |
|---|---|---|---|
| 1 | +0.0069 | -0.0361 | -0.0493 |
| 10 | +0.0605 | -0.0567 | -0.0551 |
| 50 | -0.0159 | -0.3198 | -0.9336 |
| 100 | +0.0364 | -0.2923 | -0.9815 |

**Unconfounded Experimental Setup (CE Surrogate).** To obtain a clean measure of optimization alignment, we follow the surrogate strategy of Niu et al. (2016): discretize the continuous target into 101 bins and train a 101-way classifier using cross-entropy. The gradient in this setting,

$$\nabla \ell = \hat{\mathbf{p}} - \mathbf{p},$$

contains no residual-dependent sign flip, so the cosine similarity of the gradient vectors: $\cos(\nabla \ell_{\text{head}}, \nabla \ell_{\text{tail}})$ directly reflects true optimization conflict. We compute GCS at every epoch during joint training on head and tail batches.

**Table 24:** Average GCS (Head vs. Tail) using CE Loss on Dataset A

| Epoch | layer1 | layer4 | fc |
|---|---|---|---|
| 1 | -0.0147 | -0.0527 | -0.1799 |
| 10 | -0.0370 | -0.0707 | -0.5965 |
| 50 | +0.0276 | -0.1183 | -0.4788 |
| 100 | +0.0444 | -0.1914 | -0.4905 |

**Table 25:** Average GCS (Head vs. Tail) using CE Loss on Dataset B

| Epoch | layer1 | layer4 | fc |
|---|---|---|---|
| 1 | +0.0283 | -0.0404 | -0.1258 |
| 10 | +0.0132 | -0.0512 | -0.3165 |
| 50 | +0.0154 | -0.1081 | -0.3754 |
| 100 | +0.0121 | -0.1104 | -0.3408 |

**Observation.** Across training, deeper layers (layer4, fc) exhibit persistently negative GCS values across both datasets (approximately $-0.18$ to $-0.60$ even in early epochs), while lower layers remain near zero ($-0.03$ to $+0.04$). This aligns with architectural intuition: early CNN layers encode generic edges/textures shared across the label space, whereas higher layers encode semantic attributes that differ sharply between tail and head regions. Persistent negative GCS indicates that updates lowering head loss tend to increase tail loss, and vice versa, revealing that the two regions exert inherently conflicting optimization pressures (Wang et al. (2020b)).

Combined with our Freeze-and-Probe results, this provides direct causal evidence that head and tail correspond to distinct predictive functions, and that the head–tail tradeoff arises from *task conflict*, not data scarcity.

## H EXPERIMENTS USING 80-20 SPLIT OF TRAINING DATA

To further demonstrate that our performance gains do not stem from using held-out validation data, we conducted an additional ablation experiment on Dataset A Moschoglou et al. (2017).

**Experimental Setup:** We designed the experiment as follows:

1. **Data Splitting:** We split the original training set into two subsets: (i) `train1`, comprising 80% of the training data, and (ii) `train2`, a balanced dataset (Yang et al. (2021a)) containing the remaining 20%.

2. **Baseline Training:** We trained all baselines from scratch on `train1`, using `train2` as the validation set for LDS-FDS[2] and SRL[3], using the code publicly released by the respective authors. We denote these models as LDS-FDS(*train-split*) and SRL(*train-split*) respectively.

3. **RISE Training:** On these newly trained baselines, we applied our three-stage RISE approach: (i) RISE-IDENTIFY on `train2` to identify underperforming regions, (ii) RISE-TRAIN on `train1` to train expert models on the identified regions, and (iii) RISE-INFER on `train2` to train the routing mechanism. We denote this as RISE(*train-split*).

**Results and Discussion:** Table 26 presents the results of this ablation study. We observe a consistent trend across both methods: RISE(*train-split*) not only outperforms its corresponding baseline(*train-split*) models but also surpasses the original baselines trained on the entire training data. This finding strongly supports our claim that RISE's performance improvements arise from its MOE based architecture and principled identification and targeting of underperforming regions rather than from exploiting additional validation data.

**Table 26:** Ablation result on Dataset A by training baseline and corresponding RISE configuration from scratch on 80% split of train dataset. The best result for baseline and corresponding RISE is in **bold** with the router accuracy reported in brackets.

| Method | L1 (MAE) ↓ | | | | GMEAN ↓ | | | | MSE ↓ | | | |
|---|---|---|---|---|---|---|---|---|---|---|---|---|
| | All | Many | Med | Few | All | Many | Med | Few | All | Many | Med | Few |
| *SRL Methods* | | | | | | | | | | | | |
| SRL-Original (on entire train) | 7.23 | 6.64 | 8.28 | 9.85 | 4.53 | 4.17 | 5.32 | 6.35 | 91.79 | 77.20 | 115.83 | 163.15 |
| SRL-Original+RISE (on entire train) (0.87) | **6.57** | **6.16** | **7.36** | **8.30** | **3.61** | **3.40** | **4.14** | **4.33** | **82.01** | **70.88** | **91.20** | **134.93** |
| SRL (*train1-split*) | 7.36 | 6.60 | 8.84 | 10.53 | 4.62 | 4.11 | 5.88 | 7.21 | 94.42 | 76.03 | 127.43 | 176.77 |
| SRL+RISE (*train1-split*) (0.74) | **6.88** | **6.36** | **7.73** | **9.51** | **4.05** | **3.69** | **4.84** | **5.89** | **87.36** | **74.32** | **106.97** | **156.20** |
| *LDS-FDS Methods* | | | | | | | | | | | | |
| LDS-FDS-Original (on entire train) | 7.47 | 6.91 | 8.27 | 10.58 | 4.77 | 4.44 | 5.33 | 6.87 | 95.32 | 79.71 | 118.52 | 178.58 |
| LDS-FDS-Original+RISE (on entire train) (0.56) | **7.28** | **6.79** | **8.07** | **9.72** | **4.49** | **4.25** | **4.88** | **6.04** | **92.79** | **78.88** | **116.49** | **158.63** |
| LDS-FDS(*train1-split*) | 7.71 | 6.78 | 8.83 | 12.96 | 4.95 | 4.39 | 5.77 | 9.47 | 99.51 | 75.47 | 126.50 | 248.01 |
| LDS-FDS + RISE (*train1-split*) (0.58) | **7.40** | **6.58** | **8.14** | **10.31** | **4.39** | **4.31** | **5.32** | **6.73** | **93.54** | **78.51** | **118.25** | **165.16** |

## I ADDITIONAL ROUTER TRAINING ABLATION

We evaluate routers trained on: (i) training data only, (ii) held-out validation data (our proposed approach), (iii) train+validation union, (iv) balanced training data, and (v) balanced training + validation. We include three baseline configurations: SRL trained on full training data, train+validation union (matching RISE's data access), and 80% of training data (matching RISE's reduced training set). Table 27 presents MAE performance and router accuracy on Dataset A:

**Analysis and Key Findings:**

**(1) Routing mechanism:** Comparing held-out validation router (6.57 MAE, 87% accuracy) versus train+val union router (7.24 MAE, 45% accuracy)—where expert training is identical—isolates the routing strategy's contribution. **(2) Data volume:** To address whether RISE's gains stem from using validation data unavailable to baselines, we retrained SRL on the Train+Val union, giving it all data RISE uses for meta-learning. Importantly, no RISE expert weights are trained on held-out data—it is

---

[2] `https://github.com/YyzHarry/imbalanced-regression`
[3] `https://github.com/yilei-wu/imbalanced-regression`

**Table 27:** Complete router training ablation on Dataset A. Router accuracy measures the fraction of samples correctly assigned to their ground-truth expert. Best results in bold.

| Method | Router Acc. | All | Many | Med | Few |
|---|---|---|---|---|---|
| *Baselines* | | | | | |
| SRL baseline (full train) | - | 7.23 | 6.64 | 8.28 | 9.85 |
| SRL baseline (Train+Val union) | - | 7.18 | 6.62 | 8.15 | 9.84 |
| SRL baseline (80% train) | - | 7.37 | 6.60 | 8.84 | 10.53 |
| *RISE Variants* | | | | | |
| SRL + RISE (held-out val router) | **0.87** | **6.57** | **6.16** | **7.36** | **8.30** |
| SRL + RISE (20% train held-out) | 0.74 | 6.88 | 6.36 | 7.73 | 9.51 |
| SRL + RISE (train-based router) | 0.43 | 7.26 | 6.61 | 8.34 | 10.33 |
| SRL + RISE (train+val union router) | 0.45 | 7.24 | 6.65 | 8.32 | 9.98 |

reserved exclusively for meta-learning (region discovery and router training). The baseline achieves minimal improvement (7.23 to 7.18 MAE, 0.7% reduction), while RISE achieves (7.23 to 6.57 MAE, 9.1% reduction). This demonstrates that RISE's advantage stems from architectural separation of expert training and meta-learning, not from privileged data access.**(3) Importance of held-out data:** SRL baseline trained on 80% of data achieves 7.37 MAE, while SRL+RISE with 20% held-out achieves 6.88 MAE (74% router accuracy). Despite using 20% less training data, RISE outperforms the full-data baseline by 6.6%. This result aligns with recent work showing that held-out data enables distinguishing memorization from generalization Bayat et al. (2025); Qiu et al. (2023) and effective post-hoc model improvement (Kirichenko et al., 2023). Unlike classification methods that use held-out data for reweighting or retraining a single model Liu et al. (2021); Qiu et al. (2023), RISE uses it exclusively for meta-learning—identifying failure regions and training the router—ensuring it remains an unbiased signal of generalization performance. This approach extends held-out-based failure discovery to regression, where threshold-free identification and spatially contiguous regions are essential. **(4) Router accuracy and performance correlation:** The strong correlation between router accuracy (held-out approaches: 87%, 74% vs. training-based approaches: 43–45%) and performance (6.57–6.88 vs. 7.24–7.26 MAE) confirms Theorem 2's prediction. Routers trained on training data cannot distinguish generalization failures from training artifacts; held-out data enables genuine meta-learning where the router selects experts that *generalize* best, not those that memorize best. The train+val union router (0.45 accuracy, 7.24 MAE) performs similarly to the train-based router (0.43 accuracy, 7.26 MAE) despite more data, confirming that data separation, not volume, is critical.

## J  ADDITIONAL EXPERIMENTS FOR FAIRNESS EVALUATION

Apart from well-documented DIR metrics such as bMAE (Ren et al. (2022)) and GMEAN (Yang et al. (2021b)) which provide a more robust and label-frequency agnostic evaluation of regression models. In this section we provide evaluation results for RISE using other imbalanced regression metrics such as SERA and RW-RMSE (Ribeiro & Moniz (2020); Silva et al. (2022)).

SERA (Squared Error Relevance Area) and RW-RMSE (Relevance-Weighted Root Mean Squared Error) are both measured using a relevance function (Torgo & Ribeiro (2007)) ($\phi : Y \rightarrow [0, 1]$) is a continuous function mapping label space $Y$ into a $[0, 1]$ scale of relevance where 0 and 1 represent the minimum and maximum relevance. Torgo & Ribeiro (2007) introduced a way that uses box-plot to automatically assign relevance to different labels. Once this mapping relevance function has been identified, for a dataset $D = \{(x_i, y_i)\}_{i=1}^{N}$, with $x_i$ and $y_i$ being the feature and label space respectively, we define SERA as-

$$SERA = \int_0^1 \sum_{y_i \in D^t} (\hat{y}_i - y_i)^2 dt \qquad (21)$$

with $D_t$ being defined as $D_t = \{(x_i, y_i) \in D \mid \phi(y_i) \geq t\}$. Similarly based on $\phi$, we define RW-RMSE as

$$RW - RMSE = \sqrt{\frac{\sum_{i=0}^{N} \phi(y_i)(\hat{y}_i - y_i)^2}{\sum_{i=0}^{N} \phi(y_i)}} \qquad (22)$$

For our implementation, we used IRon[4], a public R implementation of SERA metric with `method = "extremes"`, `extr.type = "both"` and `coef = 1.5` as hyper-parameters.

We report the results using these metrics on Dataset A (Moschoglou et al. (2017)), Dataset B (Rothe et al. (2018b)) and STS-B(Cer et al. (2017b)) in Table 28, 29 and 30 respectively. Across these tables, we observe that the RISE framework consistently improves SERA and RW-RMSE errors over their respective baseline.

**Table 28:** Overall SERA and RW-RMSE comparison between baseline and RISE methods on Dataset A (Moschoglou et al. (2017)) with the best results of baseline and corresponding RISE in **bold**.

| Method | Baseline | | RISE | |
|---|---|---|---|---|
| | SERA | RW-RMSE | SERA | RW-RMSE |
| Vanilla | 185632.05 | 15.34 | **178765.91** | **15.06** |
| BMSE | 104793.37 | 11.53 | **86088.50** | **10.45** |
| LDS-FDS | 89469.81 | 10.65 | **81101.40** | **10.14** |
| RankSIM | 76067.22 | 9.82 | **72396.88** | **9.58** |
| SRL | 80512.74 | 10.10 | **68728.10** | **9.33** |

**Table 29:** Overall SERA and RW-RMSE comparison between baseline and RISE methods on Dataset B (Rothe et al. (2018a)) with the best results of baseline and corresponding RISE in **bold**.

| Method | Baseline | | RISE | |
|---|---|---|---|---|
| | SERA | RW-RMSE | SERA | RW-RMSE |
| Vanilla | 1204812.95 | 14.09 | **1155859.68** | **13.85** |
| BMSE | 1199632.82 | 13.71 | **1060634.34** | **13.63** |
| LDS-FDS | 1085458.96 | 13.34 | **1065592.69** | **13.33** |
| RankSIM | 1086665.20 | 13.29 | **1038208.49** | **13.22** |
| SRL | 1113412.91 | 13.56 | **1061609.02** | **13.25** |

**Table 30:** Overall SERA and RW-RMSE comparison between baseline and RISE methods on STS-B (Cer et al. (2017b)) with the best results of baseline and corresponding RISE in **bold**.

| Method | Baseline | | RISE | |
|---|---|---|---|---|
| | SERA | RW-RMSE | SERA | RW-RMSE |
| LDS-FDS | 286.58 | 1.11 | **261.85** | **1.06** |
| RankSIM | 305.22 | 1.14 | **264.67** | **1.06** |
| SRL | 327.53 | 1.18 | **288.28** | **1.11** |

To further strengthen our fairness claims, we evaluate RISE on Dataset A and B by dividing the datasets in three groups old ($y \geq 80$), adult ($y \in [18, 80)$) and young ($y < 18$), for each of these groups ($D_g$) we calculate SERA metric, Normalized-SERA (Norm-SERA = $\frac{SERA(D_g)}{|D_g|}$) and MAE. Further, we also calculate Worst-Case Disparity (WCD) as-

$$WCD = Max(MAE(D_i)) - Min(MAE(D_i)) \, \forall i \in [1, \, ||D_g||] \tag{23}$$

WCD takes reference from the Statistical Parity Difference (SPD) Dwork et al. (2012), a key metric implemented in toolkits like AI Fairness 360 (AIF360), using the maximum difference between extreme group outcomes to identify bias. Tailored for regression, it quantifies the maximum disparity in prediction error rather than measuring differences in positive outcome rates between groups. This focus on error magnitude is crucial for assessing robust model performance for all groups, specifically confirming that the system is not concentrating its largest prediction errors on any single subgroup Sagawa et al. (2020).

The results across these groups are provided for Dataset A in Table 31 and for Dataset B in 32. For both the datasets, we consistently observe that SERA across the groups gets reduced as well as WCD is reduced for RISE vis-a-vis the corresponding baseline

---

[4] https://github.com/nunompmoniz/IRon

**Table 31:** SERA values and normalized SERA across groups, along with MAE per group and worst-group disparity on Dataset A (Moschoglou et al. (2017)) with the best result of baseline and its corresponding RISE configuration in **bold**.

| Method | Young SERA | | Adult SERA | | Old SERA | | MAE by Group | | | |
| | Raw | Normalized | Raw | Normalized | Raw | Normalized | Young | Adult | Old | WCD |
|---|---|---|---|---|---|---|---|---|---|---|
| Vanilla | 47057.23 | 480.18 | 107441.23 | 57.92 | 41548.28 | 222.18 | 19.59 | 10.37 | 14.50 | 9.21 |
| Vanilla+RISE | **36642.55** | **373.90** | **99623.81** | **53.71** | **32084.87** | **171.58** | **17.28** | **9.70** | **12.91** | **7.58** |
| LDS-FDS | 22511.14 | 229.71 | 51426.41 | 27.72 | 15532.26 | 83.06 | 12.78 | 7.09 | 8.45 | 5.69 |
| LDS-FDS+RISE | **18157.90** | **185.28** | **47482.65** | **25.60** | **15460.84** | **82.68** | **10.79** | **6.98** | **8.38** | **3.80** |
| RankSIM | 16393.81 | 167.28 | 45531.07 | 24.55 | 14142.34 | 75.63 | 10.31 | 6.71 | 8.46 | 3.60 |
| RankSIM+RISE | **15335.53** | **156.49** | **44271.32** | **23.87** | **12790.03** | **68.40** | **9.67** | **6.70** | **7.83** | **2.97** |
| SRL | 15880.41 | 162.05 | 47009.29 | 25.34 | 17623.04 | 94.24 | 10.01 | 6.86 | 9.36 | 3.15 |
| SRL+RISE | **13874.21** | **141.57** | **41224.47** | **22.22** | **13629.41** | **72.88** | **8.90** | **6.26** | **7.57** | **2.65** |

**Table 32:** SERA values and normalized SERA across groups, along with MAE per group and worst-group disparity on Dataset B (Rothe et al. (2018a)) with the best result of baseline and its corresponding RISE configuration in **bold**.

| Method | Young SERA | | Adult SERA | | Old SERA | | MAE by Group | | | |
| | Raw | Normalized | Raw | Normalized | Raw | Normalized | Young | Adult | Old | WCD |
|---|---|---|---|---|---|---|---|---|---|---|
| Vanilla | 338905.27 | 225.64 | 636982.43 | 70.50 | 228925.25 | 472.01 | **10.31** | 7.49 | 14.35 | 6.86 |
| Vanilla+RISE | **334993.39** | **223.03** | **621537.61** | **68.79** | **199328.68** | **410.99** | 11.21 | **7.20** | **12.73** | **5.53** |
| LDS-FDS | 293761.17 | 195.58 | 593789.73 | 65.72 | 197908.06 | 408.06 | 9.77 | 7.00 | 14.99 | 7.98 |
| LDS-FDS+RISE | **293284.22** | **195.26** | **584400.34** | **64.68** | **187908.12** | **387.44** | **9.72** | **6.93** | **13.75** | **6.83** |
| RankSIM | 298707.74 | 198.87 | 607310.86 | 67.22 | 180646.59 | 372.47 | 9.81 | **7.05** | 12.96 | 5.91 |
| RankSIM+RISE | **268222.22** | **178.58** | **593226.72** | **65.66** | **176759.56** | **364.45** | **8.79** | 7.18 | **12.89** | **5.71** |
| SRL | 281178.22 | 187.20 | 640468.17 | 70.89 | 191766.52 | 395.39 | 9.63 | **7.09** | 13.54 | 6.46 |
| SRL+RISE | **270906.40** | **180.36** | **604839.63** | **66.94** | **185863.00** | **383.22** | **8.73** | 7.17 | **12.89** | 5.72 |

## K  ADDITIONAL EXPERIMENT ON ENSEMBLE TRAINING

In addition to comparing RISE with bagging ensemble models (Table 16), we conducted experiments comparing RISE against boosting-based approaches, specifically AdaBoost.

**AdaBoost Implementation:** Our AdaBoost implementation (Solomatine & Shrestha (2004)) employs a frozen ResNet-50 backbone with multiple expert heads. Each expert is sequentially trained on weighted bootstrap samples, where the algorithm adaptively reweights training instances based on age group prediction errors. This enables subsequent experts to focus on poorly performing age ranges. Final predictions are obtained by combining expert outputs using learned alpha weights proportional to their individual performance. We trained ensembles with $k = 3$ and $k = 5$ experts using both LDS-FDS and SRL loss functions, and compared their performance against RISE variants.

**Results and Analysis:** Table 33 presents the comparative results. While AdaBoost outperforms both the baseline and bagging methods (see Table 16), its performance gains are primarily concentrated in the *all* and *many* data bands, with the *median* and *few* bands showing only marginal improvements over the baseline. In contrast, RISE consistently outperforms all AdaBoost configurations across all evaluation metrics and data frequency bands, achieving superior performance with significantly fewer additional parameters. This demonstrates RISE's ability to effectively handle underperforming regions across the entire data distribution, not just the well-represented segments.

**Table 33:** Comparison of baseline and rise with over-parameterized AdaBoost ensemble configurations for Dataset A (Moschoglou et al. (2017)). The best result between baseline and corresponding Adaboost configuration is in **bold** and the best overall result is presented in color blue.

| Method | Extra Parameters | L1 (MAE) ↓ | | | | GMEAN ↓ | | | | MSE ↓ | | | |
|---|---|---|---|---|---|---|---|---|---|---|---|---|---|
| | | All | Many | Med | Few | All | Many | Med | Few | All | Many | Med | Few |
| **LDS-FDS Methods** | | | | | | | | | | | | | |
| LDS-FDS-Baseline | 0 | 7.47 | 6.91 | **8.27** | **10.58** | 4.77 | 4.44 | 5.33 | 6.87 | 95.32 | 79.71 | **118.52** | **178.58** |
| LDS-FDS+RISE | 2,100,224 | 7.28 | 6.79 | 8.07 | 9.72 | 4.49 | 4.25 | 4.88 | 6.04 | 92.79 | 78.88 | 116.49 | 158.63 |
| LDS-FDS-k=3 | 3,150,336 | 7.44 | 6.86 | 8.29 | 10.60 | 4.74 | 4.39 | 5.33 | 6.73 | 94.70 | 77.48 | 119.11 | 178.84 |
| LDS-FDS-k=5 | 5,250,560 | **7.43** | **6.82** | 8.31 | 10.61 | **4.71** | **4.36** | **5.33** | **6.81** | **94.29** | **76.63** | 119.44 | 178.98 |
| **SRL Methods** | | | | | | | | | | | | | |
| SRL-Baseline | 0 | 7.23 | 6.64 | **8.28** | 9.85 | 4.53 | 4.17 | **5.32** | 6.35 | 91.79 | 77.20 | **115.83** | **163.15** |
| SRL+RISE | 2,100,224 | 6.57 | 6.16 | 7.36 | 8.30 | 3.61 | 3.40 | 4.14 | 4.33 | 82.01 | 70.88 | 91.20 | 134.93 |
| SRL-k=3 | 3,150,336 | 7.17 | 6.61 | 8.28 | 9.84 | 4.52 | 4.16 | 5.34 | 6.34 | 90.57 | 76.17 | 115.79 | 163.15 |
| SRL-k=5 | 5,250,560 | **7.15** | **6.59** | 8.28 | **9.85** | **4.51** | **4.13** | 5.33 | **6.34** | **89.78** | **74.18** | 115.83 | 163.18 |

