# OpenReview forum: "RISE : Regression Imbalance Handling using Switching Experts"
_ICLR.cc/2026/Conference — Submitted to ICLR 2026_

### Official Review · Reviewer_wPda · 2025-10-30

**Soundness:** 3
**Presentation:** 3
**Contribution:** 3
**Rating:** 6
**Confidence:** 3

**Summary:**

This paper studies the problem of Deep Imbalanced Regression (DIR). The authors observe that existing DIR methods often suffer from strong distributional heterogeneity. They theoretically demonstrate that such approaches are biased in data-scarce regions. To address this issue, the authors propose RISE (Regression Imbalance handling via Switching Experts), a modular framework inspired by the Mixture-of-Experts architecture. RISE identifies underperforming regions and achieves improved performance across diverse settings.

**Strengths:**

1. The authors claim to be the first to identify the distributional heterogeneity issue in standard DIR benchmarks.
2. The paper includes theoretical analysis that strengthens the overall conclusions.
3. The proposed method is a natural extension of the theoretical findings and demonstrates improved performance.

**Weaknesses:**

1. I have some concerns regarding the novelty of the work. The relationship between RISE and prior methods is not clearly articulated, and it is unclear how RISE fundamentally differs from or improves upon existing approaches. Please see the questions below for more details.
2. The writing could be improved for clarity and organization; see the questions below.

**Questions:**

1. What is the relationship between this work and prior studies? Are there additional related works that should be discussed? How does this paper differ from previous approaches in terms of novelty and contribution?
2. In Section 4.2, T2 performs significantly better, whereas in Section 4.3, I3 achieves the best results. Could the authors provide explanations or insights into these differences?

---

> ### Author Response · Authors · 2025-11-21
>
> We would like to thank the reviewer for their thoughtful comments. Please find our response below.
>
> *Comment:*
> What is the relationship between this work and prior studies? Are there additional related works that should be discussed? How does this paper differ from previous approaches in terms of novelty and contribution?
>
> *Response:*
> We thank the reviewer for the opportunity to clarify our contributions relative to prior work. Below we summarize the most relevant families of prior work and then state the precise ways in which RISE differs.
>
>   * **1. Global reweighting and representation-regularization methods**
>       * Prior DIR approaches either address global imbalance through label distribution smoothing or cost-sensitive weighting (e.g., LDS/FDS Yang et al. (2021), Balanced-MSE Ren et al. (2022)) or improve representations via feature-space regularization (RankSim Gong et al. (2022), ConR Keramati et al. (2024), SRL Dong et al. (2025)). All of these approaches, however, assume a single global predictor and a single shared representation across the entire label space.
>      * **RISE contribution:** Our new experiments (Freeze-and-Probe and Gradient Cosine Similarity analysis; see Appendix G) reveal the limitations of this single-model assumption: different label regions often represent conflicting tasks (functional heterogeneity)  that a single monolithic model cannot jointly optimize. We formally show in Theorem 1 that global models incur a persistent heterogeneity bias and demonstrate how imbalance amplifies this bias which cannot be removed by reweighting or representation regularization.
>   * **2. Ensemble/MoE methods**
>     * RIDE (Wang et al., 2020a), BBN (Zhou et al., 2020), and related multi-expert models (Xiang et al., 2020) are standard methods used in long-tailed classification. However, their direct application to regression is non-trivial because (i) regression has no natural discrete classes for partitioning, and (ii) region boundaries must preserve label continuity.
>     * **RISE contribution:** RISE introduces a continuity-aware partitioning mechanism. By modeling the joint distribution of validation loss and label values using Gaussian Mixture Models (GMMs), we identify experts that are both performance-aware (high error) and spatially contiguous (preserving label correlation). This is a novel adaptation of divide-and-conquer principles to the continuous domain.
>     * **Novelty in training experts:** Unlike prior MoE frameworks—which typically assume balanced or uniform expert sampling—RISE introduces an performance-guided upsampling strategy that selectively amplifies underrepresented and high-error regions within each expert’s training distribution. This preserves global coverage (reducing variance) while inducing targeted specialization (reducing heterogeneity bias).
>         Theorem 2 formalizes this trade-off: upsampling increases each expert’s effective sample size $n_{eff}$ while reducing the persistent heterogeneity bias that limits monolithic regressors. This theoretical condition directly motivates RISE’s training strategy, which prior MoE methods do not address.
>   * **3. Implicit subgroup / failure discovery methods**
>     * Recent work uses held-out data to identify underperforming subgroups in classification without group annotations: Liu et al., 2021 discovers failures via loss thresholds then upweights them; Kirichenko et al., 2023 retrains classifiers on balanced held-out data; and recent methods (Bayat et al., 2025; Qiu et al., 2023) distinguish memorization from generalization. Extending these to regression is largely unaddressed as it requires (i) threshold-free identification in continuous label space, and (ii) spatially contiguous regions respecting label correlation.
>     *  **RISE contribution:** RISE extends failure discovery by adapting it to regression and moving from identification to specialized remediation. Rather than retraining one model on reweighted data, RISE creates dedicated experts for failure regions.  RISE's GMM-based approach provides both failure identification (via held-out loss, avoiding training artifacts) and expert region assignments (via continuous, threshold-free clustering in the joint (loss, label) space)
>
> **RISE is a complementary meta-framework that, grounded in our theory of functional heterogeneity, consistently improves any DIR backbone—outperforming smoothing, regularization, ensembles, and boosting under controlled capacity—and establishes new state-of-the-art results across datasets, providing a principled new direction for DIR.**

---

> > ### Author Response · Authors · 2025-11-21
> >
> > *Comment:*
> > In Section 4.2, T2 performs significantly better, whereas in Section 4.3, I3 achieves the best results. Could the authors provide explanations or insights into these differences?
> >
> > *Response:*
> > Thank you for the question. T2 and I3 operate on different stages of RISE. T2 is an expert training strategy (how to train experts), while I3 is an inference routing strategy (how to select experts at test time). T2 trains experts on the full dataset with region-specific upsampling , which reduces the high variance caused by hard partitions in T1, consistent with Theorem 2’s bias–variance analysis. I3 is an inference-time routing strategy: training the router on held-out data avoids inheriting experts’ training-set memorization and teaches it which expert actually generalizes. Thus, T2 prevents expert overfitting, while I3 prevents router overfitting—and each is optimal in its respective stage.
> > Our ablations (in Sec. 6.3 and 6.4) isolate each factor — when comparing T1 vs. T2, both use I3 routing; when comparing I1 vs. I2 vs. I3, all use T2 training strategy. Similar ablation study is conducted on Dataset B in Appendix E.3.
> > Across all the datasets best configuration of RISE uses T2+I3 strategy.
> > We will incorporate these insights to make this aspect explicit and further improve the clarity of the manuscript.
> >
> > Finally, if our response is satisfactory, we respectfully ask that the reviewer to consider increasing their score. If it is still unsatisfactory, please let us know if there are any additional questions or anything else that we can do to improve the current submission.

---

> > > ### Author Response · Authors · 2025-11-27
> > >
> > > Dear Reviewer wPda,
> > >
> > > We want to express our sincere gratitude for the time and care you’ve put into reviewing our work. We have thoughtfully addressed the questions and suggestions you raised and have incorporated these valuable insights into the revised version, which we believe further enhances the completeness of the manuscript and the generalizability of its conclusions.
> > >
> > > We understand that this is a particularly busy time and that you all have your own papers and commitments. We sincerely appreciate the time you have devoted to ours. We very much look forward to the upcoming discussions with you. Please don’t hesitate to reach out with any questions, big or small—we are happy to provide any additional clarification.

---

### Official Review · Reviewer_Grbf · 2025-10-31

**Soundness:** 3
**Presentation:** 3
**Contribution:** 4
**Rating:** 6
**Confidence:** 3

**Summary:**

The paper tackles the Deep Imbalanced Regression (DIR) problem, where target distributions are heavily skewed and standard models fail to generalize across head and tail regions. It introduces RISE, a modular Mixture-of-Experts framework that identifies regions of poor model performance using validation-loss statistics, trains specialized experts for these regions, and dynamically routes instances to the most suitable expert at inference. The approach is supported by solid theoretical analysis of bias–variance trade-offs and validated across multiple benchmarks with consistent performance gains. Overall, this is a well-written and well-structured paper with a clear problem definition, a sound hypothesis, and a technically coherent proposal.

**Strengths:**

- Paper is very well written. The problem statement and main hypothesis are well stated in the intro and sound. The experiments are guided by RQ stated at the beginning clearly setting the direction of this section and making the analysis serve the right purpose.
- Authors tackle a problem a very relevant problem that is quite often neglected, as the imbalanced learning literature focuses more on classification than regression.
- The paper makes several strong claims about heterogeneity in DIR, that any monolithic model in DIR suffers from an irreducible heterogeneity bias amplified by imbalance,  and RISE’s performance improvement over baselined and they are well analyzed and validated either theoretically or empirically.
- Proposed MOE is tuned for imbalanced regression based on errors/model complementarity (monolithic predictor) rather than feature space regions which is interesting and well justified.
- Ablation studies analyzing different training strategies (e.g., router training).

**Weaknesses:**

- Figure 3 would benefit from larger font sizes and reorganization. It is hard to read even with a large zoom (200% of original size) as the font size is too small.

- The manuscript describes RISE as “an orthogonal meta-framework” but does not clarify what orthogonality refers to. A clearer justification or rewording (e.g., “modular” or “complementary”) would improve precision.

- While the evaluation follows prior DIR work using MAE, MSE, and bMAE, it would be more informative to include imbalance-aware metrics such as SERA or Relevance-Weighted RMSE (RW-RMSE). These would better capture performance across skewed target regions and align with the paper’s focus on heterogeneity and fair regression. In particular, SERA is an interesting metric for measuring imbalanced regression problems as it is not based on target region discretization [1].

- The analysis for RQ5 is unconvincing. The random sampling approach is poorly described and the baselines are not competitive. A meaningful comparison must include boosting methods as they share similarities with the proposal (fitting new models on the errors of the previous one). That could even help highlight the benefits of the proposed MoE scheme.

- The paper uses fairness in a broad, informal sense (equal performance across label densities) rather than a principled fairness metric or definition. As such, its fairness claims seem overstated.

- In my opinion, a third option for the router training (union of Train and Val) should be considered as it has been shown to be relevant in the training of other ensemble selection approaches (see) and would make the analysis complete.

- No actual public code or dataset links are provided, so reproducibility depends on re-implementing from the description.

Refs:

[1] Ribeiro, Rita P., and Nuno Moniz. “Imbalanced Regression and Extreme Value Prediction.” Machine Learning, vol. 109, no. 9, 2020, pp. 1803-1835.

**Questions:**

1) What exactly does “orthogonal” mean in describing RISE as a meta-framework? Does it refer to architectural independence, complementarity to existing DIR baselines, or some form of objective orthogonality?
2) The manuscript refers to improved “fairness” primarily as balanced performance across label densities. Could the authors elaborate on whether this aligns with any established fairness frameworks or metrics, or clarify the intended interpretation of fairness in this context?
3) Present results with other imbalanced regression metrics such as SERA and RW-RMSE or provide a proper justification why they are not suitable for the scenarios in this paper.
4) Including a third router-training regime that uses the union of Train + Validation data could make the ablation study more complete.
5) Consider a stronger ensembling approach for RQ5 (like a boosting) and provide more details on the sampling used.

Proper answer to these questions I would definitely raise my score.

---

> ### Author Response · Authors · 2025-11-21
>
> We would like to thank the reviewer for their thoughtful comments. Please find our response below
>
> *Comment:*
> Figure 3 would benefit from larger font sizes and reorganization.
>
> *Response:*
> We thank the reviewer for the suggestion. We have updated Figure 3 in our revised manuscript.
>
> *Comment:*
> Manuscript describes RISE as “an orthogonal meta-framework” but does not clarify what orthogonality refers to.
>
> *Response:*
> We agree that “orthogonal” was unclear and have replaced it with “complementary framework” as the reviewer suggested to accurately convey that RISE can be integrated with existing DIR methods without modifying their core architectures or losses, and can systematically enhance any pre-trained DIR baseline.
>
> *Comment:*
> It would be more informative to include imbalance-aware metrics such as SERA or RW-RMSE to better capture performance across skewed target regions.
>
> *Response:*
> Thank you for the helpful suggestion to help us improve our paper. We have now added SERA [1] and Relevance-Weighted RMSE (RW-RMSE) [2] to our evaluation across all datasets, using SRL as the baseline for RISE. We use the official implementation from the IRon toolkit (default parameters) at: https://github.com/nunompmoniz/IRon. The results are consistent with our MAE/MSE findings and further support RISE’s improvements in skewed and underrepresented regions. Results using other baselines are included in Appendix J (Tables 28-30) of the revised manuscript. We will incorporate these metrics into the main paper in the camera-ready version.
> | Dataset | Baseline (SRL) |  | SRL+RISE |  |
> |--------|----------|----------|------|------|
> |        | SERA | RW-RMSE | SERA | RW-RMSE |
> | Dataset A | 80512.74 | 10.10 | **68728.10** | **9.33** |
> | Dataset B | 1113412.91 | 13.56 | **1061609.02** | **13.25** |
> | STS-B | 327.53 | 1.18 | **288.28** | **1.11** |
>
> *Comment:*
> Analysis for RQ5 is unconvincing .. random sampling approach is poorly described .. meaningful comparison must include boosting methods.
>
> *Response:*
> In our original RQ5 setup, each ensemble member is trained on an independently sampled bootstrap subset of the training data (standard bagging). The purpose of this baseline was to compare RISE against high-capacity but unstructured ensembles. However, per the reviewer’s suggestion, we have now also included structured ensembles—boosting methods—which provide a stronger and more relevant comparison. Following [3] the boosting model uses a frozen ResNet-50 backbone with sequential expert heads, where each expert is trained on adaptively reweighted samples (based on prediction error) and final predictions are combined via learned alpha weights. We trained boosting ensembles with k=3 and k=5 experts using SRL as baseline on Dataset-A and report below.
>
>  Method | Additional Parameters | L1 (MAE) ↓ |  |  |  |
> |--------|------------------|-----|-----|-----|-----|
> |        |                  | All | Many | Med | Few |
> | SRL-Baseline | 0 | 7.23 | 6.64 | 8.28 | 9.85 |
> | SRL+RISE | +2,100,224 | **6.57** | **6.16** | **7.36** | **8.30** |
> | SRL-k=3 | +3,150,336 | 7.17 | 6.61 | 8.28 | 9.84 |
> | SRL-k=5 | +5,250,560 | 7.15 | 6.59 | 8.28 | 9.85 |
>
> We observe that boosting improves over vanilla bagging and over the base regressor in the All and Many regions, but its gains on the Medium and Few regions remain marginal. In contrast, RISE consistently outperforms all boosting configurations across all bands—even with substantially fewer additional parameters. Similar results on LDS-FDS as baseline is added in Appendix K of our revised manuscript. These results strengthen RQ5 by showing that RISE’s improvements stem from its principled architecture (region discovery + specialization + routing), not from parameter count or generic ensemble effects.
>
> [1] Ribeiro et al., Imbalanced Regression and Extreme Value Prediction, Machine Learning, vol. 109, 2020
>
> [2] Moniz et al., Prediction and ranking of highly popular web content, University of Porto, 2017
>
> [3] Solomatine et al., Adaboost rt: a boosting algorithm for regression, IEEE international joint conference on neural networks, 2004

---

> > ### Author Response · Authors · 2025-11-21
> >
> > *Comment:*
> > Paper uses fairness in a broad, informal sense (equal performance across label densities) ... elaborate on whether this aligns with any established fairness frameworks ... clarify intended interpretation of fairness in this context.
> >
> > *Response:*
> > We clarify that our use of “fairness’’ refers to equitable performance across the continuous label range, not to fairness with respect to protected attributes. In our setting, fairness means minimizing performance disparity between data-dense (head) and data-sparse (tail) regions of the target variable. This corresponds to outcome-based performance parity in regression, where segments of the label space act as subpopulations for which a model should provide comparable utility rather than systematically underperforming in the hardest or least frequent regions.
> >
> > Unlike classification, where groups are predefined, regression naturally induces subpopulations through contiguous segments of the target variable. In many real-world continuous prediction tasks, extreme target values often correspond to high-consequence cases—e.g., rapid changes in clinical risk, peak-load conditions in energy systems, or rare but critical environmental events. Performance concentrated only near the distribution’s center may therefore miss the most consequential regions. This motivates evaluating and improving performance parity across label-defined regions, which is the notion of fairness we adopt.
> >
> > Following established practice in DIR literature, we report the following metrics for fairness in our paper throughout: (1) Geometric Mean Error (GMEAN) (Yang et al., ICML'21), which characterizes the fairness (uniformity) of model predictions using the geometric mean instead of arithmetic mean over prediction errors; (2) Balanced-MAE (bMAE) (Ren et al., CVPR'22), which averages MAE over uniform label bins, directly measuring regional disparity; and (3) Region-specific performance on Many/Medium/Few frequency bands, explicitly showing disparity reduction.
> >
> > We acknowledge that standard Many/Medium/Few frequency bands don't directly reflect specific semantic groups (e.g., Few band includes labels $<17$  and  $>82$ in Dataset A), but they still represent meaningful data-sparse groups that warrant improved performance. To make societal implications more explicit we now additionally report performance on semantically contiguous groups. We evaluate RISE on Dataset A and B by dividing the datasets in three groups old ($y\ge 80$), adult ($y \in [18, 80)$ ) and young ($y < 18$). To strengthen the fairness evaluation for each of these groups ($D_g$) we calculate MAE, SERA and Worst-case Disparity (WCD). WCD [1] used in toolkits like AI Fairness-360, measures the maximum error gap across groups, ensuring the model does not concentrate large errors on any subgroup, similar to [2].
> >
> > | Method | Dataset | SERA-Young | SERA-Adult | SERA-Old | MAE-Young | MAE-Adult | MAE-Old | WCD |
> > | :--- | :--- | :---: | :---: | :---: | :---: | :---: | :---: | :---: |
> > | SRL | A | 15880 | 47009 | 17623 | 10.01 | 6.86 | 9.36 | 3.15 |
> > | **SRL+RISE** | A | **13874** | **41224** | **13629** | **8.90** | **6.26** | **7.57** | **2.65** |
> > | SRL | B | 281178 | 640468 | 191766 | 9.63 | **7.09** | 13.54 | 6.46 |
> > | **SRL+RISE** | B | **270906** | **604839** | **185863** | **8.73** | 7.17 | **12.89** | **5.72** |
> >
> > Result using other baselines is presented in Appendix J (Tables 31,32) in revised manuscript. Across both datasets, we consistently observe that SERA decreases and WCD is reduced when using RISE compared to the corresponding baseline.
> >
> > [1] Dwork et al., Fairness through awareness. Proceedings in theoretical computer science, 2012
> >
> > [2] Sagawa et al., Distributionally Robust Neural Networks for Group Shifts, ICLR 2020

---

> > > ### Author Response · Authors · 2025-11-21
> > >
> > > *Comment:*
> > > Including a third router-training regime that uses the union of Train + Validation data could make the ablation study more complete.
> > >
> > > *Response:*
> > > We thank the reviewer for this valuable suggestion. We evaluated routers trained on (i) training data only, (ii) held-out validation data (our proposed approach), and (iii) the union of training and validation data, and presented in table 7 in our main manuscript.
> > >
> > > | Method | Router Acc. | All | Many | Med | Few |
> > > | :--- | :---: | :---: | :---: | :---: | :---: |
> > > | Train-based router | 0.43 | 7.26 | 6.61 | 8.34 | 10.33 |
> > > | **Held-out val (proposed)** | **0.87** | **6.57** | **6.16** | **7.36** | **8.30** |
> > > | Train+Val union router | 0.45 | 7.24 | 6.65 | 8.32 | 9.98 |
> > >
> > > The train+val union router shows slight improvement over the train-based router (routing accuracy increases from 43% to 45%, MAE decreases from 7.26 to 7.24), demonstrating that additional data provides marginal benefit. However, proposed held-out validation router (87% accuracy, 6.57 MAE) outperforms both. This demonstrates that routers trained on the training set cannot effectively learn which expert generalizes best, while held-out data enables true meta-learning by providing an unbiased signal of generalization performance. Complete details, including additional router configuration variants (balanced sampling, train+val combinations) and analysis on held-out data importance, are provided in the revised manuscript Appendix I.
> > >
> > > *Comment:*
> > > No actual public code or dataset links are provided, so reproducibility depends on re-implementing from the description.
> > >
> > > *Response:*
> > > All datasets used in our experiments are standard DIR benchmarks widely used in prior work (e.g., Yang et al., 2021). Specifically: Dataset A corresponds to Moschoglou et al. (2017), Dataset B to Rothe et al. (2018), and STS-B to Cer et al. (2017). Following confidentiality requirements, we anonymize Dataset A and Dataset B by omitting their names in the submission, but they are fully public and easily accessible. We will release our full code implementation and include a GitHub link in the camera-ready version to ensure complete reproducibility.
> > >
> > > Finally, if these clarifications address the concerns, we would be grateful if the reviewer could kindly reconsider their score. If any issues remain, we welcome further questions or suggestions to help strengthen the submission.

---

> > > > ### Comment · Reviewer_Grbf · 2025-11-22
> > > > **Thanks for addressing the points raised during  the revision**
> > > >
> > > > I would like to thank the authors for carefully answering/addressing the questions I raised during the revision. The new results and other clarifications (like the orthogonality claim) were addressed. As my main concerns were solved, I decided to raise the scores and recommend the acceptance of this paper.

---

> ### Author Response · Authors · 2025-11-27
>
> Dear Reviewer Grbf,
>
> Thank you! We greatly appreciate your time and effort in reviewing our work.
> We hope you enjoyed reading our work as much as we enjoyed conducting it.

---

### Official Review · Reviewer_aGaM · 2025-10-31

**Soundness:** 2
**Presentation:** 3
**Contribution:** 2
**Rating:** 4
**Confidence:** 4

**Summary:**

The authors propose RISE, a modular Mixture-of-Experts (MoE)–style framework motivated by the empirical observation that standard imbalanced regression benchmarks exhibit strong distributional heterogeneity. The method first identifies underperforming label regions based on validation loss, then trains specialized experts with targeted upsampling.
The authors conduct experiments on three benchmark datasets and report notable improvements on Dataset A (AgeDB-DIR) when integrating RISE with existing baseline methods.

**Strengths:**

The idea of addressing DIR through the lens of distributional heterogeneity is interesting, and it is technically reasonable to apply a MoE framework based on validation loss. The experimental results on AgeDB-DIR (Dataset A) are impressive, showing substantial improvements when RISE is integrated with existing baselines, and supported by ablation studies.

**Weaknesses:**

### 1. Distributional heterogeneity
The notion of "distributional heterogeneity" is defined through model performance rather than as an inherent property of the data. The regions identified as heterogeneous are simply those where the baseline model performs poorly, which could result from underfitting or insufficient representation in the training set rather than distinct conditional distributions. Such "heterogeneity bias" may also come from the few-shot regions being under-sampled from the true data distribution, rather than from intrinsic heterogeneity. Framing it as heterogeneous data regression thus feels overclaimed, as true heterogeneous-data modeling is a different problem scope.

Also, in Figure 2, though the opposite trend is observed on training vs. test data for Dataset A, the trend is nearly identical for Dataset B across both training and test sets. This contradicts the claim in lines 66–69 regarding the distinct conditional distributions and thus weakens the empirical evidence for distributional heterogeneity for DIR datasets.

### 2. Flawed frequency-based approach
I agree that frequency-based methods such as over- or under-sampling are often suboptimal, mainly because they can lead to overfitting on minority regions or underfitting on majority ones. However, the argument in the manuscript, based on Table 2, that "model performance is not strictly inversely proportional to frequency" is not convincing. The order of the errors still follows the same ordering as the sample frequencies, suggesting that frequency is a strong explanatory factor. The claim would be more credible if supported by a quantitative correlation plot or a continuous analysis across the entire label range, rather than only coarse adjacent bands without statistical validation or finer binning.

Also, RISE-train also incorporates targeted up-sampling seemingly not consistent with the claim here.

### 3. Experiment
The authors claim SOTA performance for DIR, but as shown in Table 9 (Appendix), adding RISE does not consistently yield the best performance on Dataset B or STS-B. The improvements is significant only on Dataset A (AgeDB-DIR), while the gains on other datasets are marginal or almost same after applying RISE to other baselines.

Moreover, as illustrated in Figure 6 of [1], Dataset A is relatively "less imbalanced" compared to the other benchmarks, which may partially explain why RISE performs particularly well on that dataset but not on more severely imbalanced ones.


[1] Yuzhe Yang, Kaiwen Zha, Yingcong Chen, Hao Wang, and Dina Katabi. Delving into deep imbalanced regression, ICML 2021

**Questions:**

- Please see my comments in Points 1 and 2 of the Weaknesses section above.

- The best empirical number of experts is reported as three. Is this choice related to the three label regions (many-, medium-, and few-shot) defined in the experiments? Do the routers actually learn to assign samples from similar or adjacent label regions to the same expert, or is the assignment largely random?

---

> ### Author Response · Authors · 2025-11-20
>
> We would like to thank the reviewer for their thoughtful comments. Please find our response below
>
> *Comment:*
> Notion of "distributional heterogeneity" is defined through model performance ... rather than distinct conditional distributions.
>
> *Response:*
> We fully agree with the reviewer that poor performance alone cannot define “distributional heterogeneity,” and we clarify that RISE does not assume heterogeneity from performance patterns. The validation-loss analysis in RISE is used only as a data-driven diagnostic tool to reveal where a single shared regressor experiences persistent training instability. This occurs because, under label imbalance, the head region dominates gradient updates. When the gradients from head and tail regions are conflicting (refer global comment above), the shared model is repeatedly pulled toward the head optimum, preventing tail-specific patterns from being learned. This gradient-dominance effect is a plausible driver of the systematically elevated validation loss in the tail region.  Thus, validation-loss (or baseline model’s under-performance) hotspots serve as hypotheses about where optimization conflicts occur, not as a definition of heterogeneity or as an assumption about the underlying data distribution. This distinction ensures that our use of validation loss is methodological rather than definitional, and does not presuppose intrinsic heterogeneity in the data.
>
> *Comment:*
> Such "heterogeneity bias" may come from the few-shot regions being under-sampled ... rather than from intrinsic heterogeneity
>
> *Response:*
> We agree this is a critical distinction. We have added a global comment section and Appendix G in our revised manuscript with two new experiments to address this concern.
>
>  * 1. Our 'Freeze-and-Probe' experiment isolates this variable. It compares a general-feature model (Model B) against a Head-feature model (Model A-Probe) on the exact same scarce tail data, using identical L2 regularization and early-stopping. We found definitive Negative Transfer. Since the “under-sampled” (or tail) data was identical for both, the performance gap cannot be a sampling artifact; it must be an intrinsic property of the features.
>  * 2. Our 'GCS' experiment provides complementary proof. It uses balanced batches (removing the imbalance effect) and still finds strong, persistent Optimization Conflict (GCS ≈−0.5).
>
> This 'functional incompatibility' is a measurable, empirical property of the tasks themselves, not an artifact of scarcity or sampling bias. This is the 'intrinsic heterogeneity' (which we have now re-termed 'functional heterogeneity') that our paper addresses.
>
> *Comment:*
> Framing it as heterogeneous data regression feels overclaimed ... different problem scope
>
> *Response:*
> We appreciate the reviewer’s observation and clarify that our work does not claim heterogeneous-data regression in the classical sense (e.g., multi-modal inputs or fundamentally different data sources).  Rather, our results show that standard DIR benchmarks exhibit imbalance-induced functional conflict: different regions of the label space produce incompatible predictive signals under a shared model. This is demonstrated through two experiments above which rule out underfitting or insufficient representation as explanations. Thus, the challenge is not “heterogeneous-data regression,” but the inability of a monolithic model to jointly satisfy conflicting gradients arising from head and tail regions. To avoid overclaiming, we refine our terminology from ‘distributional heterogeneity’ to the more precise ‘functional heterogeneity,’ meaning differences in the conditional distribution $P(y\mid x)$ —equivalently, differences in the region-specific predictive functions—across label regions. This functional heterogeneity directly induces the optimization conflict that our experiments reveal and our RISE addresses.
>
> *Comment:*
> In Fig. 2 opposite trend is observed .. thus weakens the empirical evidence for distributional heterogeneity ..
>
> *Response:*
> We agree that heteroscedastic error alone does not constitute definitive evidence of heterogeneous $P(y\mid x)$. Our intention was not to use Fig. 2 as a proof of heterogeneity, but rather as a consequences of this heterogeneity highlighting that a single global model exhibits instability across label regions: the variance in errors on test (Few-bin exhibiting significantly higher variance than many-bin) indicate that the monolithic regressor does not behave uniformly across the label space. We agree this can also arise from scarcity or overfitting. We will remove this variance analysis and replace its motivational role with new tables summarizing the Freeze-and-Probe and GCS results in the final paper, which provide stronger empirical evidence for our claim. These experiments show both datasets exhibit functional heterogeneity—head and tail give divergent optimization signals—explaining monolithic model failures, despite differing train/test variance patterns.

---

> ### Author Response · Authors · 2025-11-20
>
> *Comment:*
> Flawed frequency-based approach ... claim would be more credible if supported by a quantitative correlation plot .. across the entire label range
>
> *Response:*
> Yes, the order of the errors still follows the same ordering as the sample frequencies. However, our claim is not that frequency is irrelevant, but rather that it is an imperfect proxy for where models struggle. [1] provide exactly the kind of statistical validation the reviewer requests in Fig. 2 (Sec. 3.1) of their manuscript. They demonstrates that while classification exhibits a strong frequency-error correlation (-0.76), regression shows a significantly weaker correlation (-0.47) despite identical label distributions. This continuous, per-label analysis demonstrates that frequency alone does not explain error behavior in DIR. As they note, regression violates the independence assumptions behind frequency-based methods: neighboring targets share information and features, so the empirical label histogram does not reflect the model’s effective sample density. We have added this reference to our revised manuscript to strengthen and clarify our claim.
>
> *Comment:*
> Also, RISE-train incorporates up-sampling ... not consistent with claim.
>
> *Response:*
> RISE-train does use upsampling, but the key difference is how regions are selected. Frequency-based rebalancing amplifies all low-frequency bins uniformly, without considering neighboring label correlations or the model's actual performance patterns, which can be unreliable in regression because empirical frequency does not reflect the model’s effective difficulty [1]. In contrast, RISE applies upsampling only within validation-identified failure regions, i.e., where the model actually struggles - discovered by analyzing the joint distribution of validation loss and label values using our GMM approach. Thus, upsampling in RISE is performance-driven rather than frequency-driven, and is fully consistent with our claims.
>
> *Comment:*
> As shown in Table 9, adding RISE does not consistently yield the best performance on Dataset B or STS-B
>
> *Response:*
> We appreciate the reviewer’s observation and clarify two points. First, RISE is a modular framework whose effectiveness depends on the router’s ability to assign inputs to the appropriate expert. As discussed in Sec. 6.2 and formalized in Theorem 2, RISE yields improvements when the reduction in heterogeneity bias exceeds the associated variance cost—an effect that emerges once routing error is sufficiently low. Router accuracy, in turn, depends on the quality of the feature representations produced by the underlying base regressor, since these features form the router’s input.
>
> In Table 9, RISE uses the same backbone features as the baseline model for router inputs. In contrast, Appendix E.4 (Table 20) shows that when the router is trained on stronger feature representations, routing accuracy improves substantially and RISE yields correspondingly larger gains—often significantly outperforming the baseline across datasets. This supports the theoretical prediction that routing quality is a key determinant of RISE’s effectiveness.
>
> Second, as per Table 20, RISE achieves statistically significant improvements across datasets, though the magnitude of these gains varies by region. On Dataset A and STS-B, RISE improves the base regressor across overall metrics and across the Many/Medium/Few bands. On Dataset B, RISE produces statistically significant improvements in the Few and Medium regions, while performance in the Many region is already close to saturation for several strong baselines, resulting in overlapping confidence intervals and thus smaller changes in the overall MAE. Despite this, the improvements in data-sparse regions remain substantial. Importantly, RISE enhances tail and medium performance without degrading the Many region, directly addressing the classic head–tail tradeoff.
>
> [1] Yang et al., Delving into deep imbalanced regression, ICML 2021

---

> ### Author Response · Authors · 2025-11-20
>
> *Comment:*
> As illustrated.. Dataset A is relatively "less imbalanced" .. which may partially explain why RISE performs well on that dataset but not on more severely imbalanced ones.
>
> *Response:*
> We appreciate the reviewer’s observation. We measure imbalance using the standard Many/Few frequency ratio. Dataset A has a moderate ratio (=50), Dataset B a very high ratio (=1197), and STS-B the lowest (=17). If RISE’s gains were primarily driven by imbalance severity, we would expect improvements to decrease as this ratio increases. Instead, Dataset A (moderate imbalance) shows the largest gains, STS-B (low imbalance) shows moderate gains, and Dataset B (high imbalance) still benefits in the Medium and Few regions.
> Our experiments show that RISE is most effective when routing accuracy is high enough for bias reduction to outweigh variance costs. Because routing accuracy depends on feature quality, Dataset A’s stronger representations enable more reliable expert assignment and larger gains. Thus, RISE’s improvements are driven more by representation quality and routing reliability than by imbalance ratio alone.
>
> *Comment:*
> Best empirical number of experts is reported as three. Is this choice related to the three label regions .. in the experiments?
>
> *Response:*
> We thank the reviewer for raising this point. The choice of three experts is not tied to the Many/Medium/Few frequency bands used for reporting results. The Many/Medium/Few bins are fixed evaluation categories defined purely by label frequency ([>100, 20-100, <20] samples per label respectively).
> In our method, the expert count $K'$ is selected through validation via the trade-off described in Theorem 2: too few experts underfit heterogeneous regions, while too many fragment the data and increase variance. The empirical optimum of $K'=3$ reflects this bias–variance balance rather than any alignment with evaluation bins. Please refer to the ablation study in Sec. 6.3 (Table 5) and Sec 6.5 in our main paper. As detailed in Appendix D.2, optimal experts count (K') can vary by datasets and baseline models. For instance, in STS-B ideal K'=2 for RankSim and K'=3 for LDS+FDS and SRL, demonstrating RISE adapts to each dataset's specific failure patterns.
>
> *Comment:*
> Do the routers actually learn to assign samples from similar or adjacent label regions to the same expert, or is the assignment largely random?
>
> *Response:*
> Regarding whether the router learns meaningful structure: yes, the routing is far from random. As shown in Fig. 4b the regions discovered by RISE-Identify are contiguous in the label space because the Gaussian Mixture Model is fit over (validation loss, label value) pairs, which enforces both performance similarity and label proximity. The router trained on these regions reflects this structure: because the regions discovered by RISE-Identify are contiguous in label space, the learned routing function assigns inputs from neighboring label ranges to the same expert and separates assignments across distant ranges. This produces coherent, structured partitions rather than random or fragmented assignments, as also evidenced by the low routing error reported in Table 20 in Appendix.
>
> Finally, if our response is satisfactory, we respectfully ask that the reviewer to consider increasing their score. If it is still unsatisfactory, please let us know if there are any additional questions or anything else that we can do to improve the current submission.

---

> > ### Author Response · Authors · 2025-11-27
> >
> > Dear Reviewer aGaM,
> >
> > We greatly appreciate your time and effort in reviewing our work.
> > We have carefully addressed the questions and suggestions you raised and have incorporated these valuable insights into the revised version, which we believe further enhances the completeness of the manuscript and the generalizability of its conclusions.
> >
> > We understand that this is a particularly busy time and that you all have your own papers and commitments. We sincerely appreciate the time you have devoted to ours. We very much look forward to the upcoming discussions with you. Please don’t hesitate to reach out with any questions, big or small—we are happy to provide any additional clarification.

---

### Official Review · Reviewer_mmVS · 2025-11-01

**Soundness:** 2
**Presentation:** 3
**Contribution:** 2
**Rating:** 4
**Confidence:** 4

**Summary:**

This paper introduces RISE, a framework designed to tackle the problem of Deep Imbalanced Regression. The authors argue that the core challenge in DIR is not just an imbalance in label distribution, but the distributional heterogeneity, where different regions of the label space have fundamentally different relationships between inputs x and outputs y. The paper argues that a single  "monolithic" model is biased towards the dominant head regions and cannot capture these distinct functions.
RISE addresses this by replacing the monolithic model with a Mixture-of-Experts. The paper use a validation loss of a pre-trained baseline model to automatically discover contiguous "failure regions" in the label space. It then trains a set of specialized experts. Finally, a gating network, trained on the validation set, learns to route new inputs to the most appropriate expert. The paper provides theoretical justification for this approach and demonstrates empirically that RISE improves performance over state-of-the-art DIR methods across multiple benchmarks.

**Strengths:**

- Novel problem formulation and motivation.
- The presented method  is modular and model-agnostic, designed as a ‘meta-framework’ that can be built on top of existing DIR methods.
- The paper is well-structured and easy to follow.

**Weaknesses:**

1. The evidence for "fundamentally different predictive functions" is somewhat weak. The cosine similarities between weight vectors for "many" and "medium" regions are also very low, which dilutes the argument that the conflict is specifically a head-vs-tail problem. A more convincing experiment, such as pre-training on the "many" region and fine-tuning on the "few" region, would be needed to truly test feature transferability and the "fundamental shift" claim.

2. The evidence presented in Figure 2 is debatable. **The stated values (e.g., 24.7 for "few" in Dataset A test) do not appear to visually match the bar heights. More importantly, the contrasting behavior between Dataset A (low test error few, high test error in many) and Dataset B (high test error in few than the test error in many) undermines the narrative about heteroscedasticity.** This reduces confidence in the paper's claims based on this figure. As Figure 2 is the main evidence for the claims, I suggest the authors to make it bigger and vectorized.

3. The paper main  weakness is its experimental methodology. The router network and region-discovery mechanism are both trained on the held-out validation set. This provides the RISE framework with access to data that the baseline models (SRL, LDS+FDS, etc.) do not use for training, making the main performance comparison in Table 4 an unfair. A cleaner, more valid approach would require all methods to have access to the same information. For example, a portion of the training set should be held out to train the RISE router, ensuring a fair comparison on the common, unseen test set.

4.   Theorem 1 correctly identifies a persistent bias, but in practice, for extremely scarce tail regions, high variance is often the dominant source of error (due to small n). Furthermore, related (and unreferenced) work has also discussed the bias-variance trade-off in imbalanced regression.

5. The language in the introduction (e.g., "mastering each sub-problem") need to be weakened. Furthermore, the research gap identified for some prior work ("lack mechanisms to detect minority regions or train dedicated experts") is what the authors proposed not really a research gap.

**Questions:**

See Weaknesses.

---

> ### Author Response · Authors · 2025-11-20
>
> We would like to thank the reviewer for their thoughtful comments. Please find our response below
>
> *Comment:*
> The evidence for "fundamentally different predictive functions" is somewhat weak.
>
> *Response:*
> Thank you for raising this point. We have added a dedicated global rebuttal section and Appendix G in our revised manuscript with two new experiments that directly address this concern.
>
>  * 1.  Freeze-and-Probe: A Controlled Pretrain→Transfer Test
>
> The reviewer suggested pretraining on the “many/head’’ region and fine-tuning on the “few/tail’’ region. Direct fine-tuning, however, is inherently confounded because the tail regime is extremely small (samples $<1k$)—full-model fine-tuning leads to catastrophic overfitting, making the results uninterpretable (poor tail or lost head performance could arise solely from overfitting).
> To honor the reviewer’s intent while removing this confound, we designed a more controlled "Freeze-and-Probe" experiment. With the backbone frozen and identical scarce tail data, the head-pretrained backbone performs worse than a generic ImageNet backbone—showing negative transfer. Since model capacity, data, and regularization are matched, this provides unconfounded evidence that head-optimized features are functionally incompatible with tail prediction.
>
>  * 2. Gradient Cosine Similarity (GCS): Evidence of Optimization Conflict
>
> Using a classification surrogate to remove the MSE residual-sign confound, we observe strong, persistent negative gradient alignment between head and tail batches early in training. This shows that the two regions impose opposing optimization directions, independent of performance gaps or tail overfitting.
>
> *Comment:*
> The stated values ... do not visually match the bar heights
>
> *Response:*
> Thank you for pointing this out. The discrepancy arises because the y-axis in Fig. 2 is plotted on a logarithmic scale, so the bar heights are not visually proportional to the raw numeric values.
>
> *Comment:*
> Contrasting behavior between Dataset A (low test error few, high test error in many) and Dataset B (high test error in few than the test error in many).
>
> *Response:*
> We thank the reviewer for the careful examination. To clarify: in Dataset A (test), the error variance in the Few region (74.7) is higher than in the Many region (36.5), and similarly in Dataset B (test), the Few region (458.6) exceeds the Many region (65.7). We apologize for the mislabeling and the y-axis configuration in the figure, which caused confusion.
>
> Regarding train–test patterns: both datasets exhibit heteroscedastic test errors (higher variance in the Few region), but their train/test trends differ—Dataset A shows a flip (low train → high test variance in Few), while Dataset B shows consistently high variance in Few across both splits. As this figure has caused confusion and is not central to our contribution, we will remove it in the revised version (see comment below).
>
>
> *Comment:*
> Undermines the narrative about heteroscedasticity ..reduces confidence in the paper's claims
>
> *Response:*
> We agree that heteroscedastic error alone does not constitute definitive evidence of heterogeneous $P(y\mid x)$. Our intention was not to use Fig. 2 as a proof of heterogeneity, but rather as a consequences of this heterogeneity, that a single global model exhibits instability across label regions: the variance patterns on test-set (Few-shot bin exhibiting significantly higher variance than the Many-shot bin) indicate that the monolithic regressor does not behave uniformly across the label space. We agree this can also arise from scarcity or overfitting.
>
> We will remove Figure 2 and will replace its motivational role with new tables presenting direct evidence from our Freeze-and-Probe and GCS experiments (Appendix G) in the final paper.
>  * Freeze-and-Probe: Head-pretrained features perform markedly worse than ImageNet features on identical tail data (≈33% on Dataset A and ≈24% on Dataset B). Because data, capacity, and training conditions are matched, this indicates that features optimized for head regions do not transfer well to tail regions—evidence of functional incompatibility that is not attributable to scarcity or overfitting.
>  * GCS: We observe persistent negative gradient alignment (≈−0.5 in deeper layers) across both datasets under balanced sampling, indicating opposing optimization directions between head and tail regions. This conflict appears irrespective of imbalance levels.
>
> Together, these results show that both datasets exhibit functional heterogeneity—i.e., the two regions induce divergent optimization signals—which helps explain why monolithic regressors struggle across datasets despite differing train/test variance patterns.

---

> ### Author Response · Authors · 2025-11-20
>
> *Comment:*
> The paper main weakness is its experimental methodology
>
> *Response:*
> We appreciate the reviewer's concern. To be clear: no expert weights are trained using validation data—the validation set is used only for meta-learning tasks (region discovery and router training). This follows standard ML practice in meta-learning [1], and recent work showing held-out data enables post-hoc model improvement [2,3]. Consistent with these works, RISE uses a held-out set to obtain an unbiased estimate of generalization behavior—both for identifying underperforming regions and for training a router that selects the expert with the strongest generalization performance.
> To address the reviewer’s point empirically, we ran two experiments for Dataset A:
>
> 1. Train+Val Baseline: We retrained SRL on Train + Held-out data, giving the baseline an advantage, while RISE uses held-out data for meta-learning tasks only (backbone and expert parameters are updated using Train data). This yields marginal improvement over baseline SRL trained on 100% train data (7.18 vs 7.23 MAE) but worse than SRL+RISE (6.57 MAE).
> 2. 80-20 Train Split: We followed the reviewer’s recommendation by partitioning the original training set into an 80-20 subset. Baseline SRL was trained strictly on the 80% training split. On this baseline we train RISE—experts with 80% subset, while region discovery and router training uses 20% subset. Under this setup—where RISE and baselines receive identical training data—RISE still outperforms all baselines, including SRL trained on the full 100% train set. We report MAE in the below table.
>
> | Method | Overall | Many | Med | Few |
> | :--- | :---: | :---: | :---: | :---: |
> | **Standard validation set approach:** | | | | |
> | SRL (100% train) | 7.23 | 6.64 | 8.28 | 9.85 |
> | + RISE | **6.57** | **6.16** | **7.36** | **8.30** |
> | SRL (Train + Held-out) | 7.18 | 6.62 | 8.15 | 9.84 |
> | | | | | |
> | **Train-split approach (Reviewer's suggestion):** | | | | |
> | SRL (80% train) | 7.37 | 6.65 | 8.84 | 10.53 |
> |  + RISE | **6.88** | **6.36** | **7.73** | **9.51** |
>
> Full table containing other metrics and using LDS-FDS as baseline is added in Appendix H of revised manuscript. These results demonstrate that RISE’s improvements arise from its MoE based architecture and principled identification and targeting of underperforming regions, not from increased sample access.
>
> *Comment:*
> Theorem 1 correctly identifies a persistent bias .. source of error (due to small n)
>
> *Response:*
> We agree with the reviewer that in finite samples—especially in extremely scarce tail regions—the estimation variance term can indeed dominate the error. Our theorem does not claim otherwise. Theorem 1 isolates the population-level decomposition under linear regression and shows that, even when the variance term vanishes with large
> n a persistent bias remains whenever region-specific predictors $w_k^*$ differ. This provides a clean analytical explanation of functional heterogeneity
>
> *Comment:*
> related (unreferenced) work .. in imbalanced regression
>
> *Response:*
> We will cite these prior work [4,5] which has examined variance-dominated failure in scarce regions, but these works are primarily empirical. Our contribution is complementary: Theorem 1 gives a closed-form decomposition for pooled OLS under region-specific predictors, separating an estimation-variance term (vanishing with n) from a persistent bias term. To our knowledge, prior DIR work has not provided this explicit analytic characterization of bias stemming from regionwise functional mismatch.
>
> *Comment:*
> Language in the introduction.. need to be weakened
>
> *Response:*
> We have replaced "mastering each sub-problem” with “learning specialized solutions for each sub-problem” in Sec 2 to avoid overstating the capability of experts.
>
> *Comment:*
> Furthermore, the research gap identified for some prior work ..
>
> *Response:*
> Our intention was not to assert a broad research gap, but to note a specific limitation of regression-via-classification methods  (Pu et al., 2025; Xiong & Yao, 2024): they discretize labels into fixed bins and do not automatically identify coherent underperforming regions based on model behavior—a key challenge in continuous regression, where naïve binning often yields scattered groups. While minority subgroup detection is actively studied in classification, its extension to continuous labels remains largely unaddressed. We revised Sec. 2.1 to make this distinction more measured while preserving the motivation for RISE.
>
> [1] Kirichenko et al., Last Layer Re-Training, ICLR 23
>
> [2] Bayat et al., Pitfalls of Memorization, ICLR 25
>
> [3] Ghaznavi et al., Exploiting What Trained Models Learn, ICLR 25
>
> [4] Ribeiro et al., Imbalanced Regression extreme value, Machine Learning, vol. 109, 2020
>
> [5] Yang et al., Delving into deep imbalanced regression, ICML 21
>
> If the concerns are addressed, we kindly request reconsideration of the score; otherwise, we are happy to provide further clarification.

---

> > ### Author Response · Authors · 2025-11-27
> >
> > Dear Reviewer mmVS,
> >
> > We greatly appreciate your time and effort in reviewing our work.
> > We have carefully addressed the questions and suggestions you raised, and have incorporated these valuable insights into the revised version, which we believe will further enhance the completeness of the manuscript and the generalizability of its conclusions.
> >
> > We know that you all have your own papers that you have to deal with during this busy time, and sincerely appreciate the time you've taken to spend on ours. We are very much looking forward to the upcoming discussions with you! Please don't hesitate to ask us any questions big or small, and we are happy to provide any further clarifications.

---

### Author Response · Authors · 2025-11-20
**Stronger evidence of functional heterogeneity in DIR**

To provide stronger evidence that the head vs. tail regions in DIR Datasets A (Moschoglou et al., 2017) and Dataset B (Rothe et al.,2018) correspond to fundamentally different predictive functions, we conducted two additional experiments (Details added in revised version of the paper in Appendix G):

### **1) Freeze-and-Probe: testing feature transferability**

The goal of this experiment is to isolate feature transferability as the only factor under study.
To do so, we fix the entire ResNet--50 backbone in both models and train only a newly initialized linear layer on the scarce Tail-Train data (e.g., label values $<15$).
By freezing all convolutional layers, we eliminate effects from forgetting, overfitting, or capacity differences, ensuring that any performance difference must arise solely from the quality of the underlying feature representation.

Both models are trained under identical conditions:
identical linear probe architecture, identical L2 regularization, identical optimization hyperparameters, and identical early stopping based on the Tail-Val set. The only difference is the source of the frozen backbone:

* **Model~B (General-Feature Baseline)**: Frozen ImageNet-pretrained ResNet50 backbone. This represents strong, general-purpose features not biased toward any label region in our dataset.

* **Model~A-Probe (Head-Feature Test):** ResNet50 backbone first fine-tuned only on the Head region (e.g., label values 20-40), then frozen.  This tests whether features specialized for the head region transfer effectively to the tail.

We train only the linear layer for both models using the same Tail-Train data and evaluate the best checkpoint (chosen via Tail-Val early stopping) on the Tail-Test set.

**Table 1: Freeze-and-Probe: Tail Test MAE Comparison**

| Dataset | ImageNet (B) | Head-pretrained (A-Probe) | Relative Drop |
| :--- | :---: | :---: | :---: |
| *Dataset A* | $3.18 \pm 0.06$ | $4.75 \pm 0.07$ | $33$% |
| *Dataset B* | $2.22 \pm 0.01$ | $2.95 \pm 0.01$ | $24$% |

On both datasets, the Head-biased features (Model A-Probe) performed significantly worse than the generic ImageNet features (Model B). Since all other variables were held constant (scarce data, probe architecture, L2 regularization, and early stopping), this performance gap cannot be attributed to scarcity or overfitting.
This is a direct, un-confounded proof of **Negative Transfer**. It demonstrates that the features learned for the Head are not merely incomplete but are fundamentally incorrect and harmful for the Tail task.

### **2) Gradient Cosine Similarity (GCS): measuring optimization conflict**
We further examined whether head and tail objectives impose conflicting gradient directions, which would imply incompatible predictive functions [1].
For completeness, we computed GCS under the standard regression loss (MSE). However, MSE gradients are mechanically confounded. (Refer Appendix G.2)

Experimental Setup:

* Following [2], we discretize the label range into 101 bins (one for each label 0-100) and replace the monolithic model's (with Resnet50 backbone) head with an $N=101$ logit output layer, using Cross Entropy Loss.
* The gradient is $\nabla \ell = (\hat{\mathbf{p}} - \mathbf{p})$, where $\hat{\mathbf{p}}$ is the softmax output and $\mathbf{p}$ is the one-hot target vector which has no residual-dependent sign flip.
* The resulting normalised GCS, $\cos(\nabla \ell_{head}, \nabla \ell_{tail})$ computed on balanced batch of head and tail data points directly measures true optimization alignment/conflict.

We compute GCS per-layer and for the final- fully connected layer (fc) by flattening the corresponding parameter gradients into vectors. We report epoch-wise means over multiple runs (10 random seeds) and over several balanced mini-batches.

**Table 1: Average GCS Comparison**

| Dataset | Epoch | layer1 | layer4 | fc |
| :--- | :---: | :---: | :---: | :---: |
| **Dataset A** | 1 | -0.0147 | -0.0527 | -0.1799 |
| | 10 | -0.0370 | -0.0707 | -0.5965 |
| | 50 | +0.0276 | -0.1183 | -0.4788 |
| | 100 | +0.0444 | -0.1914 | -0.4905 |
| **Dataset B** | 1 | +0.0283 | -0.0404 | -0.1258 |
| | 10 | +0.0132 | -0.0512 | -0.3165 |
| | 50 | +0.0154 | -0.1081 | -0.3754 |
| | 100 | +0.0121 | -0.1104 | -0.3408 |

Deeper layers (layer4, fc) show persistently negative GCS (–0.18 to –0.60), while early layers remain near zero, reflecting generic vs. region-specific features. **Persistent negative alignment means updates that reduce head loss increase tail loss, revealing incompatible optimization directions.** Similar trends are also observed during training with MSE loss as shown in revised version of our paper in Appendix G.2. This confirms that a single shared model cannot optimize both regions without explicit parameter separation.

[1] Wang et al., 2021 ICLR: Gradient Vaccine-Investigating and Improving Multi-task Optimization in Multilingual Models.

[2] Niu et al., 2016 CVPR: Ordinal regression with multiple output cnn.

---

### Author Response · Authors · 2025-12-03
**Unified Summary of Key Reviewer Concerns and How We Resolved Them [1/2]**

**1. Evidence for “functional heterogeneity”**

We added two new controlled analyses to demonstrate functional heterogeneity in DIR datasets (Appendix G of our revised manuscript and below rebuttal section): Freeze-and-Probe experiments (suggested by Reviewer mmVS) show strong negative transfer when head-trained features are applied to identical tail data under controlled conditions, ruling out data scarcity as an explanation; and Gradient Cosine Similarity (GCS) analysis (Wang et al., 2021 ICLR) reveals persistent optimization conflicts (GCS ≈ −0.3) between head and tail regions even with balanced sampling. These experiments provide direct, causal evidence that DIR datasets contain incompatible predictive functions across label regions. To clarify our terminology following Reviewer aGaM's feedback, we use "functional heterogeneity" to mean systematic differences in $P(Y\mid X)$ across head and tail regions—the precise challenge RISE addresses. We will replace Figure 2 with tables summarizing these new analyses in the introduction section (using the allowed extra page in camera-ready version).

**2. Experimental Methodology and Use of Validation Data**

All RISE experts are trained only on the same Train split as the baselines; the held out validation split is used solely for meta-learning tasks—region discovery and router training—exactly in line with standard practice in model selection and post-hoc improvement (e.g., Bayat et al., ICLR 25; Ghaznavi et al., ICLR 25). To rigorously rule out any advantage from extra data, we performed two controlled experiments (Appendix H of our revised manuscript and below rebuttal section): (1) giving SRL access to Train+Val (while RISE still trains experts only on Train) yields only marginal SRL gains, but RISE still achieves substantially lower MAE; (2) under a strict 80–20 internal split (as suggested by Reviewer mmVS)—where both SRL and all RISE experts train exclusively on the same 80% subset and RISE uses the remaining 20% only for meta-learning—RISE continues to outperform all baselines, even SRL trained on the full 100% Train set. These results verify that RISE’s gains stem from its architectural innovations (loss-guided partitioning + expert specialization), not from access to additional data.

**3. Why Frequency Is Insufficient & Why RISE’s Upsampling Is Principled**

Our claim is not that frequency is irrelevant, but that it is an imperfect proxy for model difficulty in regression. (Yang et al., ICML 2021) provide direct statistical evidence which Reviewer aGaM seeks: the frequency–error correlation is strong in classification (ρ = −0.76) but markedly weaker in regression (ρ = −0.47) under the same label distribution. This shows that empirical frequency does not reflect effective sample density when targets lie on a continuous manifold, where neighboring labels share structure.
RISE’s upsampling therefore differs fundamentally from frequency-based methods: it is performance-guided, targeting only those regions that exhibit systematically high validation loss through our joint (loss, label) GMM analysis. This avoids the indiscriminate boosting that causes head–tail over/underfitting in prior methods. Thus, RISE’s training strategy is internally consistent and grounded in both empirical evidence and the structure of continuous-label regression.

**4. Performance Variability across datasets**

RISE’s theoretical advantage depends on routing accuracy (Theorem 2), which is itself determined by the quality of backbone features available to the router. In Table 9, the router receives the same features as the baseline, limiting region separability and thus the achievable routing accuracy. As shown in Appendix E.4 (Table 20), when the router is trained on stronger feature representations, routing accuracy improves and RISE delivers consistently larger, statistically significant gains—often substantially outperforming all baselines. Across all datasets, RISE consistently yields strong gains in the Few/Medium regions and preserves or improves performance in the Many region, avoiding the typical head–tail tradeoff. Thus, the variability in Table 9 reflects that RISE gains depends on baseline feature representations.

**5. Imbalance-Aware Metrics (SERA, RW-RMSE)**

We incorporated SERA and RW-RMSE metrics as suggested by Reviewer Grbf into our evaluation across all datasets. The results (Appendix J, Tables 28–30 of our revised manuscript and rebuttal below) align with our MAE/MSE findings and consistently show that RISE improves performance in skewed and underrepresented regions which further strengthens our manuscript.

**6. Ensemble Comparisons (RQ5)**

To strengthen RQ5, we expanded our ensemble baselines to include structured boosting methods (details in below rebuttal and Appendix K of our revised manuscript). This confirms that RISE’s gains arise from its principled architecture rather than generic ensemble effects.

---

> ### Author Response · Authors · 2025-12-03
> **Unified Summary of Key Reviewer Concerns and How We Resolved Them [2/2]**
>
> **7. Fairness Clarification**
>
> Our notion of “fairness’’ concerns equitable predictive performance across the continuous label spectrum—not protected-attribute fairness. Here, fairness refers to outcome parity across label-density subpopulations (Many/Medium/Few), ensuring the model does not systematically underperform in data-sparse regions—consistent with established practice in the DIR literature. To make societal implications more explicit, we conducted one more experiment and added those details in rebuttal below and in Appendix J (Tables 31–32) of our revised manuscript.
>
> **8. Core Novelty of RISE and how it differs from prior works**
>
> To the best of our knowledge, RISE is the first framework to (i) empirically establish functional heterogeneity in DIR benchmarks —demonstrating conflicting head–tail predictive relationships and (ii) explicitly address this by decomposing the regression space through continuity-aware, performance-guided clustering in the joint (loss, label) domain and training region-specialized experts with performance-guided upsampling.
> Our theory formalizes the irreducible heterogeneity bias of monolithic regressors under imbalance (Theorem 1) and provides the derivation of conditions under which region-specialized experts provably outperform a monolithic model (Theorem 2).
> Empirically, RISE functions as a general, plug-in meta-framework that consistently improves state-of-the-art DIR methods (LDS/FDS, RankSim, SRL, etc.) as well as strong ensemble and boosting baselines under controlled capacity. By directly addressing the functional conflict between head and tail regions—a limitation not addressed by existing reweighting or representation-regularization approaches—RISE offers a principled new architectural direction for robust regression in continuous, imbalanced domains.

---

### Meta-Review · Area_Chair_72PV · 2026-01-08

**Summary:**

The primary concerns stemmed from the experimental methodology, specifically whether training the router and region discovery on held-out validation data gave RISE an unfair advantage over baselines. Reviewers also questioned the validity of 'functional heterogeneity,' arguing the evidence was weak (defined by model error rather than data properties). Also, there was skepticism regarding the method's consistency, since gains were marginal on severely imbalanced benchmarks like Dataset B and STS-B , and the novelty compared to prior boosting works was initially unclear.

**Reviewer Concerns:**

The rebuttal successfully addressed requests for imbalance aware metrics (SERA, RW-RMSE) and clarified the 'orthogonality' and fairness definitions, which satisfied Reviewer Grbf. However, the fundamental disagreement regarding the use of validation data for meta-learning remains outstanding; while the authors provided a train-split experiment, Reviewer mmVS did not validate if this fully resolved the fairness concern regarding baseline comparisons. Additionally, Reviewer aGaM's concern that the 'heterogeneity' claim is overclaimed and that performance improvements are inconsistent across datasets persists as a potential weakness in generalizability.

**Reviewer Scores:**

One reviewer (with lower confidence) explicitly mentioned that he will move to Accept the paper. Whereas other high confidence reviewers likely remain at 4. The methodological concern regarding the validation set usage is structural, and without explicit sign-off on the new train-split experiment, the skepticism about fair comparison likely persists. Another concern was about the marginal empirical gains on harder datasets and the 'overclaimed' theory:though the rebuttal explained why gains vary but did not change the empirical reality of the results.

---

### Decision · Program_Chairs · 2026-01-26

Reject